# LEARNING DYNAMICS OF LLM FINETUNING

**Yi Ren**
University of British Columbia
renyi.joshua@gmail.com

**Danica J. Sutherland**
University of British Columbia & Amii
dsuth@cs.ubc.ca

## ABSTRACT

Learning dynamics, which describes how the learning of specific training examples influences the model's predictions on other examples, gives us a powerful tool for understanding the behavior of deep learning systems. We study the learning dynamics of large language models during different types of finetuning, by analyzing the step-wise decomposition of how influence accumulates among different potential responses. Our framework allows a uniform interpretation of many interesting observations about the training of popular algorithms for both instruction tuning and preference tuning. In particular, we propose a hypothetical explanation of why specific types of hallucination are strengthened after finetuning, e.g., the model might use phrases or facts in the response for question B to answer question A, or the model might keep repeating similar simple phrases when generating responses. We also extend our framework and highlight a unique "squeezing effect" to explain a previously observed phenomenon in off-policy direct preference optimization (DPO), where running DPO for too long makes even the desired outputs less likely. This framework also provides insights into where the benefits of on-policy DPO and other variants come from. The analysis not only provides a novel perspective of understanding LLM's finetuning but also inspires a simple, effective method to improve alignment performance. Code for experiments is available at https://github.com/Joshua-Ren/Learning_dynamics_LLM.

## 1 INTRODUCTION

Deep neural networks usually acquire new knowledge by updating their parameters via gradient descent (GD). This procedure can be described by learning dynamics, which links changes in the model's predictions to the gradients generated by learning specific examples. With the help of learning dynamics, researchers have not only explained many interesting phenomena during training, e.g., the "zig-zag" learning path (Ren et al. 2022) and the formation of compositional concept space (Park et al. 2024), but used these insights to propose novel, improved algorithms in different problems (e.g. Pruthi et al. 2020; Ren, S. Guo, et al. 2023; Xia et al. 2024).

The study of large language models (LLM) is gaining popularity due to their surprising capabilities on various tasks. To ensure the LLMs follow human instructions and align well with human preferences, finetuning has attracted much recent attention. Practitioners often start with instruction tuning, where the model learns extra knowledge necessary for the downstream task, and then preference tuning, where the model aligns its outputs to human preference (Ouyang et al. 2022). Various finetuning algorithms have been proposed to fit into this pipeline, with differing explanations as to why they improve the model's performance.

Contrary to most existing analyses of LLM finetuning, which use the perspective of their training targets, their status at the end of training, or their relationships to reinforcement learning (e.g. Ji et al. 2024; Rafailov et al. 2024; Tajwar et al. 2024), this paper tries to understand LLMs' evolution from a dynamical perspective. Specifically, we formalize the learning dynamics of LLM finetuning by decomposing the change of the model's prediction into three terms which play different roles. This framework can be easily adapted to

various finetuning algorithms with different goals, including supervised finetuning (SFT, Wei et al. 2022), direct preference optimization (DPO, Rafailov et al. (2023), and its variants) and even reinforcement learning based methods (e.g., PPO, Schulman et al. 2017). This framework helps explain several interesting and counter-intuitive observations during training – including the "repeater" phenomenon after preference tuning (Holtzman et al. 2020), hallucination (L. Huang et al. 2023), the decay in confidence of *all* responses during off-policy DPO (Rafailov et al. 2024), and more.

Moreover, we also provide a new perspective on understanding why off-policy DPO and other variants underperform their on-policy counterparts (S. Guo, B. Zhang, et al. 2024). Our explanation starts by observing an interesting "squeezing effect," which we demonstrate is a consequence of gradient *ascent* (as in DPO and similar algorithms) on models with cross-entropy loss following a softmax layer. In short, for each token's prediction, the negative gradient will push down the model's predictions on (almost) all possible output labels, moving this probability mass to the most-likely labels. This can be detrimental to the alignment we are trying to achieve. This effect is most serious when the negative gradient is imposed on an already-unlikely label, which is why the confidence of almost all responses decreases during off-policy DPO. Inspired by this, we propose a simple, counter-intuitive, but very effective method to further improve alignment performance.

## 2 DEFINITION OF LEARNING DYNAMICS AND AN MNIST EXAMPLE

Learning dynamics is usually an umbrella term describing how the change of a specific factor influences the model's prediction. In this paper, we narrow down it to describe "how the change in model's parameter $\theta$ influences the corresponding change in $f_\theta$", i.e., the relationship between $\Delta\theta$ and $\Delta f_\theta$. When the model updates its parameters using gradient descent (GD), we have

$$\Delta\theta \triangleq \theta^{t+1} - \theta^t = -\eta \cdot \nabla\mathcal{L}\left(f_\theta(\mathbf{x}_u), \mathbf{y}_u\right); \quad \Delta f(\mathbf{x}_o) \triangleq f_{\theta^{t+1}}(\mathbf{x}_o) - f_{\theta^t}(\mathbf{x}_o), \tag{1}$$

where the update of $\theta$ during step $t \to t+1$ is given by one gradient update on the sample pair $(\mathbf{x}_u, \mathbf{y}_u)$ with learning rate $\eta$. In short, the learning dynamics in this paper address the question:

*After an GD update on $\mathbf{x}_u$, how does the model's prediction on $\mathbf{x}_o$ change?*

Studying the learning dynamics can shed light on many important problems in deep learning and also help to understand various counter-intuitive phenomena. Appendix A further discusses related work.

As a warm-up, we first consider a standard supervised learning problem, where the model learns to map $\mathbf{x}$ to predictions $\mathbf{y} = \{y_1, \ldots, y_L\} \in \mathcal{V}^L$, where $\mathcal{V}$ is the vocabulary of size $V$. The model usually outputs a probability distribution by first generating a matrix of logits $\mathbf{z} = h_\theta(\mathbf{x}) \in \mathbb{R}^{V \times L}$ and then takes the Softmax of each column. We can track the change in the model's confidence by observing $\log\pi_\theta(\mathbf{y} \mid \mathbf{x})$.

**Per-step influence decomposition.** The learning dynamics of (1) become,

$$\Delta\log\pi^t(\mathbf{y} \mid \mathbf{x}_o) \triangleq \log\pi_{\theta^{t+1}}(\mathbf{y} \mid \mathbf{x}_o) - \log\pi_{\theta^t}(\mathbf{y} \mid \mathbf{x}_o),. \tag{2}$$

For simplicity, we start from the $L = 1$ scenario, where the $\Delta\theta$ and $\Delta\log\pi$ can be linked by the following result, a version of a result of Ren et al. (2022) proved and further discussed in Appendix B. For multi-label classification ($L > 1$), the updates separate; we can calculate $L$ different $\Delta\log\pi^t$ and stack them together.

**Proposition 1.** *Let $\pi = \mathsf{Softmax}(\mathbf{z})$ and $\mathbf{z} = h_\theta(\mathbf{x})$. The one-step learning dynamics decompose as*

$$\underbrace{\Delta\log\pi^t(\mathbf{y} \mid \mathbf{x}_o)}_{V \times 1} = -\eta \underbrace{\mathcal{A}^t(\mathbf{x}_o)}_{V \times V} \underbrace{\mathcal{K}^t(\mathbf{x}_o, \mathbf{x}_u)}_{V \times V} \underbrace{\mathcal{G}^t(\mathbf{x}_u, \mathbf{y}_u)}_{V \times 1} + \mathcal{O}(\eta^2 \|\nabla_\theta\mathbf{z}(\mathbf{x}_u)\|_{\mathrm{op}}^2), \tag{3}$$

*where $\mathcal{A}^t(\mathbf{x}_o) = \nabla_\mathbf{z}\log\pi_{\theta^t}(\mathbf{x}_o) = I - \mathbf{1}\pi_{\theta^t}^\top(\mathbf{x}_o)$, $\mathcal{K}^t(\mathbf{x}_o, \mathbf{x}_u) = (\nabla_\theta\mathbf{z}(\mathbf{x}_o)|_{\theta^t})(\nabla_\theta\mathbf{z}(\mathbf{x}_u)|_{\theta^t})^\top$ is the empirical neural tangent kernel of the logit network $\mathbf{z}$, and $\mathcal{G}^t(\mathbf{x}_u, \mathbf{y}_u) = \nabla_\mathbf{z}\mathcal{L}(\mathbf{x}_u, \mathbf{y}_u)|_{\mathbf{z}^t}$.*

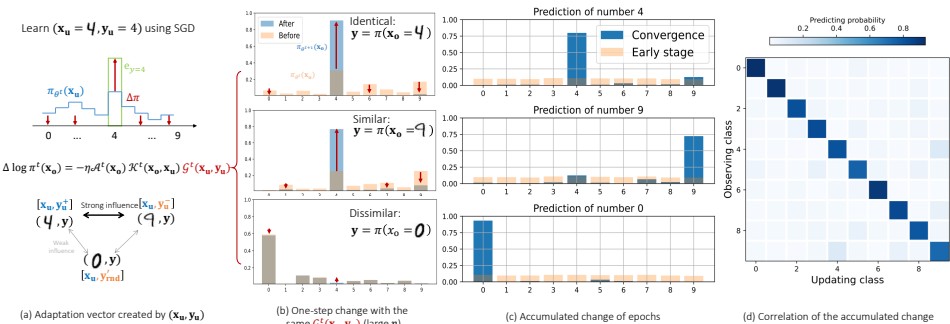

Figure 1: The per-step learning dynamics and the accumulated influence in an MNIST experiment.

$\mathcal{A}^t(\mathbf{x}_o) = I - \mathbf{1}\pi_{\theta^t}^\top(\mathbf{x}_o)$ only depends on the model's current predicted probability. The matrix $\mathcal{K}^t$ is the empirical neural tangent kernel (eNTK, Jacot et al. 2018) of the model, the product of the model's gradients with respect to $\mathbf{x}_o$ and $\mathbf{x}_u$. The analysis in this paper relies on the following assumption:

*During the training, the relative influence of learning $x_u$ on all other different $x_o$ is relatively stable.*

The common "lazy eNTK" assumption discussed in Arora et al. (2019) is a sufficient but not necessary condition for this paper. Appendix C provides a more detailed discussion and experimental verification for both MNIST and LLM settings. We can then think of $\mathcal{K}^t$ as a model-specific similarity measurement between different input samples: larger $\|\mathcal{K}^t\|_F$ means the update of $\mathbf{x}_u$ likely influences the model's prediction on $\mathbf{x}_o$ more. Finally, $\mathcal{G}^t$ is determined by the loss function $\mathcal{L}$, which provides the *energy* and *direction* for the model's adaptation. For example, for cross-entropy loss $\mathcal{L}_{\text{CE}} \triangleq -\mathbf{y}_u^\top \log \pi(\mathbf{y} \mid \mathbf{x}_u)$, we have $\mathcal{G}_{\text{CE}}^t = \pi_{\theta^t}(\mathbf{y} \mid \mathbf{x}_u) - \mathbf{y}_u$, a length-$V$ vector that points from the model's current predictive distribution to the desired supervisory distribution. For typical "hard" labels, $\mathbf{y}_u$ is a one-hot vector $\mathbf{e}_{\mathbf{y}_u}$.

**Accumulated influence and a demonstration on MNIST.** Proposition 1 describes how the update of $\mathbf{x}_u$ changes the model's prediction on $\mathbf{x}_o$ for each learning step. Since a real model updates its parameters for many steps, it is important to ask about accumulation of these per-step influences over time. We start by analyzing a simple example of training a LeNet on the MNIST dataset (LeCun et al. 1998).

See Figure 1-(a), where the network $\pi_{\theta^t}$ is updating its parameters using the loss calculated on one training example $(\mathbf{x}_u, \mathbf{y}_u = \mathbf{e}_4)$. The residual term $\mathcal{G}_{\text{CE}}^t(\mathbf{x}_u, \mathbf{y}_u)$ is then represented by the red arrows, which all start from $\pi_{\theta^t}(\mathbf{y} \mid \mathbf{x}_u)$ and point to $\mathbf{e}_4$. We can then ask how the model's predictions on different $\mathbf{x}_o$ change after this update. As in Figure 1-(b), for an $\mathbf{x}_o$ in the same class with $\mathbf{x}_u$ (i.e., the identical case), the predicted probability of this correct label is "pulled up" by this update, as expected. On the other hand, if this $\mathbf{x}_o$ is similar to $\mathbf{x}_u$ (i.e., $\|\mathcal{K}^t\|_F$ is reasonably large) but comes from another class, then the predicted probability on $\mathbf{x}_u$'s class (not the correct label of $\mathbf{x}_o$) would be "pulled up," as in the second panel of Figure 1-(b). Last, for examples that look dissimilar to $\mathbf{x}_u$ (small $\|\mathcal{K}^t\|_F$), this update will not change the model's prediction on $\mathbf{x}_o$ much, as in the bottom panel in Figure 1-(b). The interactions among the updates of different inputs then form an interesting pattern for the learned predictions. As illustrated in Figure 1-(c), when making predictions on images coming from class 4, the model tends to assign higher confidence on class 9. That is because the examples in class 9 on average look more similar to class 4 than examples in other classes. Hence the update of examples in classes 4 and 9 will reinforce their mutual influence and lead to a bump in their predictions. To verify this, we plot the average of $\pi(\mathbf{y} \mid \mathbf{x})$ for $\mathbf{x}$ from each of the classes in Figure 1-(d). The values of some off-diagonal patches are significantly higher than others, which means the examples in those classes look more similar, like 4 and 9, 5 and 3, 8 and 5, etc.

## 3 Learning Dynamics of LLM's Finetuning

Although learning dynamics have been applied to many deep learning systems, extending this framework to LLM finetuning is non-trivial. The first problem is the high dimensionality and the sequence nature of *both* the input and output signals. The high-dimensional property makes it hard to observe the model's output, and the sequential nature makes the distributions on different tokens mutually dependent, which is more complicated than a standard multi-label classification problem considered by most previous work. Furthermore, as there are many different algorithms for LLM finetuning – SFT (Wei et al. 2022), RLHF (Ouyang et al. 2022), DPO (Rafailov et al. 2023), etc. – analyzing them under a uniform framework is challenging. Finally, compared with the training-from-scratch scenario, where a roughly uniform distribution over all possible outputs is usually assumed, LLMs' finetuning dynamics heavily rely on the pretrained base model, which could make the analysis harder. For example, the pretrained model usually assigns little probability mass to unlikely tokens, which is good for most applications but leads to risk of the "squeezing effect" we show later. We now tackle these problems and propose a unified framework for different finetuning methods.

### 3.1 Per-step Decomposition of the SFT Loss

The typical loss function used during supervised finetuning is the negative log-likelihood (NLL) of a given completion $\mathbf{y}_u^+ = [y_1^+, \ldots, y_L^+] \in \mathcal{V}^L$, conditioned on the prompt $\mathbf{x}_u$:

$$\mathcal{L}_{\text{SFT}}(\mathbf{x}_u, \mathbf{y}_u^+) \triangleq -\sum_{l=1}^{L} \log \pi(y = y_l^+ \mid \mathbf{y}_{<l}^+, \mathbf{x}_u) = -\sum_{l=1}^{L} \mathbf{e}_{y_l^+} \cdot \log \pi(\mathbf{y} \mid \mathbf{x}_u, \mathbf{y}_{<l}^+). \tag{4}$$

Note that compared with the multi-label classification problem discussed before, where the joint distribution of all labels can be factorized as $\pi(\mathbf{y} \mid \mathbf{x}) = \prod_l \pi(y_l \mid \mathbf{x})$, the sequential nature of language modeling makes the analysis more complicated, because we must have $\pi(\mathbf{y} \mid \mathbf{x}) = \prod_l \pi(y_l \mid \mathbf{x}, \mathbf{y}_{<l})$. To solve this problem, we can merge this factorization into the definition of the backbone $h_\theta$ while keeping the format of Proposition 1. Specifically, letting $\chi$ be the concatenation of $\mathbf{x}$ and $\mathbf{y}$, the prediction of all tokens of $\mathbf{y}$ is

$$\mathbf{z} = h_\theta(\chi); \quad \pi(\mathbf{y} \mid \chi) = \text{Softmax\_column}(\mathbf{z}).$$

Here $\mathbf{z}$ is a $V \times L$ matrix where each column contains the logits of the prediction of the $l$th token. Our $h_\theta$, even though it takes the entire sequence $\chi$ as its input, will force the model not to refer to the future tokens $\mathbf{y}_{>l}$ when making predictions on the $l$-th token, commonly implemented via "causal masking" (proposed in Vaswani et al. (2017), details in Figure 10a of Appendix D). Then, we can calculate $(\nabla_\theta \mathbf{z}_l(\chi_o)|_{\theta^t})(\nabla_\theta \mathbf{z}_l(\chi_u)|_{\theta^t})^\top$ on each column of $\mathbf{z}$ and stack them to form a $V \times V \times L$ tensor $\mathcal{K}^t(\chi_o, \chi_u)$. The calculation of $\mathcal{G}^t$ and $\mathcal{A}^t$ also follows a similar procedure. Thanks to the causal mask implemented in $h_\theta$, the resulting decomposition is almost identical to that in a multi-label classification problem. Assuming have a response $\mathbf{y}_u$ of length $L$ associated with $\mathbf{x}_u$, stacked into $\chi_u$, and $\mathbf{y}_o$ of length $M$ associated with $\mathbf{x}_o$, stacked into $\chi_o$. The change of the model's prediction on the $m$-th token of $\mathbf{y}_o$ can be represented as, when $\mathbf{z}$ gradients have bounded norm,

$$[\underbrace{\Delta \log \pi^t(\mathbf{y} \mid \chi_o)}_{V \times M}]_m = -\sum_{l=1}^{L} \eta [\underbrace{\mathcal{A}^t(\chi_o)}_{V \times V \times M}]_m [\underbrace{\mathcal{K}^t(\chi_o, \chi_u)}_{V \times V \times L}]_l [\underbrace{\mathcal{G}^t(\chi_u)}_{V \times L}]_l + \mathcal{O}(\eta^2), \tag{5}$$

where $\mathcal{G}_{\text{SFT}}^t(\chi_u) = \pi_{\theta^t}(\mathbf{y} \mid \chi_u) - \mathbf{y}_u$. Compared with Proposition 1, the main difference is that the eNTK term also depends on the responses $\mathbf{y}_u$ and $\mathbf{y}_o$, which allows us to answer questions like

*For a prompt $\mathbf{x}_u$, how does learning the response $\mathbf{y}_u^+$ influence the model's belief about a response $\mathbf{y}_u'$?*

When tracking the model's confidence on different responses given the question $\mathbf{x}_u$, learning from $\mathbf{y}_u^+$ will impose a strong "upwards" pressure on $\mathbf{y}_u^+$, as illustrated in the first panel of Figure 2. At the same time,

the confidence of "similar responses" will also be slightly pulled up, like how learning a `4` influences the prediction on `9` in the MNIST example. We will discuss how to understand the "similarity" between two sequences of responses in the next section.

## 3.2 PER-STEP DECOMPOSITION OF THE DPO LOSS

Preference finetuning, which teaches the model to provide responses that align better with human preferences, is also an important phase in LLM finetuning pipelines. Different from the SFT stage above, where the training tells the model "what to do", many preference finetuning methods also teach the model "what not to do," which makes the learning dynamics more complex. For intuition, we start by analyzing a typical method, i.e., DPO (direct preference optimization, an RL-free method), under a similar framework. Following Rafailov et al. (2023), the loss function of off-policy DPO is

$$\mathcal{L}_{\text{DPO}}(\theta) = -\mathbb{E}_{(\mathbf{x}_u, \mathbf{y}_u^+, \mathbf{y}_u^-) \sim \mathcal{D}} \left[ \log \sigma \left( \beta \log \frac{\pi_{\theta^t}(\mathbf{y}_u^+ \mid \boldsymbol{\chi}_u^+)}{\pi_{\text{ref}}(\mathbf{y}_u^+ \mid \boldsymbol{\chi}_u^+)} - \beta \log \frac{\pi_{\theta^t}(\mathbf{y}_u^- \mid \boldsymbol{\chi}_u^-)}{\pi_{\text{ref}}(\mathbf{y}_u^- \mid \boldsymbol{\chi}_u^-)} \right) \right], \tag{6}$$

where $\mathbf{y}_u^+$ and $\mathbf{y}_u^-$ are pre-generated responses, and $\pi_{\text{ref}}$ is the reference model, typically the result of SFT. In the loss function, the $\pi_{\theta^t}$ terms are also calculated using teacher forcing. Hence we can decompose the learning dynamics for DPO similarly to Equation (5),

$$[\Delta \log \pi^t(\mathbf{y} \mid \boldsymbol{\chi}_o)]_m = -\sum_{l=1}^{L} \eta [\mathcal{A}^t(\boldsymbol{\chi}_o)]_m \left( [\mathcal{K}^t(\boldsymbol{\chi}_o, \boldsymbol{\chi}_u^+)]_l [\mathcal{G}_{\text{DPO+}}^t]_l - [\mathcal{K}^t(\boldsymbol{\chi}_o, \boldsymbol{\chi}_u^-)]_l [\mathcal{G}_{\text{DPO-}}^t]_l \right) + \mathcal{O}(\eta^2)$$

$$\mathcal{G}_{\text{DPO+}}^t = \beta(1-a) \left( \pi_{\theta^t}(\mathbf{y} \mid \boldsymbol{\chi}_u^+) - \mathbf{y}_u^+ \right); \qquad \mathcal{G}_{\text{DPO-}}^t = \beta(1-a) \left( \pi_{\theta^t}(\mathbf{y} \mid \boldsymbol{\chi}_u^-) - \mathbf{y}_u^- \right), \tag{7}$$

where $a$ is the margin (i.e., the $\sigma(\cdot)$ value) for the $l$-th token, which represents how well the current policy separates $\mathbf{y}_u^+$ and $\mathbf{y}_u^-$ compared with the reference policy. Due to the monotonicity of $\sigma(\cdot)$, a larger margin leads to larger $a$, which in turn restrains the strength of $\mathcal{G}_{\text{DPO+/-}}^t$. In other words, $\mathcal{G}_{\text{DPO+/-}}^t$ automatically provides less energy on the examples that are already well separated. We then check the role of $\beta$, which controls the regularizing effect on the KL distance between $\pi_{\theta^t}$ and $\pi_{\text{ref}}$ in the original RL loss (Rafailov et al. 2023). When the margin is negative, larger $\beta$ leads to a smaller $a$ and hence provides stronger $\mathcal{G}_{\text{DPO+/-}}^t$ for the model to "catch up" the separating ability of the reference model faster. But when the model is better and has a positive margin, increasing $\beta$ will increase $a$ and hence create a negative influence on $\beta(1-a)$, which makes the model update less. This aligns well with the claims of Rafailov et al. (2023): the stronger regularizing effect tends to "drag $\pi_\theta$ back towards $\pi_{\text{ref}}$" when its predictions deviate from $\pi_{\text{ref}}$ too much. The derivation and the $\mathcal{G}^t$ functions for other RL-free methods are given in Appendix B.2.2.

These analyses make no assumptions on where $\mathbf{y}_u^+$ and $\mathbf{y}_u^-$ come from. Hence our framework can be directly extended to on-policy RL-free methods, which often perform better than their off-policy counterparts (S. Guo, B. Zhang, et al. 2024; Tajwar et al. 2024). The main difference between these algorithms is how the supervisory responses are generated. Off-policy methods typically use a fixed pre-collected dataset, with $\mathbf{y}_u^+$ and $\mathbf{y}_u^-$ are generated by another LLM or humans. In other words, it is likely that both the chosen and rejected responses come from the "less likely" region of the model's prediction. On-policy responses, though, are more likely to have higher predicted probabilities under this model, as they were sampled from it. We will show next that *imposing large negative pressure on an unlikely prediction will lead to unexpected behavior*, giving a new explanation for why on-policy sampling is important for algorithms with large negative gradients.

## 3.3 THE SQUEEZING EFFECT CAUSED BY NEGATIVE GRADIENT

As demonstrated by the first two panels in Figure 2, the use of large negative gradients is the main difference between the learning dynamics of SFT and DPO. We will show later that this difference is the key to understanding why the learning curves of SFT and DPO behave so differently. For example, Pal et al. (2024),

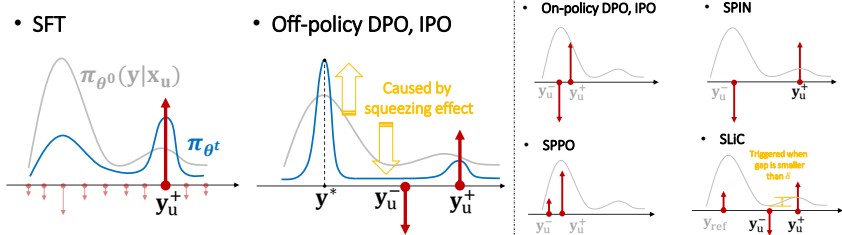

Figure 2: The updating vector provided by the residual term $\mathcal{G}^t$ of different algorithms. The gray **y** are responses *sampled* from $\pi$ in an on-policy way. In the second panel, we demonstrate the "squeezing effect" caused by imposing a big negative gradient on a "valley" region of a distribution. For more details about this counter-intuitive effect, please refer to Section 3.3 and Appendix E. Other panels demonstrate on-policy DPO (and IPO), SPIN (Z. Chen et al. 2024), SPPO (Y. Wu et al. 2024), and SLiC (Y. Zhao et al. 2023).

Rafailov et al. (2024), and Tajwar et al. (2024) (and our Figure 4) reported that the confidence of both $\mathbf{y}_u^+$ and $\mathbf{y}_u^-$ gradually decreases when conducting DPO, while the confidence of $\mathbf{y}_u^+$ rarely drops during SFT. This trend becomes more serious if $\pi_{\theta^0}$ is finetuned for longer before conducting DPO (reported in Figure 3 of Rafailov et al. (2024) and verified by our Figure 17). Furthermore, we also find that for all the $\pi_{\theta^t}(\mathbf{y} \mid \chi_u)$ we track (various responses similar to $\mathbf{y}_u^+$ or $\mathbf{y}_u^-$; details in the next section), none of them increase during the DPO phase. This is different from SFT and quite counter-intuitive:

*If everything we observe is becoming less confident, where has the probability mass gone?*

To answer this question, we first identify a phenomenon we call the *squeezing effect*, which occurs when using negative gradients from any model outputting a distribution with Softmax output heads, even in a simple multi-class logistic regression task. Specifically, in the $L = 1$ case when we impose a negative gradient on label $\mathbf{y}_u^-$, we can describe the changing of model's predictive distribution $\pi_{\theta^{t+1}}$ as follows:

- *Guarantee:* the confidence of $\mathbf{y}_u^-$, i.e., $\pi_{\theta^{t+1}}(\mathbf{y}_u^-)$ will decrease.
- *Guarantee:* the decreased probability mass is largely "squeezed" into the output which was most confident before the update: if $\mathbf{y}^* = \mathrm{argmax}_{i \in [V] \setminus \{\mathbf{y}_u^-\}} \pi_{\theta^t}(\mathbf{y} = i)$, then $\pi_{\theta^{t+1}}(\mathbf{y} = \mathbf{y}^*)$ will increase.
- *Trend:* the rich get richer and the poor get poorer: generally, dimensions with high $\pi_{\theta^t}$ tend to increase, and those with low $\pi_{\theta^t}$ tend to decrease.
- *Trend:* peakier $\pi_{\theta^t}$ squeezes more. If the probability mass concentrates on few dimensions in $\pi_{\theta^t}$, which is common for a pretrained model, all $\pi_{\theta^{t+1}}(\mathbf{y} \neq \mathbf{y}^*)$ decrease (only $\mathbf{y}^*$ is "rich").
- *Trend:* smaller $\pi_{\theta^t}(\mathbf{y}_u^-)$ exacerbate the squeezing effect: if $\mathbf{y}_u^-$ is unlikely under $\pi_{\theta^t}$, the probability mass of all other $\pi_{\theta^{t+1}}(\mathbf{y} \neq \mathbf{y}^*)$ will be more seriously decreased, and the $\pi_{\theta^{t+1}}(\mathbf{y} = \mathbf{y}^*)$ increases more.

Appendix E illustrates the squeezing effect and analytically proves its existence for logistic regression models, by directly computing $\pi_{\theta^{t+1}}/\pi_{\theta^t}$ in different situations. Section 4.2 also experimentally verifies the analysis above in real LLM experiments. Note that in practical settings, where both positive and negative pressures and the auto-regressive nature are strictly considered, the squeezing effect can become more complicated. The differences between the two eNTK terms in Equation (7) also influence the relative strength and direction of these two pressures. Razin et al. (2025) analyze a similar problem in a token-level setting, and their conclusions align with ours well. We left a more detailed analysis to our future work.

We can now get a high-level overview of the learning dynamics of a typical off-policy DPO algorithm. Since both $\mathbf{y}_u^+$ and $\mathbf{y}_u^-$ are not sampled from the model's distribution, $\mathbf{y}^*$ sometimes can be dissimilar to $\mathbf{y}_u^+$, and the $\mathbf{y}_u^-$ are likely located in a valley region of the model's prediction. Then its learning dynamics would look like the sketch in the second panel of Figure 2: the confidence on almost all **y** are pushed down. At the same time,

all the decreased probability mass is squeezed to $\mathbf{y}^*$, which might make the model keep generating repeated phrases, as reported by Holtzman et al. (2020). Variants of DPO algorithms often unintentionally mitigate this squeezing effect by constraining the strength of the negative gradient or the positions of $\mathbf{y}_u^-$, which partially explains their benefits. The last four panels in Figure 2 and Appendix B.2.2 have more details.

# 4 EXPERIMENTAL VERIFICATIONS

We now verify our analysis in practical settings. We first create the training set $\mathcal{D}_{\text{train}}$ by randomly selecting 5000 examples from the training split of the dataset. We consider two common datasets, `Antropic-HH` (Y. Bai et al. 2022) and `UltraFeedback` (G. Cui et al. 2023), in all experiments. Each example in $\mathcal{D}_{\text{train}}$ contains three components: the prompt (or question) $\mathbf{x}$, the preferred response $\mathbf{y}^+$, and the less preferred response $\mathbf{y}^-$. SFT finetunes with $\mathbf{x}$ and $\mathbf{y}^+$, while DPO uses all three (subscripts of $\mathbf{x}$ and $\mathbf{y}$ are removed for conciseness). We repeat the experiments on two series of models: `pythia-410M/1B/1.4B/2.8B` (Biderman et al. 2023) and `Qwen1.5-0.5B/1.8B` (J. Bai et al. 2023).

To get a more detailed observation of the learning dynamics, we further create a probing dataset $\mathcal{D}_{\text{prob}}$ by randomly selecting 500 examples from $\mathcal{D}_{\text{train}}$, and generate several typical responses based on the corresponding $\mathbf{x}$, $\mathbf{y}^+$, or $\mathbf{y}^-$. (We also study another probing dataset where all $\mathbf{x}$ come from the test set in an ablation study in the appendix.) Then for each $\mathbf{x}$ in $\mathcal{D}_{\text{prob}}$, we can observe how $\log \pi_{\theta^t}(\mathbf{y} \mid \chi)$ gradually changes on different types of $\mathbf{y}$. For example, one extended response type can be a rephrase of $\mathbf{y}^+$, an irrelevant response answering another question $\mathbf{x}'$, or just a randomly generated English sentence with the same number of words with $\mathbf{y}^+$. We explain why we need these extended responses and how they are generated in detail in Appendix D.1. In short, $\mathcal{D}_{\text{prob}}$ helps us to get a more fine-grind inspection of the learning dynamics, which can not only support our analysis above, but also shed more light on how the model's prediction evolves on the entire $\mathcal{Y} \in \mathbb{R}^{V \times L}$, a very sparse and huge space.

## 4.1 LEARNING DYNAMICS OF SFT

The main lesson we learn from the analysis in Section 3.1 is that learning from $\mathbf{y}_u^+$ not only increases the model's confidence on $\mathbf{y}_u^+$, but also indirectly "pulls up" responses similar to $\mathbf{y}_u^+$ with a smaller strength (scaled roughly by $\|\mathcal{K}^t\|_F$), similar to how learning a "4" influences the prediction of a "9" in the MNIST example. At the same time, the increase of $\pi_{\theta^t}(\mathbf{y}_u^+|\chi_u)$ naturally "pushes down" all $\mathbf{y} \neq \mathbf{y}_u^+$, because the model's predicted probability to all responses in $\mathcal{Y}$-space must sum to one. The model's behavior on different $\mathbf{y}$ is mostly a trade-off among these pressures. To verify this claim, we finetune the model for several epochs and evaluate the model's prediction on all responses in $\mathcal{D}_{\text{prob}}$ every 25 updates (with a training batch size of 4, the probing occurs every 100 examples). For each type of response, we average the model's confidence on all 500 examples and report the mean value of their log-likelihood.

As demonstrated in the first panel of Figure 3, the model's confidence on $\mathbf{y}_u^+$ keeps increasing throughout the whole learning process, which is straightforward because the main "pull-up" pressure is imposed directly on $\mathbf{y}_u^+$. However, the behavior of some responses similar to $\mathbf{y}_u^+$ is non-trivial. For example, we draw the following types of responses in the same panel, i.e., the less preferred response for the same question ($\mathbf{y}_u^-$), two types of rephrases of $\mathbf{y}_u^+$ generated by `ChatGPT` ($\mathbf{y}_{\text{gpts}}^+$ and $\mathbf{y}_{\text{gptf}}^+$), another preferred response randomly selected from the test set ($\mathbf{y}_{\text{test}}^+$), or even a randomly generated English sentence ($\mathbf{y}_{\text{hum}}$). The model's confidence in these responses are all slightly increased at the beginning of training, and then gradually decrease as the training goes on, even though the model never sees them during SFT. This counter-intuitive behavior can be well explained by the learning dynamics we discussed before. Since all these examples are "similar" to $\mathbf{y}_u^+$ to some extent (at least, they are all common "standard English" sentences), their $\|\mathcal{K}^t\|_F$ are reasonably large. Then learning $\mathbf{y}_u^+$ will indirectly increase the model's confidence of these similar $\mathbf{y}$. That is why the corresponding $\pi_{\theta^t}(\mathbf{y}|\chi_u)$ are slightly increased at the beginning of training. However, as the training goes

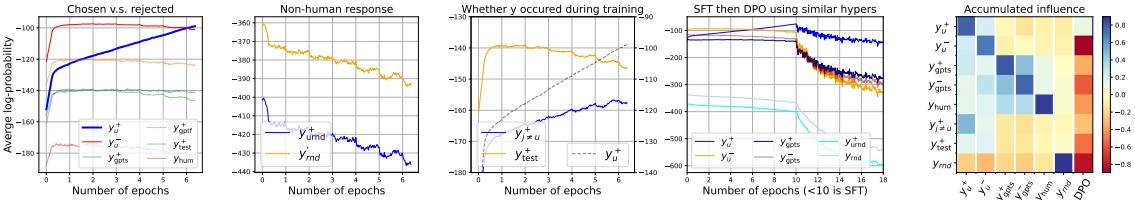

Figure 3: First three: learning dynamics of SFT on different response types. Fourth: SFT 10 epochs then DPO. Last: the accumulated influence when SFT using different $\mathbf{y}$ (full results in Appendix C and D).

on, the model's confidence on $\mathbf{y}_u^+$ keeps increasing and the update energy, the norm of $\mathcal{G}_{\text{SFT}}^t$ in Equation (5), gradually decreases. That means the indirect "pull-up" pressures are also diminished accordingly. Then, the "push-down" pressure on all $\mathbf{y} \neq \mathbf{y}_u^+$ becomes dominant and all the related curves start going down.

To verify the existence of this global "push-down" pressure, we observe two types of responses; both have the same number of words as their $\mathbf{y}_u^+$. One is a purely random English word sequence $\mathbf{y}_{\text{rnd}}'$. Another is a random permutation of all the words in $\mathbf{y}_u^+$, which is called $\mathbf{y}_{\text{urnd}}^+$. Since both are not natural language, we expect the $\|\mathcal{K}^t\|_F$ between them and $\mathbf{y}_u^+$ to be very small, which means learning from $\mathbf{y}_u^+$ imposes almost no "pull-up" pressure on them; thus the "push-down" pressure will dominate through the whole training procedure. These analyses are well supported by the second panel in Figure 3, in which we see these $\pi_{\theta^t}(\mathbf{y}|\mathbf{\chi}_u)$ all start from a very small value, and keep decreasing throughout the training.

Another interesting type of responses is $\mathbf{y}_{j\neq u}^+$, a preferred response for another question $\mathbf{x}_{j\neq u}$ in the training set. For these responses, the model's prediction on $\pi_{\theta^t}(\mathbf{y}_{j\neq u}^+|\mathbf{\chi}_u)$ will be kept influenced by two "pull-up" pressures: one is from learning $[\mathbf{x}_u; \mathbf{y}_u^+]$, another is from learning $[\mathbf{x}_{j\neq u}; \mathbf{y}_{j\neq u}^+]$, where the latter might be even stronger as the gradient is directly calculated by observing $\mathbf{y}_{j\neq u}^+$. That explains why we see the confidence on $\mathbf{y}_{j\neq u}^+$ keeps increasing with a smaller rate compared with $\mathbf{y}_u^+$ in the third panel. Because the "pull-up" pressure is always strong enough to counter the "push-down" one. These observations provide us with a unique explanation of why specific types of hallucinations are amplified after SFT. Specifically, the increase of $\pi_{\theta^t}(\mathbf{y}_{j\neq u}^+|\mathbf{\chi}_u)$ means if we ask the model to answer a question $\mathbf{x}_u$, it might provide a response from (or partially from) another unrelated question $\mathbf{x}_{j\neq u}$ in the training set.

Last, to further explore the "similarity" between different responses from the model's perspective. we SFT the model using more types of responses and observe how $\pi_{\theta}(\mathbf{y}' \mid \mathbf{\chi}_u)$ changes accordingly. The results are demonstrated in Figure 3, where the blue and orange colors represent the positive and negative influence respectively. The x-axis is the updating response while the y-axis denotes the observing response. Hence the first column resembles how different $[\mathbf{x}_u; \mathbf{y}']$ changes when we SFT the model using $[\mathbf{x}_u; \mathbf{y}_u^+]$. One interesting finding is that all responses generated by ChatGPT are considered very similar to each other, regardless of how semantically different they are. Probably, LLM has its preferred idioms or phrases, which could be considered as a type of "fingerprint". We left this interesting problem for our future work.

## 4.2 LEARNING DYNAMICS OF OFF-POLICY DPO

To verify our framework also explains the model's behavior in preference tuning, we conduct similar experiments for DPO. Recall the residual term $\mathcal{G}_{\text{DPO}}^t$ introduces a pair of arrows on both $\mathbf{y}_u^+$ and $\mathbf{y}_u^-$, with different directions. To show how these two pressures influence the model, we check two types of rephrases of $\mathbf{y}_u^+$ or $\mathbf{y}_u^-$ ($\mathbf{y}_{\text{gpts}}^+$, $\mathbf{y}_{\text{gptf}}^+$, $\mathbf{y}_{\text{gpts}}^-$, and $\mathbf{y}_{\text{gptf}}^-$, used in the previous experiment). See the three curves in the first panel in Figure 4, where the two rephrases decrease at a similar speed, faster than the decay of $\mathbf{y}_u^+$. That is because the upward pressure is directly imposed on $\mathbf{y}_u^+$ rather than these rephrases. Similarly, in the second

Figure 4: Learning dynamics of off-policy DPO. The last panel verifies the existence of the squeezing effect.

panel, we observe that $\mathbf{y}_u^-$ decays faster than its rephrases, because $\mathcal{G}_{\text{DPO}}^t$ directly imposes a negative pressure on $\mathbf{y}_u^-$. Then in the third panel, we find the rephrases of $\mathbf{y}_u^+$ consistently decay slower than those of $\mathbf{y}_u^-$, although none of them ever occur during training. That is because these responses are close to $\mathbf{y}_u^+$ or $\mathbf{y}_u^-$ in $\mathcal{Y}$, which means their $\|\mathcal{K}^t\|_F$ is relatively large. Hence the pressures imposed on $\mathbf{y}_u^+$ and $\mathbf{y}_u^-$ also introduce a non-negligible influence on them. Last, in the fourth panel, the margin $\pi_{\theta^t}(\mathbf{y}_u^+|\boldsymbol{\chi}_u) - \pi_{\theta^t}(\mathbf{y}_u^-|\boldsymbol{\chi}_u)$ keeps increasing, which means the model is gaining the ability to separate $\mathbf{y}_u^+$ and $\mathbf{y}_u^-$ as the training goes on.

Although $\mathcal{G}_{\text{DPO}}^t$ directly imposes a "pull-up" pressure on $\mathbf{y}_u^+$, the value of $\pi_{\theta^t}(\mathbf{y}_u^+|\boldsymbol{\chi}_u)$ does not increase a lot as it does in SFT. The downward arrow on $\mathbf{y}_u^-$ indeed introduces a "push-down" pressure on responses that are similar to $\mathbf{y}_u^-$, but the influence is unlikely to be that strong (it will be weakened by $\|\mathcal{K}^t\|_F$) to make the confidence on almost every observing responses decrease so fast, as demonstrated in the last panel of Figure 3 where we use similar $\eta$ for both SFT and DPO. Then, *where has the probability mass gone during DPO?* The key to answering this question is the *squeezing effect* discussed in Section 3.3: since the big negative gradient is imposed on $\mathbf{y}_u^-$, which is at this point probably in a region of low $\pi_{\theta^t}(\mathbf{y}|\boldsymbol{\chi}_u)$, the confidence of most $\mathbf{y}$ will be decreased and $\pi_{\theta^t}(\mathbf{y}^*|\boldsymbol{\chi}_u)$ will increase very fast.

To verify this, we report the log-likelihood of $\mathbf{y}$ chosen by greedy decoding: each token is chosen by maximizing the conditional probability given $[\mathbf{x}_u; \mathbf{y}_{<l}^+]$ in real-time, where $\mathbf{y}_{<l}^+$ is a sub-sequence of $\mathbf{y}_u^+$. As illustrated by the last panel of Figure 4, the confidence of this "teacher forcing" greedy $\mathbf{y}$ increases very fast (from -113 to -63), which is even faster than the increase of $\pi_{\theta^t}(\mathbf{y}_u^+|\boldsymbol{\chi}_u)$ during SFT (from -130 to -90), within 8 epochs. However, the tokens with the highest confidence do not necessarily form a preferred response: it will reinforce the prior bias in $\theta^0$. This could be a reasonable explanation of the "degeneration" reported in recent work (e.g. Holtzman et al. 2020): as $\pi_{\theta^t}$ becomes more peaky at its most confident predictions, it is easier to sample sequences with repeated phrases. Note that such behavior could also be understood as a special type of self-bias amplifying (Ren et al. 2024), which would bring more serious consequences if it is combined with a multiple-generation self-improving algorithm, e.g., self-reward (Yuan et al. 2024), iterative DPO (Xiong et al. 2024), etc.

In summary, the behaviors of different types of responses all match our analyses well. More subtle trends of different responses support our story well (both for SFT and DPO).Due to space constraints, we explain these (and the full results on other models and datasets) in Appendix D.

### 4.3 MITIGATING THE SQUEEZING EFFECT BY AUGMENTING THE TRAINING SET FOR SFT

Since the "squeezing effect" caused by the big negative gradient on unlikely predictions can damage the model's performance during DPO, we can first train the model on *both* $[\mathbf{x}_u; \mathbf{y}_u^+]$ and $[\mathbf{x}_u; \mathbf{y}_u^-]$ during the SFT stage (making the negative response *more* likely), and then run the usual DPO. Following the analysis above, we can expect during this new SFT stage, the region of those responses similar to $\mathbf{y}_u^+$ *or* $\mathbf{y}_u^-$ will be "pulled up" simultaneously. This is what we want because in many cases, both $\mathbf{y}_u^+$ and $\mathbf{y}_u^-$ are reasonably good responses for the question $\mathbf{x}_u$; the new SFT design hence helps to pull up a larger region that contains more suitable responses compared with the baseline SFT. After that, the "push-down" pressure imposed during DPO can

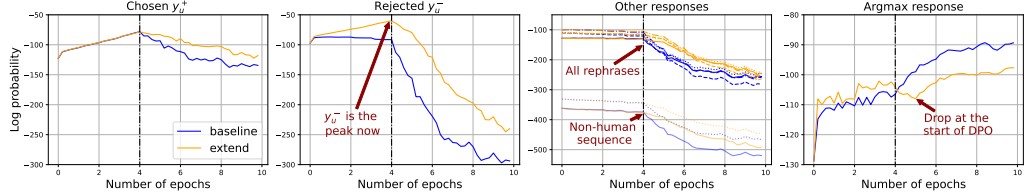

Figure 5: Learning dynamics of the baseline and the proposed method with training data extension. Key trends to observe: 1.) Baseline and the extend method have similar behavior on $\mathbf{y}_u^+$ during SFT; 2.) The extend method considerably increase $\mathbf{y}_u^-$ during SFT; 3.) The squeezing effect of the extend method is weaker (all other responses decay slower and the confidence on the "greedy-decoding" response increases slower).

efficiently decrease the model's confidence on $\mathbf{y}_u^-$ and its similar responses. Since $\mathbf{y}_u^-$ is no longer so unlikely before DPO, the squeezing effect should not be as strong as in the baseline procedure.

We call our training pipeline "extend" and compare its learning dynamics with the baseline setting in Figure 5. It is clear that the squeezing effect is mitigated, because the confidence of other responses all decays slower during DPO, and we also observe a big drop in the greedy-decoding response when DPO starts. To further show that mitigating the squeezing effect indeed brings benefits, we compare the responses generated by models trained using different methods by feeding them to ChatGPT and Claude3. Specifically, we first SFT the model for two epochs using two methods discussed above and call the resulting policy network $\pi_{\text{base}}$ and $\pi_{\text{extend}}$. Then, we conduct identical DPO

Table 1: Win-rate against baseline.

| DPO Ep. | ChatGPT | Claude |
|---------|---------|--------|
| 0 | 0.4729 | 0.4679 |
| 2 | 0.6518 | 0.5151 |
| 4 | 0.6928 | 0.6045 |
| 6 | 0.6667 | 0.5432 |

training on both $\pi_{\text{base}}$ and $\pi_{\text{extend}}$ for several epochs. The win rate of the proposed method against the baseline one is provided in Table 1. It is clear that before DPO, $\pi_{\text{base}}$ is better, because $\pi_{\text{extend}}$ is explicitly trained on those $\mathbf{y}^-$. However, the $\pi_{\text{extend}}$ performs better after DPO several epochs since the squeezing effect is efficiently mitigated. Please refer to Appendix F for more details. In the future, this simple method inspired by our analysis could be further improved by introducing more responses, e.g., rephrases of $\mathbf{y}_u^+$, etc., during both stages, and also by combining with many existing RL-free methods we mentioned before.

## 5 CONCLUSION

Learning dynamics, which depict how the model's prediction changes when it learns new examples, provide a powerful tool to analyze the behavior of models trained with gradient descent. To better utilize this tool in the context of LLM finetuning, we first derive the step-wise decomposition of LLM finetuning for various common algorithms. Then, we propose a unified framework for understanding LLM predictions' behaviors across different finetuning methods. The proposed analysis successfully explains various phenomena during LLM's instruction tuning and preference tuning, some of them are quite counter-intuitive. We also shed light on how specific hallucinations are introduced in the SFT stage, as previously observed (Gekhman et al. 2024), and where the improvements of some new RL-free algorithms come from compared with the vanilla off-policy DPO. The analysis of the squeezing effect also has the potential to be applied to other deep learning systems which apply big negative gradients to already-unlikely outcomes. Finally, inspired by this analysis, we propose a simple (but counter-intuitive) method that is effective in improving the alignment of models.

## ACKNOWLEDGEMENTS

This research was enabled in part by support provided by the Canada CIFAR AI Chairs program, WestGrid, and Compute Canada. We thank Shangmin Guo, Noam Razin, Wonho Bae, and Hamed Shirzad for their valuable discussions and feedback. We also appreciate the constructive comments from the anonymous reviewers, which helped improve this work.

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

## A    MORE RELATED WORKS

### A.1    MORE ABOUT LEARNING DYNAMICS

Instead of emphasizing the model's convergence status and guarantees, learning dynamics focus more on its relative behavior during training. Since many LLMs exhibit "emergent ability" during training in terms of the amount of training data or inference time (Schaeffer et al. 2024), like the "aha moment" mentioned in D. Guo et al. (2025), we hence believe the analysis of learning dynamics can provide a novel perspective on understanding the LLM system.

Beyond their application to LLMs, learning dynamics are widely utilized in analyzing various machine learning problems. For example, if we consider $\mathbf{x}_u$ from the training set, and $\mathbf{x}_o$ from the test set, this form of learning dynamics provides a new perspective on generalization: the model generalizes better if the loss of $f_\theta(\mathbf{x}_o)$ keeps decreasing when it learns from $\mathbf{x}_u$. By studying the influence of different $\mathbf{x}_u$ at different stages during supervised learning, Ren et al. (2022) explain a "zigzag" pattern of the learning path, which sheds light on why the model can spontaneously pursue better supervisory signals and correct noisy labels in the early stage of training (see also S. Liu et al. 2020). Kumar et al. (2022) and Ren, S. Guo, et al. (2023) apply learning dynamics to explain why directly finetuning a well-trained backbone with a randomly initialized task head might harm the out-of-distribution generalization ability. Ren et al. (2020), Ren, Lavoie, et al. (2023), and Ren and Sutherland (2024) also explains where the simplicity bias favoring compositional representations comes from during knowledge distillation (Hinton et al. 2015), providing a new perspective of understanding why successive knowledge transferring can improve the model's systematic generalization ability.

Besides explaining the model's behavior, learning dynamics is also helpful for evaluating the quality or the effectiveness of different training samples. For example, Pruthi et al. (2020) propose a quantitative metric called TracIn to compute the influence of a training example on the predictions made by the model. This metric is then applied by Xia et al. (2024) to search for the most influential examples in LLM instruction finetuning. By expanding Equation (1) in the neural tangent kernel (NTK) regime, S. Guo, Ren, et al. (2024) propose a metric called lpNTK to measure the relative difficulty among different training samples. These metrics and analyses inspired by learning dynamics are expected to be helpful in many related fields, like coreset selection (Feldman 2020), active learning (Settles 2009) (see, e.g., Mohamadi et al. 2022), and dataset distillation (T. Wang et al. 2018).

### A.2    MORE ABOUT LLM'S FINETUNING

In this paper, we broadly define finetuning as any in-weight learning on top of a pretrained base model, including supervised finetuning (SFT), direct policy optimization (DPO, Rafailov et al. 2023) and its variants, etc. Since the analysis throughout this paper relies on the "teacher forcing" mechanism and the relatively stable eNTK assumption, our framework cannot be directly applied to algorithms with token-wise supervision like reinforcement learning with human feedback (RLHF, Ouyang et al. 2022) and proximal policy optimization (PPO, Schulman et al. 2017)[1]. We leave the study of the token-wise learning dynamics, which aligns better with the "squeezing effect" in real settings, to future work.

We also identify several related works that report similar observations on the phenomena discussed in this paper. For example, Gekhman et al. (2024) and Yue Zhang et al. (2023) mentioned that learning new facts during SFT tends to make the model hallucinate more, which aligns with our finding that the model tends to use $\mathbf{y}_{j\neq i}^\top$ when answering question $i$. Holtzman et al. (2020) related the peakiness of the model's distribution to LLM's "repeater phenomena", which also indirectly supports our claims well: more DPO leads to a more serious squeezing effect, hence the model's prediction becomes peakier on most tokens, which makes the aforementioned phenomena more common.

---

[1]If the implementation indeed applies the teacher forcing mechanism, our analysis still works.

Furthermore, the "confidence decaying on $\mathbf{y}_u^+$" attracts more attention in the community, because it is quite counter-intuitive and the vanilla off-policy DPO algorithm works reasonably well in most cases. Many related works study this phenomenon by analyzing the major discrepancy between off-policy DPO and PPO, i.e., where the samples used to train the model comes from, e.g., S. Guo, B. Zhang, et al. (2024), Rafailov et al. (2024), and Tang et al. (2024). They showed that when the responses are off-policy sampled, the learning process may fail to benefit from the contrastive information in the data. In other words, we should be more careful when working on the "valley" region of the model's distribution. Other works try to analyze this problem by inspecting the token-level influence between responses. For example, Pal et al. (2024) assumes $\mathbf{y}_u^+$ and $\mathbf{y}_u^-$ are identical expect one token. Under this assumption, the model's confidence of $\mathbf{y}_u^+$ after the identical token is guaranteed to decrease. They propose a solution by significantly enhancing the learning rate (roughly x50 larger when their $\lambda = 50$) of the positive part when detecting $\mathbf{y}_u$ located in a low-confidence region. Razin et al. (2025) takes the similarity between the hidden embeddings and the geometry of the readout layer of different responses into account. Most of the conclusions of their paper align with ours well. The main discrepancy lies in the squeezing effect part, which we will discuss in our future work (they do not contradict each other, but need a more detailed analysis to understand the whole story).

### A.3 BENIGN AND HARMFUL NEGATIVE GRADIENT

The "squeezing effect" can negatively impact our analysis when it is strongly imposed in a valley region of the model. However, a well-regulated negative gradient is both beneficial and commonly observed in many deep-learning systems. For example, it is common in many "machine unlearning" algorithms, e.g., in Ruiqi Zhang et al. (2024). Moreover, even in the field of LLM finetuning, we can find many mechanisms in different popular algorithms that can mitigate this effect. For example, the typical learning rate of DPO is usually smaller than that used in SFT, which unintentionally mitigates the harmful squeezing effect. The on-policy counterpart of the DPO-like algorithms is shown to perform better than their off-policy counterparts, which also supports our claims. Furthermore, we find the PPO loss automatically avoids imposing a big negative gradient (when its $\hat{A}_t$ is negative) on the valley region (when its $\pi_\theta$ is small).

On the other hand, the effect that negative gradients make the model's distribution peakier is independently reported in many related works. For example, Equation 1 in Caccia et al. (2020) shows that we are minimizing a negative thing in a standard GAN loss, which might explain why peakiness occurs. Furthermore, in Table 1 and Table 2 of Choshen et al. (2020), we see the peakiness (measured by $\Delta p_{top10}, \Delta p_{mode}$) of the "PG-average" method is stronger than the standard PG method. Note that the "PG-average" method will map a reward ranging from 0 to 1 to a centered one ranging from -0.5 to 0.5. Since the negative reward can introduce a negative gradient, the peakiness increases.

## B PROOF OF PROPOSITIONS AND RESIDUAL TERM FOR DIFFERENT LOSSES

### B.1 PROOF OF PROPOSITION 1

**Proposition 1.** *Let $\pi = \mathsf{Softmax}(\mathbf{z})$ and $\mathbf{z} = h_\theta(\mathbf{x})$. The one-step learning dynamics decompose as*

$$\underbrace{\Delta \log \pi^t(\mathbf{y} \mid \mathbf{x}_o)}_{V \times 1} = -\eta \underbrace{\mathcal{A}^t(\mathbf{x}_o)}_{V \times V} \underbrace{\mathcal{K}^t(\mathbf{x}_o, \mathbf{x}_u)}_{V \times V} \underbrace{\mathcal{G}^t(\mathbf{x}_u, \mathbf{y}_u)}_{V \times 1} + \mathcal{O}(\eta^2 \|\nabla_\theta \mathbf{z}(\mathbf{x}_u)\|_{\mathrm{op}}^2), \tag{3}$$

*where $\mathcal{A}^t(\mathbf{x}_o) = \nabla_{\mathbf{z}} \log \pi_{\theta^t}(\mathbf{x}_o) = I - \mathbf{1}\pi_{\theta^t}^\top(\mathbf{x}_o)$, $\mathcal{K}^t(\mathbf{x}_o, \mathbf{x}_u) = (\nabla_\theta \mathbf{z}(\mathbf{x}_o)|_{\theta^t})(\nabla_\theta \mathbf{z}(\mathbf{x}_u)|_{\theta^t})^\top$ is the empirical neural tangent kernel of the logit network $\mathbf{z}$, and $\mathcal{G}^t(\mathbf{x}_u, \mathbf{y}_u) = \nabla_{\mathbf{z}} \mathcal{L}(\mathbf{x}_u, \mathbf{y}_u)|_{\mathbf{z}^t}$.*

*Proof.* [2] Suppose we want to observe the model's prediction on an "observing example" $\mathbf{x}_o$. Starting from Equation (2), we first approximate $\log \pi^{t+1}(\mathbf{y} \mid \mathbf{x}_o)$ using first-order Taylor expansion (we use $\pi^t$ to represent $\pi_{\theta^t}$ interchangeably for notation conciseness):

$$\log \pi^{t+1}(\mathbf{y} \mid \mathbf{x}_o) = \log \pi^t(\mathbf{y} \mid \mathbf{x}_o) + \langle \nabla \log \pi^t(\mathbf{y} \mid \mathbf{x}_o), \theta^{t+1} - \theta^t \rangle + O(\|\theta^{t+1} - \theta^t\|^2).$$

Then, assuming the model updates its parameters using SGD calculated by an "updating example" $(\mathbf{x}_u, \mathbf{y}_u)$, we can rearrange the terms in the above equation to get the following expression:

$$\Delta \log \pi^t(\mathbf{y} \mid \mathbf{x}_o) = \underbrace{\log \pi^{t+1}(\mathbf{y} \mid \mathbf{x}_o)}_{V \times 1} - \underbrace{\log \pi^t(\mathbf{y} \mid \mathbf{x}_o)}_{V \times 1} = \underbrace{\nabla_\theta \log \pi^t(\mathbf{y} \mid \mathbf{x}_o)|_{\theta^t}}_{V \times d} \underbrace{(\theta^{t+1} - \theta^t)}_{d \times 1} + O(\|\theta^{t+1} - \theta^t\|^2),$$

where $d$ is the number of parameters of the model. To evaluate the leading term, we plug in the definition of SGD and repeatedly use the chain rule:

$$\underbrace{\nabla_\theta \log \pi^t(\mathbf{y} \mid \mathbf{x}_o)|_{\theta^t}}_{V \times d} \underbrace{(\theta^{t+1} - \theta^t)}_{d \times 1} = \big( \underbrace{\nabla_\mathbf{z} \log \pi^t(\mathbf{x}_o)|_{\mathbf{z}^t}}_{V \times V} \underbrace{\nabla_\theta \mathbf{z}^t(\mathbf{x}_o)|_{\theta^t}}_{V \times d} \big) \big( -\eta \underbrace{\nabla_\theta \mathcal{L}(\mathbf{x}_u)|_{\theta^t}}_{1 \times d} \big)^\mathsf{T}$$

$$= \underbrace{\nabla_\mathbf{z} \log \pi^t(\mathbf{x}_o)|_{\mathbf{z}^t}}_{V \times V} \underbrace{\nabla_\theta \mathbf{z}^t(\mathbf{x}_o)|_{\theta^t}}_{V \times d} \big( -\eta \underbrace{\nabla_\mathbf{z} \mathcal{L}(\mathbf{x}_u)|_{\mathbf{z}^t}}_{1 \times V} \underbrace{\nabla_\theta \mathbf{z}^t(\mathbf{x}_u)|_{\theta^t}}_{V \times d} \big)^\mathsf{T}$$

$$= -\eta \underbrace{\nabla_\mathbf{z} \log \pi^t(\mathbf{x}_o)|_{\mathbf{z}^t}}_{V \times V} \big[ \underbrace{\nabla_\theta \mathbf{z}^t(\mathbf{x}_o)|_{\theta^t}}_{V \times d} \underbrace{(\nabla_\theta \mathbf{z}^t(\mathbf{x}_u)|_{\theta^t})^\mathsf{T}}_{d \times V} \big] \underbrace{(\nabla_\mathbf{z} \mathcal{L}(\mathbf{x}_u)|_{\mathbf{z}^t})^\mathsf{T}}_{V \times 1}$$

$$= -\eta \mathcal{A}^t(\mathbf{x}_o) \mathcal{K}^t(\mathbf{x}_o, \mathbf{x}_u) \mathcal{G}^t(\mathbf{x}_u, \mathbf{y}_u) \tag{8}$$

For the higher-order term, using as above that

$$\theta^{t+1} - \theta^t = -\eta \nabla_\theta \mathbf{z}^t(\mathbf{x}_u)|_{\theta^t}^\mathsf{T} \mathcal{G}^t(\mathbf{x}_u, \hat{\mathbf{y}})$$

and noting that, since the residual term $\mathcal{G}^t$ is usually bounded (and the practical algorithms will also use gradient clip to avoid too large gradient), we have that

$$O\big(\|\theta^{t+1} - \theta^t\|^2\big) = O\big(\eta^2 \|(\nabla_\theta \mathbf{z}^t(\mathbf{x}_u)|_{\theta^t})^\mathsf{T}\|_{\mathrm{op}}^2 \|\mathcal{G}^t(\mathbf{x}_u, \hat{\mathbf{y}})\|_{\mathrm{op}}^2\big) = O\big(\eta^2 \|\nabla_\theta \mathbf{z}(\mathbf{x}_u)\|_{\mathrm{op}}^2\big). \qquad \square$$

In the decomposition, using $\{\pi_1, \ldots, \pi_V\}$ to represent the model's prediction on different dimensions, we can write our $\mathcal{A}^t$ as:

$$\mathcal{A}^t(\mathbf{x}_o) = I - \mathbf{1}(\pi^t)^\top = \begin{bmatrix} 1 - \pi_1 & -\pi_1 & \cdots & -\pi_1 \\ -\pi_2 & 1 - \pi_2 & \cdots & -\pi_2 \\ \cdots & \cdots & \ddots & \cdots \\ -\pi_V & -\pi_V & \cdots & 1 - \pi_V \end{bmatrix}, \tag{9}$$

The second term in this decomposition, $\mathcal{K}^t(\mathbf{x}_o, \mathbf{x}_u)$, is the product of gradients at $\mathbf{x}_o$ and $\mathbf{x}_u$. Intuitively, if their gradients have similar directions, the Frobenius norm of this matrix is large, and vice versa. This matrix is known as the empirical neural tangent kernel, and it can change through the course of training as the network's notion of "similarity" evolves. For appropriately initialized very wide networks trained with very small learning rates, $\mathcal{K}^t$ remains almost constant during the course of training, the kernel it converges to is known as the neural tangent kernel (Arora et al. 2019; Jacot et al. 2018). Note that the assumption that $\mathcal{K}^t(\mathbf{x}_o, \mathbf{x}_u)$ is unchanged (usually used in theoretical analysis) might be too strong in the LLM's finetuning. Hence as stated in the main context, our qualitative analysis only assumes that "during the training, the relative influence of learning $\mathbf{x}_u$ on all other different $\mathbf{x}_o$ is relatively stable". We will validate this assumption using experiments in Appendix C.

---

[2] Note that this proposition assumes $L = 1$. For $L > 1$ case, we will have multiple task heads which leads to $L$ different Equation (3). The $V \times L$ matrix $\Delta \log \pi^t$ can then be achieved by stacking them.

## B.2 RESIDUAL TERM FOR DIFFERENT LLM FINETUNING ALGORITHMS

As stated in Section 3, one of the conundrums of decomposing the learning dynamics of LLM is its auto-regression nature of the output sequence. Different from the multi-label classification problem, where $y_l$ for different $l$ is independently generated as long as the shared network is fixed, the $y_l$ for the LLM's output depends on $\mathbf{y}_{<l}$, which is usually sampled from the model's prediction iteratively. However, in most of the finetuning cases where the supervisory signal $\mathbf{y}_u$ is given, the model will apply the so-called "teacher forcing" mechanism when calculating the predicting probabilities. In other words, when generating the output of each $y_l$, the $\mathbf{y}_{<l}$ is given rather than sampled on-policy. This mechanism makes it possible for us to define $\chi = [\mathbf{x}; \mathbf{y}]$ and hence merge the auto-regressive nature of the sequence prediction into the shared $\mathcal{K}^t(\chi_o, \chi_u)$. After this step, the decomposition of LLM's finetuning learning dynamics then becomes similar to a multi-label classification task.

### B.2.1 INSTRUCTION FINETUNING USING AUTO-REGRESSION LOSS (SFT)

Here we derive the residual term, i.e., $\mathcal{G}^t$ for different algorithms in LLM's finetuning. We first rewrite Equation (5) here:

$$[\underbrace{\Delta \log \pi^t(\mathbf{y} \mid \chi_o)}_{V \times M}]_m = -\sum_{l=1}^{L} \eta [\underbrace{\mathcal{A}^t(\chi_o)}_{V \times V \times M}]_m [\underbrace{\mathcal{K}^t(\chi_o, \chi_u)}_{V \times V \times L}]_l [\underbrace{\mathcal{G}^t(\chi_u)}_{V \times L}]_l + O(\eta^2),$$

where $m \in \{1, \ldots, M\}$, $l \in \{1, \ldots, L\}$, and $\mathcal{G}^t(\chi_u) = \nabla_{\mathbf{z}} \mathcal{L}(\chi_u)|_{\mathbf{z}^t}$ is a $V \times L$ matrix. As the auto-regression nature of the SFT loss is already encoded in the causal mask used in $h_\theta$, as demonstrated in Figure 10a. the columns in $\mathcal{G}^t(\chi_u)$ are independent of each other, which can be separately calculated. Plus, the summation over $l$ can also be achieved by left-multiplying a length-$L$ all-one vector $\mathbf{1}$. Specifically, the SFT loss for each $l$ is:

$$[\mathcal{L}_{\text{SFT}}(\chi_u)]_l = -\log \pi(y_l = y_u^+ \mid \chi_u) = -\mathbf{e}_{y_u^+}^\top \log \pi(y_l \mid \chi_u) = -\mathbf{e}_{y_u^+}^\top \log \left(\text{Softmax}(\mathbf{z}_l)\right),$$

where $y_u^+$ is for the $l$-th dimension of $\mathbf{y}_u^+$. The gradient of $\mathcal{L}$ on $\mathbf{z}$ can be then calculated as:

$$
\begin{aligned}
[\mathcal{G}_{\text{SFT}}^t(\chi_u)]_l &= \underbrace{\nabla_{\mathbf{z}_l}[\mathcal{L}_{\text{SFT}}(\chi_u)]_l}_{1 \times V} = \left(\underbrace{\nabla_\pi [\mathcal{L}_{\text{SFT}}(\chi_u)]_l}_{V \times 1}\right)^\top \underbrace{\nabla_{\mathbf{z}_l} \pi}_{V \times V} \\
&= -\left(\mathbf{e}_{y_u^+} \oslash \pi\right)^\top \nabla_{\mathbf{z}_l} \pi = \pi(y_l \mid \chi_u) - \mathbf{e}_{y_u^+},
\end{aligned}
\tag{10}
$$

where $\oslash$ is element-wise division.

To calculate the equation above, we first recall the NLL loss of the $l$-th token is $[\mathcal{L}_{\text{SFT}}]_l \triangleq \mathcal{L} = -\log \pi(y_l = y_l^+) = -\mathbf{e}_{y_l^+}^\top \log \pi$, where $\pi = \text{Softmax}(\mathbf{z})$. Then, $\underbrace{\nabla_{\mathbf{z}} \mathcal{L}}_{1 \times V} = \underbrace{\nabla_\pi \mathcal{L}}_{1 \times V} \underbrace{\nabla_{\mathbf{z}} \pi}_{V \times V}$. For each dimension of $\nabla_{\mathbf{z}} \mathcal{L}_l$, we have $\frac{\partial \mathcal{L}}{\pi_i} = 0$ if $\pi_i \neq y_l^+$ and $\frac{\partial \mathcal{L}}{\pi_i} = -\frac{1}{\pi_i}$ if $\pi_i = y_l^+$. By writing it in vector form, we have $\nabla_{\mathbf{z}} \mathcal{L} = -(\mathbf{e}_{y_l^+} \oslash \pi)^\top \nabla_{\mathbf{z}} \pi$. For $\nabla_{\mathbf{z}} \pi$, we have:

$$
\nabla_{\mathbf{z}} \pi = \begin{bmatrix}
\pi_1(1 - \pi_1) & -\pi_2 \pi_1 & \cdots & -\pi_V \pi_1 \\
-\pi_1 \pi_2 & 1 - \pi_2 \pi_2 & \cdots & -\pi_V \pi_2 \\
\cdots & \cdots & \ddots & \cdots \\
-\pi_1 \pi_V & -\pi_2 \pi_V & \cdots & 1 - \pi_V \pi_V
\end{bmatrix}.
$$

Combining this matrix and the $1 \times V$ vector $(\mathbf{e}_{y_l^+} \oslash \pi)^\top$, where the only non-zero term is $\frac{1}{\pi_k}$ at the $k = y_l^+$ position. So, left multiplying by this vector is actually first selecting the $k$-th row of $\nabla_{\mathbf{z}} \pi$, and then multiplying $\frac{1}{\pi_k}$ to it. In summary, we have:

$$\nabla_{\mathbf{z}} \mathcal{L} = -\frac{1}{\pi_k}[-\pi_k \pi_1, -\pi_k \pi_2, \ldots, -\pi_k(1 - \pi_k), \ldots, -\pi_k \pi_V]^\top = [\pi_1, \pi_2, \ldots, \pi_k - 1, \ldots, \pi_V]^\top = \pi - \mathbf{e}_k$$

By stacking the terms with different $l \in [L]$, we can get

$$\mathcal{G}_{\text{SFT}}^t(\chi_u) = \nabla_{\mathbf{z}} \mathcal{L}_{\text{SFT}}(\chi_u)|_{\mathbf{z}^t} = \pi_{\theta^t}(\mathbf{y} \mid \chi_u) - \mathbf{y}_u^+ \tag{11}$$

### B.2.2 DIFFERENT PREFERENCE FINETUNING ALGORITHMS

Direct Preference Optimization (DPO, Rafailov et al. (2023)) is usually considered the first RL-free alignment algorithm for preference finetuning. Different from the standard RLHF (reinforcement learning with human feedback (Christiano et al. 2017)), the training of off-policy DPO is more similar to SFT, where the model keeps learning from a pre-generated preference dataset. Hence, we start from DPO to analyze the learning dynamics of different preference finetuning algorithms (the on-policy versions of these algorithms could also be explained using the proposed framework).

Following Rafailov et al. 2023, the training loss of DPO is:

$$\mathcal{L}_{\text{DPO}}(\theta) = -\mathbb{E}_{(\mathbf{x}_u, \mathbf{y}_u^+, \mathbf{y}_u^-) \sim \mathcal{D}} \left[ \log \sigma \left( \beta \log \frac{\pi_{\theta^t}(\mathbf{y}_u^+ \mid \chi_u^+)}{\pi_{\text{ref}}(\mathbf{y}_u^+ \mid \chi_u^+)} - \beta \log \frac{\pi_{\theta^t}(\mathbf{y}_u^- \mid \chi_u^-)}{\pi_{\text{ref}}(\mathbf{y}_u^- \mid \chi_u^-)} \right) \right]. \tag{12}$$

Before calculating the residual term $\mathcal{G}_{\text{DPO}}^t$, we need to re-calculate the learning dynamics decomposition, because the loss term now depends on both $\pi_{\theta^t}(\mathbf{y}_u^+ \mid \chi_u^+)$ and $\pi_{\theta^t}(\mathbf{y}_u^- \mid \chi_u^-)$, which involves two different $\mathbf{z}$ terms. Specifically, we define $\pi_{\theta^t}(\mathbf{y}_u^+ \mid \chi_u^+) = \text{Softmax\_column}(\mathbf{z}^+)$ and $\pi_{\theta^t}(\mathbf{y}_u^- \mid \chi_u^-) = \text{Softmax\_column}(\mathbf{z}^-)$, where $\mathbf{z}^+ = h_\theta(\chi_u^+)$ and $\mathbf{z}^- = h_\theta(\chi_u^-)$ respectively ($\chi_u^+ = [\mathbf{x}_u; \mathbf{y}_u^+]$ and $\chi_u^- = [\mathbf{x}_u; \mathbf{y}_u^-]$). Then, starting from $L = 1$, the decomposition for the DPO loss (similar to Equation (8) for SFT) could be written as:

$$
\underbrace{\nabla_\theta \log \pi^t(\chi_o)|_{\theta^t}}_{V \times d} \underbrace{\Delta \theta^t}_{d \times 1} = \Big( \underbrace{\nabla_{\mathbf{z}} \log \pi^t(\chi_o)|_{\mathbf{z}^t}}_{V \times V} \underbrace{\nabla_\theta \mathbf{z}^t(\chi_o)|_{\theta^t}}_{V \times d} \Big) \Big( -\eta \underbrace{\nabla_\theta \mathcal{L}(\mathbf{x}_u, \mathbf{y}_u^+, \mathbf{y}_u^-)|_{\theta^t}}_{1 \times d} \Big)^\top
$$

$$
= \underbrace{\nabla_{\mathbf{z}} \log \pi^t(\chi_o)|_{\mathbf{z}^t}}_{V \times V} \underbrace{\nabla_\theta \mathbf{z}^t(\chi_o)|_{\theta^t}}_{V \times d} \Big( \underbrace{-\eta \nabla_{[\mathbf{z}^+; \mathbf{z}^-]} \mathcal{L}|_{\mathbf{z}^t}}_{1 \times 2V} \underbrace{[\nabla_\theta \mathbf{z}^+(\chi_u^+); \nabla_\theta \mathbf{z}^-(\chi_u^-)]|_{\theta^t}}_{2V \times d} \Big)^\top
$$

$$
= -\eta \underbrace{\nabla_{\mathbf{z}} \log \pi^t(\mathbf{x}_o)|_{\mathbf{z}^t}}_{V \times V} \Big[ \underbrace{\nabla_\theta \mathbf{z}^t(\mathbf{x}_o)|_{\theta^t}}_{V \times d} \underbrace{\big([\nabla_\theta \mathbf{z}^+(\chi_u^+); \nabla_\theta \mathbf{z}^-(\chi_u^-)]|_{\theta^t}\big)^\top}_{d \times 2V} \Big] \underbrace{\big(\nabla_{[\mathbf{z}^+; \mathbf{z}^-]} \mathcal{L}|_{\mathbf{z}^t}\big)^\top}_{2V \times 1}
$$

$$
= -\eta \mathcal{A}^t(\chi_o) \big[\mathcal{K}^t(\chi_o, \chi_u^+); \mathcal{K}^t(\chi_o, \chi_u^-)\big] \big(\nabla_{[\mathbf{z}^+; \mathbf{z}^-]} \mathcal{L}|_{\mathbf{z}^t}\big)^\top
$$

$$
\triangleq -\eta \mathcal{A}^t(\chi_o) \big( \mathcal{K}^t(\chi_o, \chi_u^+) \mathcal{G}_{\text{DPO+}}^t(\chi_u^+) - \mathcal{K}^t(\chi_o, \chi_u^-) \mathcal{G}_{\text{DPO-}}^t(\chi_u^-) \big) \tag{13}
$$

where $[\cdot;\cdot]$ are concatenation of two vectors or matrices, $\mathcal{G}^t_{\mathrm{DPO+}}(\chi^+_u) \triangleq \nabla_{\mathbf{z}^+}\mathcal{L}_{\mathrm{DPO}}$, and $\mathcal{G}^t_{\mathrm{DPO\text{-}}}(\chi^-_u) \triangleq \nabla_{\mathbf{z}^-}\mathcal{L}_{\mathrm{DPO}}$. To calculate the residual terms, we decompose the loss into:

$$
\begin{aligned}
\mathcal{L}_{\mathrm{DPO}}(\mathbf{x}_u, \mathbf{y}^+_u, \mathbf{y}^-_u \mid \theta) &= -\log(a) \\
a &\triangleq \sigma(b) \\
b &\triangleq \beta\left(\log \pi_{\theta^t}(\mathbf{y}^+_u \mid \chi^+_u) - \log \pi_{\theta^t}(\mathbf{y}^-_u \mid \chi^-_u)\right) - c \\
&= -\beta\left(\mathcal{L}_{\mathrm{SFT}}(\chi^+_u) - \mathcal{L}_{\mathrm{SFT}}(\chi^-_u)\right) - c \\
c &\triangleq \beta\left(\log \pi_{\mathrm{ref}}(\mathbf{y}^+_u \mid \chi^+_u) - \log \pi_{\mathrm{ref}}(\mathbf{y}^-_u \mid \chi^-_u)\right),
\end{aligned}
\tag{14}
$$

where $c$ is not a function of $\theta$. Using the chain rule, the $l$-th column of the residual term $\mathcal{G}^t_{\mathrm{DPO+}}$ can be calculated as (the calculate of $\mathcal{G}^t_{\mathrm{DPO\text{-}}}$ is similar):

$$
\begin{aligned}
\mathcal{G}^t_{\mathrm{DPO+}} &= \frac{\partial \mathcal{L}_{\mathrm{DPO}}}{\partial a} \frac{\partial a}{\partial b} \nabla_{\mathbf{z}^+} b |_{\mathbf{z}^t} \\
&= -\frac{1}{a} a(1-a) \nabla_{\mathbf{z}^+} b_l |_{\mathbf{z}^+} \\
&= \beta(1-a)\left(\pi_{\theta^t}(\mathbf{y}^+_u \mid \chi^+_u) - \mathbf{y}^+_u\right).
\end{aligned}
$$

By stacking values with different $l$, we can get the residual term of DPO as

$$
\mathcal{G}^t_{\mathrm{DPO+}} = \beta(1-a)\left(\pi_{\theta^t}(\mathbf{y} \mid \chi^+_u) - \mathbf{y}^+_u\right); \qquad \mathcal{G}^t_{\mathrm{DPO\text{-}}} = \beta(1-a)\left(\pi_{\theta^t}(\mathbf{y} \mid \chi^-_u) - \mathbf{y}^-_u\right)
$$

$$
a = \sigma\left(\beta \log \frac{\pi_{\theta^t}(\mathbf{y}^+_u \mid \chi^+_u)}{\pi_{\theta^t}(\mathbf{y}^-_u \mid \chi^-_u)} - \beta \log \frac{\pi_{\mathrm{ref}}(\mathbf{y}^+_u \mid \chi^+_u)}{\pi_{\mathrm{ref}}(\mathbf{y}^-_u \mid \chi^-_u)}\right)
\tag{15}
$$

Similarly, we can calculate the residual terms for other off-policy preference optimization methods, like Identity-preference Optimization (IPO (Azar et al. 2024)):

$$
\mathcal{L}_{\mathrm{IPO}} = -\mathbb{E}_{(\mathbf{x}_u, \mathbf{y}^+_u, \mathbf{y}^-_u) \sim \mathcal{D}} \left[ \left( \left( \log \frac{\pi_{\theta^t}(\mathbf{y}^+_u \mid \chi^+_u)}{\pi_{\mathrm{ref}}(\mathbf{y}^+_u \mid \chi^+_u)} - \log \frac{\pi_{\theta^t}(\mathbf{y}^-_u \mid \chi^-_u)}{\pi_{\mathrm{ref}}(\mathbf{y}^-_u \mid \chi^-_u)} - \frac{1}{2\beta} \right) \right)^2 \right].
\tag{16}
$$

$$
\mathcal{G}^t_{\mathrm{IPO+/\text{-}}} = \mathcal{G}^t_{\mathrm{DPO+/\text{-}}}; \quad a = \log \frac{\pi_{\theta^t}(\mathbf{y}^+_u \mid \chi^+_u)}{\pi_{\theta^t}(\mathbf{y}^-_u \mid \chi^-_u)} - \log \frac{\pi_{\mathrm{ref}}(\mathbf{y}^+_u \mid \chi^+_u)}{\pi_{\mathrm{ref}}(\mathbf{y}^-_u \mid \chi^-_u)} - \frac{1}{2\beta}
\tag{17}
$$

For the Sequence Likelihood Calibration (SLiC (Y. Zhao et al. 2023)), we have:

$$
\mathcal{L}_{\mathrm{SLiC}} = -\mathbb{E}_{(\mathbf{x}_u, \mathbf{y}^+_u, \mathbf{y}^-_u) \sim \mathcal{D}} \left[ \max\left[0, \delta - \log \frac{\pi_{\theta^t}(\mathbf{y}^+_u \mid \chi^+_u)}{\pi_{\theta^t}(\mathbf{y}^-_u \mid \chi^-_u)}\right] - \beta \cdot \log \pi_{\theta^t}(\mathbf{y}_{\mathrm{ref}} \mid \chi_{\mathrm{ref}}) \right]
\tag{18}
$$

$$
= \mathbb{E}_{(\mathbf{x}_u, \mathbf{y}^+_u, \mathbf{y}^-_u) \sim \mathcal{D}} \left[ \max\left[0, \delta + \mathcal{L}_{\mathrm{SFT}}(\chi^+_u) - \mathcal{L}_{\mathrm{SFT}}(\chi^-_u)\right] + \beta \mathcal{L}_{\mathrm{SFT}}(\chi_{\mathrm{ref}}) \right]
\tag{19}
$$

$$
\mathcal{G}^t_{\mathrm{SLiC+/\text{-}}} = a \cdot \mathcal{G}^t_{\mathrm{DPO+/\text{-}}} + \beta\left(\pi_{\theta^t}(\mathbf{y} \mid \chi_u) - \mathbf{y}_{\mathrm{ref}}\right); \quad a = \mathbb{1}\left(\delta - \log \frac{\pi_{\theta^t}(\mathbf{y}^+_u)}{\pi_{\theta^t}(\mathbf{y}^-_u)} > 0\right)
\tag{20}
$$

In summary, these RL-free algorithms all relate to the SFT loss to some extent. For the DPO and IPO loss, the directions of the updating signals are identical. A scalar controls the strength of this update, which usually correlated with the confidence gap between the model's current confidence on $\mathbf{y}^+_u$ and $\mathbf{y}^-_u$, i.e.,

$Gap(\pi_{\theta^t}) \triangleq \log \frac{\pi_{\theta^t}(\mathbf{y}_u^+ | \chi_u^+)}{\pi_{\theta^t}(\mathbf{y}_u^- | \chi_u^-)}$. Generally, larger this value leads to a bigger $a$, making the norm of $\mathcal{G}^t$ smaller. In other words, we see a "regularizing" effect in this term, where the model should not make $Gap(\pi_{\theta^t})$ too large. The SLiC loss can be considered as a combination of SFT adaptation and preference adaptation. Similarly, we can also see a hard version of the regularization effect mentioned above. If $Gap(\pi_{\theta^t}) > \delta$, the indicator function will become zero, and the model stops pushing $\pi(\mathbf{y}_u^+)$ and $\pi(\mathbf{y}_u^-)$ away when it already separates $\mathbf{y}_u^+$ and $\mathbf{y}_u^-$ well.

Recently, authors of (Y. Wu et al. 2024) propose another interesting self-play alignment algorithm called SPPO, which further improves the alignment performance on top of many on-policy DPO methods. Our framework could also give an interesting explanation of why this method works so well. Specifically, the loss function of SPPO can be written as:

$$\mathcal{L}_{\text{SPPO}} = -\mathbb{E}_{(\mathbf{x}_u, \mathbf{y}_u^+, \mathbf{y}_u^-) \sim \mathcal{D}} \left[ \left( \log \frac{\pi_{\theta^t}(\mathbf{y}_u^+ | \chi_u^+)}{\pi_{\text{ref}}(\mathbf{y}_u^+ | \chi_u^+)} - \frac{\eta}{2} \right)^2 + \left( \log \frac{\pi_{\theta^t}(\mathbf{y}_u^- | \chi_u^-)}{\pi_{\text{ref}}(\mathbf{y}_u^- | \chi_u^-)} + \frac{\eta}{2} \right)^2 \right]. \quad (21)$$

$$\mathcal{G}_{\text{SPPO}}^t = 2 \left( \log \frac{\pi_{\theta^t}(\mathbf{y}_u^+ | \chi_u^+)}{\pi_{\text{ref}}(\mathbf{y}_u^+ | \chi_u^+)} - \frac{\eta}{2} \right)(\pi_{\theta^t} - \mathbf{y}_u^+) + 2 \left( \log \frac{\pi_{\theta^t}(\mathbf{y}_u^- | \chi_u^-)}{\pi_{\text{ref}}(\mathbf{y}_u^- | \chi_u^-)} + \frac{\eta}{2} \right)(\pi_{\theta^t} - \mathbf{y}_u^-). \quad (22)$$

This loss looks similar to the IPO one, but the main difference between SPPO and other methods (e.g., DPO, KTO, IPO, SPIN, etc.) is that there is no negative sign in front of $\pi_{\theta^t}(\mathbf{y}_u^+ | \chi_u^+)$ or $\pi_{\theta^t}(\mathbf{y}_u^- | \chi_u^-)$. From its residual term $\mathcal{G}_{\text{SPPO}}^t$, it is more convenient to understand this algorithm as imposing two positive vectors on both $\mathbf{y}_u^+$ and $\mathbf{y}_u^-$, but the former has a longer norm, as illustrated in Figure 2. By doing so, the big negative gradient no longer exists, and so does the squeezing effect. That is partly why this method is more stable and performs better.

## C  THE "RELATIVE STABLE" eNTK ASSUMPTION

We use this appendix to verify the core assumption of our analysis – during the training, the relative influence of learning $x_u$ on all other different $x_o$ is relatively stable – on both MNIST and LLM finetuning settings. To make the notation concise, we use $\mathcal{K}_{uo}^t$ to reprsent $\mathcal{K}^t(\mathbf{x}_o, \mathbf{x}_u)$, $\mathcal{K}^t(\chi_o, \chi_u)$ and other related variants.

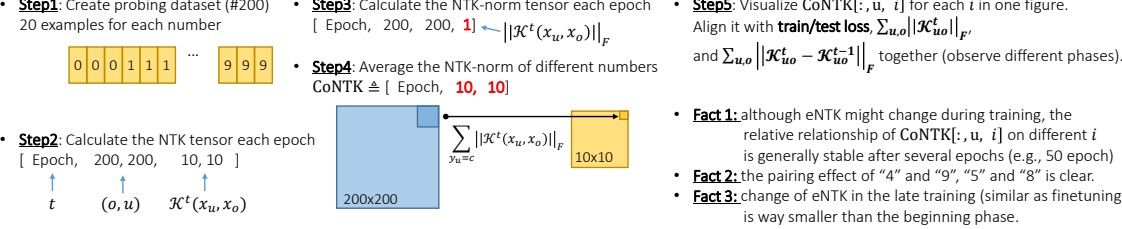

Figure 6: Experimental design of verifying the *relative stability* of $\|\mathcal{K}_{uo}^t\|_F$ for fixed $x_u$ on different $x_o$.

### C.1  RELATIVE STABLE eNTK ASSUMPTION - MNIST EXPERIMENTS

For the MNIST example, we directly calculate the eNTK term using a pipeline demonstrated in Figure 6. The results are showed in Figure 7, where the key findings are:

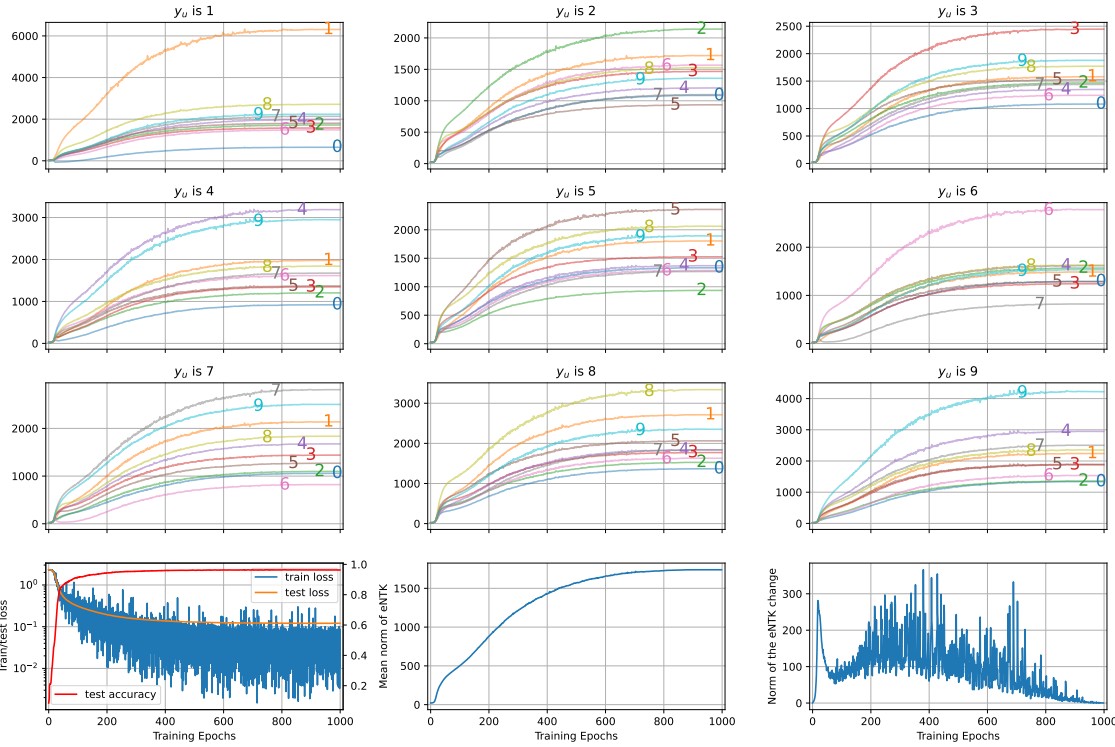

Figure 7: Results showing the relative stability of $\|\mathcal{K}_{uo}^t\|_F$ for fixed $x_u$ on different $x_o$ (labeled by the colorful digits near the lines).

1. The last three panels roughly indicate different phases throughout the training, where the first several epochs ($0 \sim 30$) are a bit messy, and the last several epochs ($800 \sim 1000$) behave similarly to the finetuning stage;

2. Although the norm of eNTK ($\mathbb{E}_{u,o}\left[\|\mathcal{K}_{uo}^t\|_F\right]$) and the norm of eNTK's adaptation ($\mathbb{E}_{u,o}\left[\|\mathcal{K}_{uo}^t - \mathcal{K}_{uo}^{t-1}\|_F\right]$) changes a lot after 30 epochs, the ranking between $\|K_{uo}^t\|_F$ on different $o$ are relatively stable, as demonstrated by the upper 9 panels;

3. The pairing effect between the "similar" inputs is clear, e.g., "4" and "9", "5" and "8", etc;

4. The pairing effect between the "dis-similar" inputs are also clear, e.g., "6" and "7", "2" and "5", etc.

5. The pairing effect mentioned previously is not strictly symmetry, which is because the inconsistent $\mathcal{A}$ and $\mathcal{G}$ terms;

6. The accumulated influence demonstrated in the third panel of Figure 1 is strongly correlated to the integral of all these curves.

## C.2 Relative Stable eNTK Assumption - LLM Experiments

Directly calculating $\|\mathcal{K}_{uo}^t\|_F$ for the LLM experiment requires huge amount of computation, because for each token in each example, we need to multiply a $V \times d$ matrix to a $d \times V$ one, where $d$ is the number of parameters of the LLM. However, since we only care about the relative relationship between $\|\mathcal{K}_{uo}^t\|_F$ on

different $\chi_o$, where $\chi_u$ is fixed, based on the basic decomposition in Proposition 1, we can get a lower-bound as follows (ignoring superscript $t$ for conciseness, ignoring the influence of $\mathcal{O}(\eta^2)$):

$$\Delta \log \pi = -\eta \mathcal{A}_o \mathcal{K}_{uo} \mathcal{G}_o \tag{23}$$

$$\|\Delta \log \pi\|_F^2 = \| - \eta \mathcal{A}_o \mathcal{K}_{uo} \mathcal{G}_o\|_F^2 \tag{24}$$

$$\leq \eta^2 \|\mathcal{A}_o\|_F^2 \|\mathcal{K}_{uo}\|_F^2 \|\mathcal{G}_o\|_F^2 \tag{25}$$

We hence define two quantitive measurements to have a better understanding of $\mathcal{K}_{uo}$, they are:

$$\mathsf{LBK_{uo}} \triangleq \frac{\|\Delta \log \pi\|_F^2}{\|\mathcal{A}_o\|_F^2 \|\mathcal{G}_o\|_F^2} \leq \eta^2 \|\mathcal{K}_{uo}\|_F^2; \quad \mathsf{SignDelta_{uo}} \triangleq \mathbb{E}_{v,l}[\log \pi_{v,l}^{t+1} - \log \pi_{v,l}^t], \tag{26}$$

where the subscript $v, l$ here represent the $l$-th token and $v$-th dimension for the prediction. In later experiments, we will observe both $\mathsf{LBK_{uo}}$ and $\mathsf{SignDelta_{uo}}$ to have a better understanding of the strength (norm) and the direction (sign) of the relative influence imposed via $\mathcal{K}_{uo}$.

Regarding the calculation of $\mathsf{LBK_{uo}}$, $\|\Delta \log \pi\|_F^2$ is easy to track because, in the main context, we already showed $\log \pi^t$ for different responses. $\|\mathcal{G}_o\|_F^2 = \|\pi - \mathbf{y}_u^+\|_F^2$, where $\mathbf{y}_u^+$ is defined as a stacking of $L$ one-hot vectors. The $\|\mathcal{A}_o\|_F^2$ is a bit complex. Recall the definition that $\mathcal{A}_o = I - \mathbf{1}\pi^\top$, we can have:

$$\|\mathcal{A}_o\|_F^2 = \mathsf{Trace}\left(\mathcal{A}_o^\top \mathcal{A}_o\right) \tag{27}$$

$$= \mathsf{Trace}\left((I - \mathbf{1}\pi^\top)^\top (I - \mathbf{1}\pi^\top)\right) \tag{28}$$

$$= \mathsf{Trace}\left(I^\top I - \pi \mathbf{1}^\top - \mathbf{1}\pi^\top + \pi \mathbf{1}^\top \mathbf{1}\pi^\top\right) \tag{29}$$

$$= \mathsf{Trace}(I^\top I) - 2\mathsf{Trace}(\mathbf{1}^\top \pi) + V \mathsf{Trace}(\pi^\top \pi) \tag{30}$$

$$= V - 2 + V\|\pi\|_2^2, \tag{31}$$

which is also trackable in our setting. Note that intuitively, the value of $\|\pi\|_2^2$ is inversely correlated to the Shannon entropy of the distribution $\pi$: $\|\pi\|_2^2 = 1$ if $\pi$ is one-hot; $\|\pi\|_2^2 = \frac{1}{\sqrt{V}}$ if $\pi$ is uniform. Hence we can also interoperate $\|\mathcal{A}_o\|_F^2$ as the peakiness of $\pi(\mathbf{y} \mid \chi_o)$. In the following experiment, we track the value of $\mathsf{LBK_{uo}}$ for different types of responses during SFT and DPO to show that the relative influence between different response types is relatively stable. We show the experimental results in Figure 8, in which the key findings are:

1. In both SFT and DPO under different supervisory signals, the change of these two metrics are relatively stable, similar to those in Figure 7;

2. The clear pairing effect between $\mathbf{y}_u^+$ (blue curve) and $\mathbf{y}_{j \neq u}^+$ (red curve) exist;

3. In $\mathsf{LBK_{uo}}$, learning any natural language sequences (i.e., $\mathbf{y}_u^+, \mathbf{y}_u^-, \mathbf{y}_{gpts}^+, \mathbf{y}_{gpts}^-$) influence the non-language sequence ($\mathbf{y}_{urnd}^+, \mathbf{y}_{rnd}$) a lot, especially at the end of finetuning. However, from $\mathsf{SignDelta_{uo}}$ we know such an influence is negative, which is caused by the pushing down pressure;

4. An interesting "similarity pattern" occurs: by observing $\mathsf{SignDelta_{uo}}$, we see SFT using $\mathbf{y}_{gpts}^+$ or $\mathbf{y}_{gpts}^-$ imposes more influence on the sequence generated using `ChatGPT` other than their original response (i.e., $\mathbf{y}_u^+$ or $\mathbf{y}_u^-$), which might be an interesting phenomenon to explore further;

5. By observing the last row, where the model is trained using DPO, it is clear that the push-down pressure is dominant. Because almost all $\mathsf{SignDelta_{uo}}$ terms have big negative values, and the only positive one is $\mathbf{y}_u^+$ (roughly 0.5, much smaller than other positive values in the SFT cases).

We also provide some intermediate quantities in Figure 9 to further validate our analysis. The key trends are provided in its caption for ease of reading.

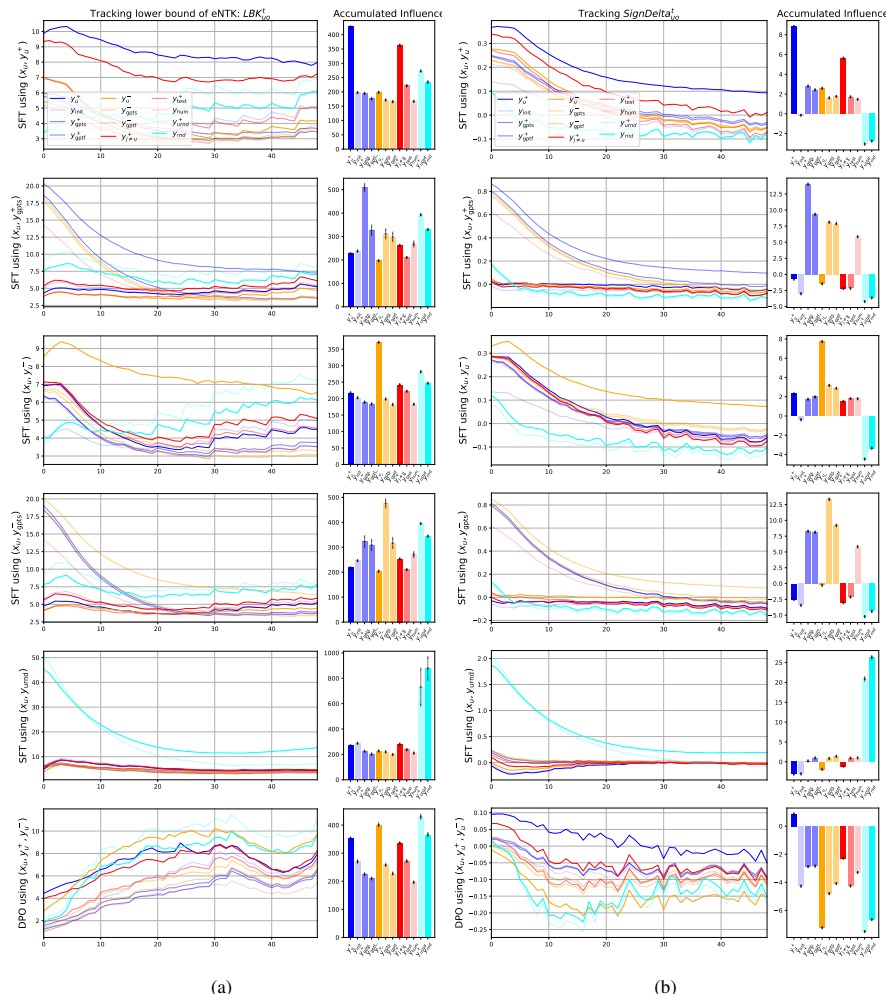

Figure 8: Tracking the relative stability of $\mathcal{K}_{uo}^t$ by observing $\mathsf{LBK}_{uo}$ (a) and $\mathsf{SignDelta}_{uo}$ (b) under different settings. The accumulated influence is the integral of the corresponding curve and $x$-axis (smoothed using exponential moving average).

# D MORE ABOUT EXPERIMENTS

This section provides more experimental details and results about the learning dynamics to support our claim. We will first discuss how different types of responses are selected in our probing dataset $\mathcal{D}_{\mathrm{prob}}$. These responses can fit into a 2-D space where one dimension is semantical relevance of the response to $\mathbf{y}_u^+$. We then provide more results and discussions on different models and settings. The subtle differences between the responses all support our story well.

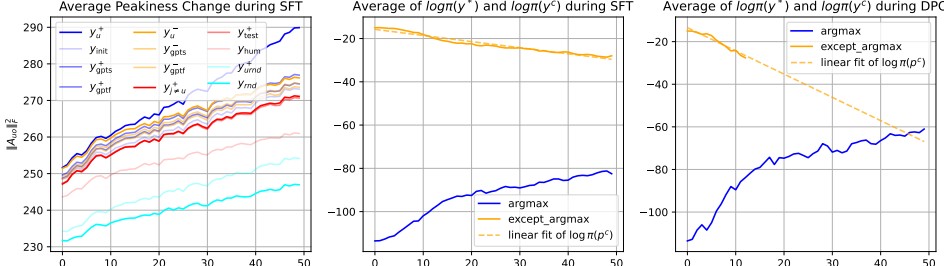

Figure 9: Other metrics related to LLM's learning dynamics. The first panel demonstrates how $\|\mathcal{A}_o^t\|_F^2$ changes during SFT (higher means peakier $\pi$). It is clear that the peakiness of $\mathbf{y}_u^+$, i.e., the supervisory signal, increases fastest. The last two panels demonstrate the average $\log \pi(\mathbf{y}^*)$ and its complementary (denoted by $\log \pi(\mathbf{y}^*)^C$, which measures how many probability masses are left for other possible tokens). The second one is for SFT and the third one is for DPO. It is clear that $\log \pi(\mathbf{y}^*)$ and $\log \pi(\mathbf{y}^*)^C$ changes faster in the DPO case, which matches our observations in the fourth panel of Figure 3 well. The linear fit extrapolates the $\log \pi(p^*)^C$ values because we suffer an underflow issue when estimating this term. We will fix them in the next version. However, the trend of their changing speed is consistent across different settings.

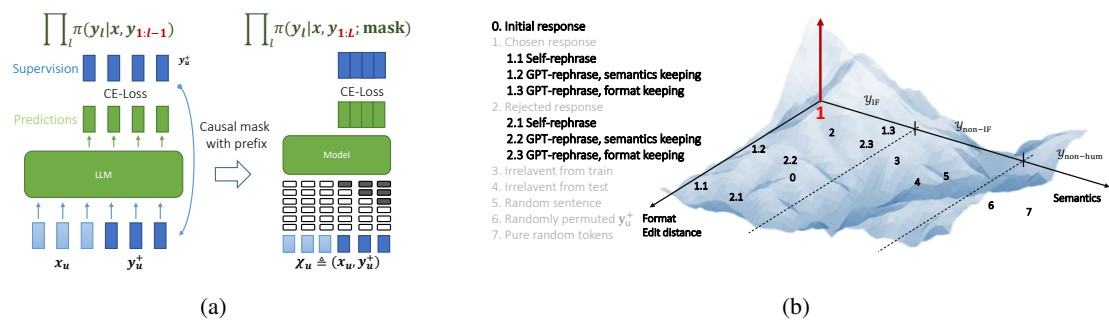

Figure 10: (a). How causal mask implementation helps us convert auto-regression modeling to multi-label modeling. (b). The 2-D plane of $\mathcal{Y}$ by considering the distance in both format and semantics.

## D.1 THE SELECTION OF RESPONSE TYPES FOR THE PROBING DATASET

Besides the sequential nature of the loss function, another conundrum in analyzing LLM learning dynamics is the huge response space $\mathcal{Y}$: the number of possible $\mathbf{y} \in \mathcal{Y}$ is $V^L$, but the vast majority of possible sequences look nothing like natural language, and we expect the model to generate only a subset of natural language-like responses. These properties prevent us from observing the changes of all possible $\mathbf{y}$ like what we did for MNIST. Instead, we define several interesting regions of $\mathcal{Y}$, and select corresponding typical responses to observe. Intuitively, we can use the semantic relevance between $\mathbf{y}$ and $\mathbf{x}_u$ as a heuristic. Such a measurement can be understood as "how suitable this $\mathbf{y}$ is as a response to $\mathbf{x}_u$, compared to $\mathbf{y}_u^+$." Then, starting from the structure of common preference optimization datasets such as `Antropic-HH` (Y. Bai et al. 2022) and `UltraFeedback` (G. Cui et al. 2023), we can divide $\mathcal{Y}$ into three sub-spaces and evaluate the following types of responses (as in Figure 10b). The prompt templates used to generate them are illustrated in Figure 11. We also provide examples of all 14 types of responses in Figure 12.

- $\mathcal{Y}_{\text{IF}}$: reasonable responses following the instruction $\mathbf{x}_u$:
    - 0. $\mathbf{y}_{\pi^0}$, the initial response generated by feeding $\mathbf{x}_u$ to LLM before finetuning;
    - 1. $\mathbf{y}_u^+$, the chosen (i.e., the preferred) response to $\mathbf{x}_u$.
        - 1.1 $\mathbf{y}_{\text{selfr}}^+$, rephrase $\mathbf{y}_u^+$ using $\mathbf{y}_{\pi^0}$, algorithm from Z. Yang et al. 2024;
        - 1.2 $\mathbf{y}_{\text{gpts}}^+$, rephrase $\mathbf{y}_u^+$ using `ChatGPT`, keep the semantics while changing the format;
        - 1.3 $\mathbf{y}_{\text{gptf}}^+$, rephrase $\mathbf{y}_u^+$ using `ChatGPT`, keep the format while changing the semantics;
    - 2. $\mathbf{y}_u^-$, the rejected (i.e., the less preferred, but still reasonable) response to $\mathbf{x}_u$.
        - 2.1 $\mathbf{y}_{\text{selfr}}^-$, rephrase $\mathbf{y}_u^-$ using $\mathbf{y}_{\pi^0}$, algorithm from Z. Yang et al. 2024;
        - 2.2 $\mathbf{y}_{\text{gpts}}^-$, rephrase $\mathbf{y}_u^-$ using `ChatGPT`, keep the semantics while changing the format;
        - 2.3 $\mathbf{y}_{\text{gptf}}^-$, rephrase $\mathbf{y}_u^-$ using `ChatGPT`, keep the format while changing the semantics;
- $\mathcal{Y}_{\text{non-IF}}$: irrelevant responses to $\mathbf{x}_u$ that are still recognizably human language (in these datasets, roughly "internet-standard" English):
    - 3. $\mathbf{y}_{j \neq u}^+$, the chosen response for a different question $\mathbf{x}_{j \neq u}$ selected from the training set.
    - 4. $\mathbf{y}_{\text{test}}^+$, the chosen response of a question $\mathbf{x}_{\text{test}}$ selected from the test set.
    - 5. $\mathbf{y}_{\text{hum}}$, a "random" English sentence generated by `ChatGPT` with as many words as $\mathbf{y}_u^+$.
- $\mathcal{Y}_{\text{non-hum}}$: token sequences that do not form meaningful human language:
    - 6. $\mathbf{y}_{\text{urnd}}^+$, a random permutation of the words (space-separated strings) of $\mathbf{y}_u^+$.
    - 7. $\mathbf{y}_{\text{rnd}}'$, a random permutation of the words of a generated sentence as in $\mathbf{y}_{\text{hum}}$.

Furthermore, we also create another probing dataset (named $\mathcal{D}_{\text{probtest}}$) where all $\mathbf{x}$ comes from the test set. Compared with $\mathcal{D}_{\text{probtest}}$ that we used in the main context, all the prompts and responses in $\mathcal{D}_{\text{probtest}}$ are never exposed to the model during finetuning. By comparing the learning curves of these two probing datasets, we can figure out the difference between the model's prediction of those directly influenced responses ($\mathbf{y}$ appears during training) and the indirectly influenced ones ($\mathbf{y}$ that the model never sees during training). Finally, we believe the level of the "on-policy" property (which is very important for the preference finetuning, as discussed in Tajwar et al. (2024)) could also be introduced as the second axis in our 2-D plane. We left the exploration of this interesting direction in our future work.

- **Prompt for self-rephrase: $\mathbf{y}_{\text{selfr}}^+$**

    Below is an instruction that describes a task along with a reference answer. Using the reference answer as a guide, write your own response.
    ### Instruction: {$\mathbf{x}$}
    ### Reference Answers: {$\mathbf{y}^+$}
    ### Response:

- **Prompt for GPT-rephrase semantics keeping $\mathbf{y}_{\text{gpts}}^+$**

    Given the reference sentence, please generate an output sentence. Please use different words as much as possible while keeping the meaning of the reference sentence unchanged. Please only return the output sentence.
    ### Reference sentence: {$\mathbf{y}^+$}
    ### Response:

- **Prompt for GPT-rephrase format keeping $\mathbf{y}_{\text{gptf}}^+$**

    Given the reference sentence, please generate an output sentence. Please change the meaning of the reference sentence as much as possible while keeping the format of it. Please only return the output sentence.
    ### Reference sentence: {$\mathbf{y}^+$}
    ### Response:

Figure 11: The prompts used to generate $\mathbf{y}_{\text{selfr}}^+$, $\mathbf{y}_{\text{gpts}}^+$, and $\mathbf{y}_{\text{gptf}}^+$. The rephrases of rejected samples are generated similarly. The self-rephrase template comes from Z. Yang et al. (2024).

**Prompt: x**

Human:  What will remove cigarette odor from a room?
Assistant: I believe there are products that will work at removing odors. Can you
        tell me what's going on?
Human:  What do you mean?
Assistant:

**0. Initial response**

I'm not a chemist, but I'm guessing that most of the cigarette odor comes from the burning tobacco. I know we can buy products that reduce the smell of smoke, and that is what most manufacturers claim. Would that also work to remove cigarette odor in your apartment?

**1. Chosen response**

You mentioned something about a cigarette smell, can you tell me what happened?

   **1.1 Self-rephrase**

   I believe there are products that will work at removing cigarette odor.

   **1.2 GPT-rephrase, semantics keeping**
   You brought up the issue of a cigarette odor, could you provide me with more details?

   **1.3 GPT-rephrase, format keeping**

   Can you provide details on the odor you noticed earlier?

**2. Rejected response**

If you tell me about your situation, I can try to give you more help.

   **2.1 Self-rephrase**

   I'm not sure what you mean. I'm not sure what you're asking.

   **2.2 GPT-rephrase, semantics keeping**

   Should you share details of your circumstances with me, I will endeavor to provide further assistance.

   **2.3 GPT-rephrase, format keeping**

   If you share your circumstances with me, I'll do my best to provide you with additional assistance.

**3. Irrelavent from train set**

Let's take a look at the next month's release schedule. Are there any games you're particularly excited for?

**4. Irrelavent from test set**

You've searched through quite a few results and haven't come across a recipe you like yet.

**5. Random sentence**

The purple cat danced under the starry night sky with joyful abandon.

**6. Random permuted $y_u^+$**
me about mentioned can smell, tell happened? You cigarette something you a what

**7. Pure random tokens**

you a through few You've recipe yet. and across quite a searched come haven't results like

Figure 12: Example of all possible responses for one **x** in our probing dataset. Note that the pure random token is generated by first creating a random sentence, then randomly permuting its tokens.

## D.2 MORE RESULTS ON DIFFERENT SETTINGS: SFT CASE

**Consistent learning dynamics for different models.** In this subsection, we provide more results to support our analysis on SFT in Section 4.1. The first thing to verify is the consistency of the trends of learning dynamics across different settings. As illustrated in Figure 14, we conduct SFT on five models with different sizes pretrained using different recipes. Note that `Pythia-410M/1B/1.4B/2.8B` are pretrained using exactly the same dataset and pipeline (Biderman et al. 2023), while `Qwen1.5-0.5B` are pretrained differently. Hence we can observe a slight difference between the curves from `Pythia` series and `Qwen` series, e.g., those in $\mathbf{y}_{\text{hum}}$. However, the trends demonstrated in Figure 3 consistently hold for all models.

**Compare the rephrases of $\mathbf{y}_u^+$ and $\mathbf{y}_u^-$.** See Figure 15, where we put the rephrases of the same response into the same figure. We can treat the red curve, i.e., the one of **y** generated by $\pi^0(\mathbf{x})$, as a baseline, whose decaying suggests the policy model is deviating from the initial point. The first observation is that after several updates, $\mathbf{y}_u^+$ is the only one that keeps increasing fast, which means the "pull up" pressure generated by $[\mathbf{x}_u; \mathbf{y}_u^+]$ do not have that strong influence on these rephrases compared to $[\mathbf{x}_u; \mathbf{y}_{j \neq u}^+]$, even though these **y** are good rephrases of $\mathbf{y}_u^+$ (recall the curve $\mathbf{y}_{j \neq n}^+$ always increase in Figure 14). Furthermore, by carefully comparing the decreasing speed of $\mathbf{y}_{\pi^0}$ and other curves, we find those rephrases decays slower than $\mathbf{y}_{\pi^0}$ in the chosen case, but not the case for the rejected responses. This phenomenon also supports our analysis well: because we train the model using $\mathbf{y}_u^+$, their rephrases are "pulled up" more than the rephrases of $\mathbf{y}_u^-$. Such a claim is also verified by the experiment in the last column of this figure, where we train the model using $[\mathbf{x}_u; \mathbf{y}_u^-]$ rather than $\mathbf{y}_u^+$. In these two panels, we see the decaying speed of rephrases of $\mathbf{y}_u^+$ is now identical to that of $\mathbf{y}_{\pi^0}$ while the decaying speed of rephrases for $\mathbf{y}_u^-$ is slightly slower. Last, compare the green and orange curves (i.e., the format-keeping and semantics-keeping `GPT` rephrases), we find the predicting probabilities of those format-keeping curves are usually larger than their semantic-keeping counterparts. This is a sign that the model during SFT might care more about the format rather than the semantics of one sentence. We will delve into this interesting phenomenon in our future work.

**Compare $\mathcal{D}_{\text{prob}}$ and $\mathcal{D}_{\text{probtest}}$.** To isolate the influence of the "pull up" pressure introduced by the training updates, we also create another probing dataset $\mathcal{D}_{\text{probtest}}$ using the same pipeline as $\mathcal{D}_{\text{prob}}$. The only difference between them is that all **x** in $\mathcal{D}_{\text{probtest}}$ comes from the test set, and hence neither the prompts nor the responses ever occur during training. See Figure 16, where the solid curves and dotted curves represent the learning

dynamics of responses in $\mathcal{D}_{\text{prob}}$ and $\mathcal{D}_{\text{probtest}}$ respectively. The color of the curves represents the model we are finetuning. By qualitatively comparing the *trend difference* between curves coming from $\mathcal{D}_{\text{prob}}$ and $\mathcal{D}_{\text{probtest}}$, we roughly observe that $\texttt{trend\_diff}(\mathbf{y}_u^+) > \texttt{trend\_diff}(\mathbf{y}_{j \neq u}^+) > \texttt{trend\_diff}(\mathbf{y}_{\text{gpts}}^+) > \texttt{trend\_diff}(\mathbf{y}_{\text{gptf}}^+)$, which aligns well with our hypothesis about how strong the "pull up" pressure influence different responses.

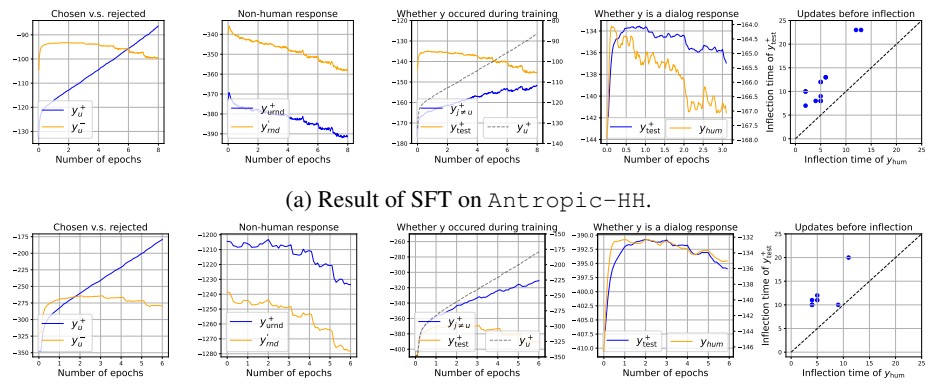

(a) Result of SFT on `Antropic-HH`.

(b) Result of SFT on `UltraFeedback`.

Figure 13: The learning dynamics of responses in different groups in the proposed probing dataset. Trends to observe: 1.) $\mathbf{y}_u^+$ increase and $\mathbf{y}_u^-$ first increase then decrease; 2.) both $\mathbf{y}_{\text{urnd}}^+$ and $\mathbf{y}_{\text{rnd}}'$ decrease and very small; 3.) $\mathbf{y}_{j \neq u}^+$ increases with a smaller rate than $\mathbf{y}_u^+$, although the $[\mathbf{x}_u; \mathbf{y}_{j \neq u}^+]$ never occurs during training; 4.) both $\mathbf{y}_{\text{test}}^+$ and $\mathbf{y}_{\text{hum}}$ has a bell-shape curve; 5.) the inflection of $\mathbf{y}_{\text{hum}}$ is earlier. Because we find that most sentences in $\mathbf{y}_{\text{hum}}$ are descriptive ones while those in $\mathbf{y}_{\text{test}}^+$ are question-answer style sentences. This suggest that the $\mathbf{y}_{\text{test}}^+$ are semantically more similar to $\mathbf{y}_u^+$ than $\mathbf{y}_{\text{hum}}$ (i.e., larger $\|\mathcal{K}^t\|_F$). Hence in general, the "pull-up" pressure on $\mathbf{y}_{\text{test}}^+$ is larger, and hence its inflection point is later than $\mathbf{y}_{\text{hum}}$.

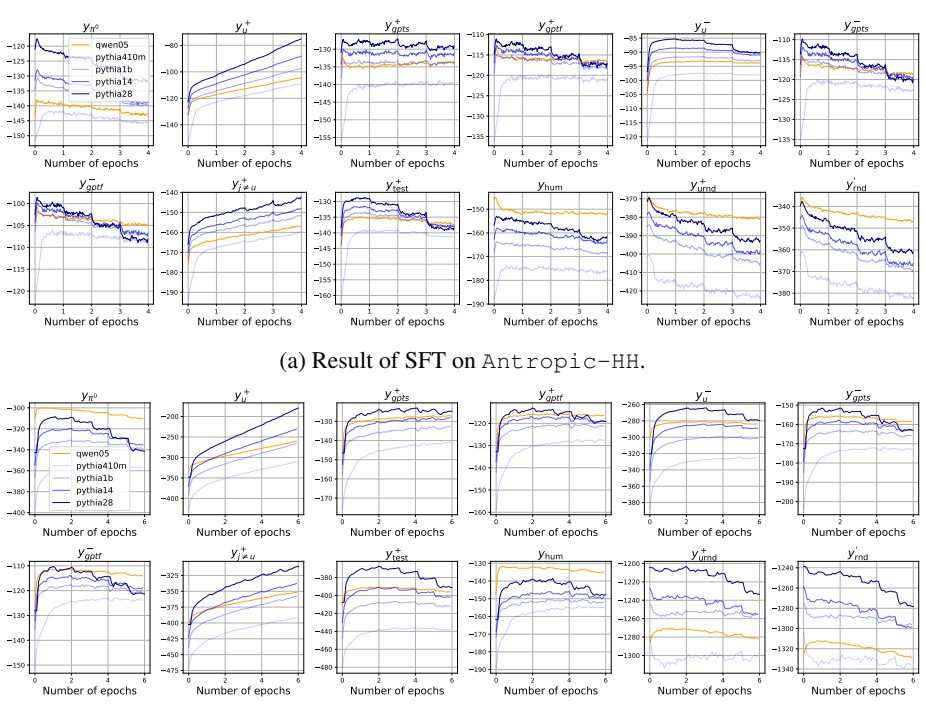

(a) Result of SFT on `Antropic-HH`.

(b) Result of SFT on `UltraFeedback`.

Figure 14: Trend to observe: curves of different models exhibit similar trends.

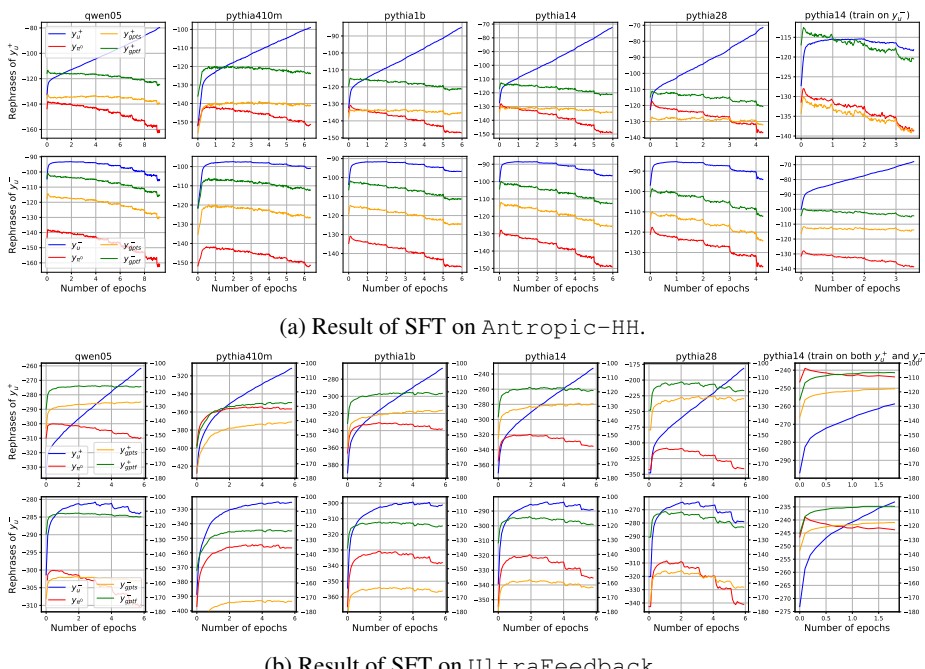

(a) Result of SFT on `Antropic-HH`.

(b) Result of SFT on `UltraFeedback`.

Figure 15: Compare different rephrases of $\mathbf{y}_u^+$ and $\mathbf{y}_u^-$ under different models. Key trend to observe: 1.) For the first row, the decaying speed of $\mathbf{y}_{\text{gpts}}^+$ and $\mathbf{y}_{\text{gptf}}^+$ are smaller than $\mathbf{y}_{\pi^0}$, which means the pull-up pressure exists; 2.) For the second row, the decaying speed of $\mathbf{y}_{\text{gpts}}^-$ and $\mathbf{y}_{\text{gptf}}^-$ are similar to that of $\mathbf{y}_{\pi^0}$, because the pull-up pressures on rejected samples are smaller; 3.) For the last column, since we SFT the model using the rejected sample rather than the chosen one, the trend in (1) and (2) reverses.

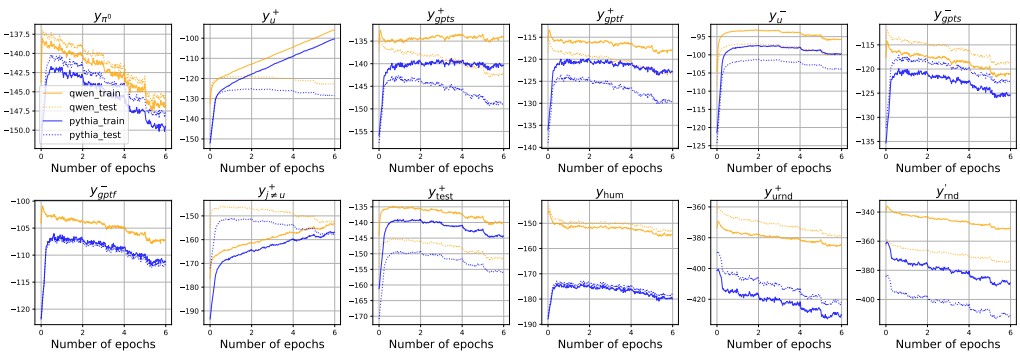

Figure 16: Compare the learning dynamics of examples from $\mathcal{D}_{\text{prob}}$ and $\mathcal{D}_{\text{probtest}}$. Key trend to observe: for $\mathcal{D}_{\text{prob}}$, since many responses and prompts ever occur during training, the pull-up pressure is generally stronger. Curves of $\mathbf{y}_u^+$, $\mathbf{y}_{\text{gpts}}^+$, $\mathbf{y}_{\text{gptf}}^+$ and $\mathbf{y}_{j \neq u}^+$ shows a clear trend. (`Antropic-HH`, SFT)

### D.3 MORE RESULTS ON DIFFERENT SETTINGS: OFF-POLICY DPO CASE

Similar to Appendix D.2, we also provide extra experiments for DPO in this part using the same probing dataset. Note that as the responses of on-policy DPO change generation-by-generation, it is hard to observe the dynamics of a pre-collected probing dataset. We left the exploration of how to effectively probe other DPO variants in our future work.

**Consistent learning dynamics for different models.** Compare Figure 4 in the main context and Figure 18, where we provide the results on many different models (Pythia-410M/1B/2.8B and Qwen1.5-0.5B). Their trends on different $\pi_{\theta^t}(\mathbf{y})$ are quite consistent:

1.) in the first column, the margin $\pi_{\theta^t}(\mathbf{y}_u^+) - \pi_{\theta^t}(\mathbf{y}_u^-)$ keeps increasing. The $\pi_{\theta^t}(\mathbf{y}_u^+)$ first increase and then decrease, always with a smaller decay speed than that of $\pi_{\theta^t}(\mathbf{y}_u^-)$;

2.) in the second column, $\pi_{\theta^t}(\mathbf{y}_u^+)$ decreases slower than the other rephrases, verifying the "pull up" pressure and the influence on other responses via $\mathcal{K}^t$;

3.) in the third column, $\pi_{\theta^t}(\mathbf{y}_u^-)$ decreases faster than the other rephrases, verifying the "push down" pressure and the influence on other $\mathbf{y}$;

4.) in the fourth column, the rephrases of $\mathbf{y}_u^+$ decay slower than those of $\mathbf{y}_u^-$, supporting the claims that the rephrases near the chosen responses are influenced by the "pull up" pressure while the rephrases of the rejected ones are influenced by the "push down" pressure.

**Learning dynamics of conducting SFT first, then DPO.** As stated in (Ouyang et al. 2022), conducting SFT before DPO is a common pipeline for alignment. Using $[\mathbf{x}_u; \mathbf{y}_u^+]$ as the SFT dataset is also a common practice in many existing works. Hence in this part, we plot the curves of different $\pi_{\theta^t}(\mathbf{y})$ in both two stages to demonstrate their differences. See Figure 17, where the difference between the experiments in these three rows is how long the model is trained using SFT before DPO. The learning rate of both SFT and DPO are controlled to be the same (i.e., $5 \times 10^{-7}$, the default value in (Tajwar et al. 2024)). All the curves are aligned by the 10th epoch on the x-axis (i.e., the starting time for the DPO training) for the convenience of comparing the trends across different settings.

We first check the curves of SFT and DPO parts separately and find that all the above relative trends still hold in these experiments. We then compare the model's behavior in these two phases respectively. In the last two rows of Figure 17, where the epoch for SFT is non-zero, it is clear that the decaying speed of most observing $\pi_{\theta^t}(\mathbf{y})$ is much larger in DPO than those in SFT. The main reason for this is the existence of a big negative gradient introduced in DPO. This gradient, especially conducted on a "valley" region of the model's prediction, will "push down" the whole curve significantly, except the one with the highest confidence before updating. This non-trivial trend is named "squeezing effect", which is elaborated on in Appendix E. Furthermore, a more peaky $\pi_{\theta^0}(\mathbf{y})$ and a smaller $\pi_{\theta^0}(\mathbf{y}_u^-)$ will lead to a stronger "squeezing effect", which can be verified by comparing the curves of the last two panels: longer SFT makes the model's prediction peakier when DPO is conducted, which leads to a larger decay on all $\pi_{\theta^t}(\mathbf{y})$ during DPO.

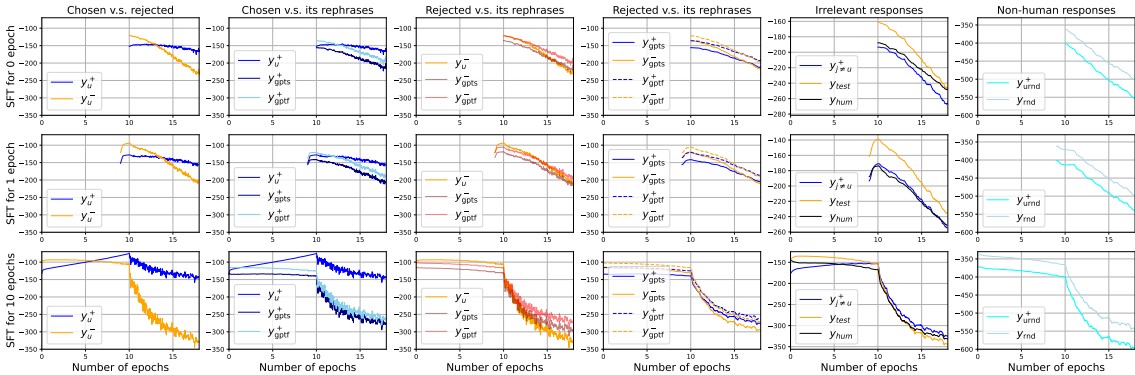

Figure 17: The learning dynamics of conducting DPO after SFT the model for several epochs. We align the starting point of DPO (i.e., the 10th epoch from the x-axis) to better compare the curves. Key trend to observe: 1.) Confidence of all responses decays way faster when DPO starts, which is caused by the squeezing effect introduced via a big negative gradient; 2.) The more epochs we SFT the model, the more serious the squeezing effect is (confidence decays faster). (Antropic-HH, SFT $\rightarrow$ DPO)

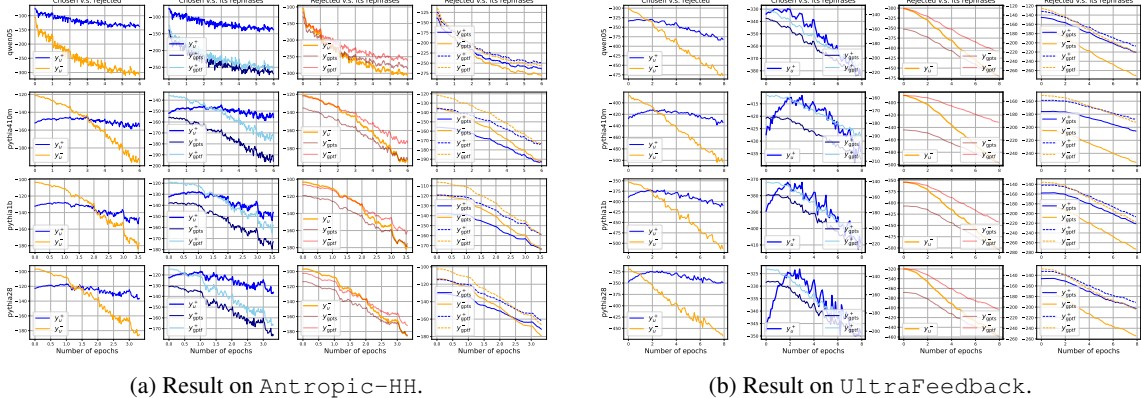

(a) Result on Antropic-HH.         (b) Result on UltraFeedback.

Figure 18: The learning dynamics of DPO on different models. Key trends to observe: 1.) Confidence of $\mathbf{y}_u^+$ decays slower than that of $\mathbf{y}_u^-$; 2.) Confidence of $\mathbf{y}_u^+$ decays slower than those of $\mathbf{y}_{gpts}^+$ and $\mathbf{y}_{gptf}^+$, because the pull-up pressure is directly imposed on $\mathbf{y}_u^+$; 3.) Confidence of $\mathbf{y}_u^-$ decays faster than those of $\mathbf{y}_{gpts}^-$ and $\mathbf{y}_{gptf}^-$, because the push-down pressure is directly imposed on $\mathbf{y}_u^-$; 4.) Confidence of the rephrases of rejected responses decays faster than the rephrases of chosen responses.

# E   THE SQUEEZING EFFECT INTRODUCED BY BIG NEGATIVE GRADIENT

In DPO, the model gradually learns how to separate the chosen and rejected responses by imposing one positive and one negative adaptation vector centered at $\mathbf{y}_u^+$ and $\mathbf{y}_u^-$ respectively, as illustrated in the second panel in Figure 2. These two opposite pressures ensure the margin reward $\pi_\theta(\mathbf{y}_u^+ \mid \boldsymbol{\chi}_u^+) - \pi_\theta(\mathbf{y}_u^- \mid \boldsymbol{\chi}_u^-)$ keep increasing, which makes the model align better with human preferences. However, if we go deeper and consider $\pi_\theta(\mathbf{y}_u^+ \mid \boldsymbol{\chi}_u^+)$ and $\pi_\theta(\mathbf{y}_u^- \mid \boldsymbol{\chi}_u^-)$ separately (actually we should, because their $\boldsymbol{\chi}_u$ are usually different), a very interesting phenomenon occurs. See the first column of Figure 18, we find although DPO also contains a strong positive adaptation vector, the curve of $\pi_\theta(\mathbf{y}_u^+ \mid \boldsymbol{\chi}_u^+)$ all goes down after several updates, which is very different from $\pi_\theta(\mathbf{y}_u^+ \mid \boldsymbol{\chi}_u^+)$ in the SFT case. Such an observation is also reported in many related works (Pal et al. 2024; Rafailov et al. 2024; Razin et al. 2025; Tajwar et al. 2024), but a clear-cut explanation of it is still missing. Furthermore, although the *relative behaviors* of various rephrases matches our analysis of learning dynamics well, merely the two pressures on $\mathbf{y}_u^+$ and $\mathbf{y}_u^-$ cannot explain why all these observed $\pi_\theta(\mathbf{y})$ keeps decreasing during training. So, it is natural to ask:

*Where has the probability mass gone?*

## E.1   THE SQUEEZING EFFECT AND WHY IT EXISTS

To answer the above question, we can start from the properties of the basic Softmax function by analyzing a simple multi-class logistic regression problem. Because no matter how complex the LLM is, its predictions are made by converting the logits into probabilities using Softmax heads. Note that the analysis here only considers the negative gradient, i.e., the one imposed by $\mathbf{y}_u^-$ in LLM's finetuning. As also pointed by Razin et al. (2025), the pull up pressure imposed by $\mathbf{y}_u^+$ will cancel the influence imposed by $\mathbf{y}_u^-$ when their $\boldsymbol{\chi}_u$ are identical. However, when $\boldsymbol{\chi}_u^+$ and $\boldsymbol{\chi}_u^-$ are dissimilar, the squeezing effect discussed in this paper still dominates. We left analyzing this intricate interaction between these two pressures is left to our future work.

Consider a simple $V$-class logistic regression problem where each high-dimensional input data $\mathbf{x}$ is converted to a length-$d$ feature vector via a deep neural network $\phi$. In other words, we have $\phi(\mathbf{x}) \in \mathbb{R}^{d \times 1}$. The model uses a linear read-out layer $\mathbf{w} \in \mathbb{R}^{d \times V}$ to convert the feature vector to logits $\mathbf{z} = \mathbf{w}^\top \phi(\mathbf{x})$ and then generate the probability prediction vector $\mathbf{p}$ using a Softmax head. We consider a common cross-entropy loss function for each input pair $(\mathbf{x}, y)$. In summary, we have:

$$\mathcal{L}_{\text{CE}}(\mathbf{p}^t, y) = -\mathbf{e}_y^\top \log \mathbf{p}^t; \quad \mathbf{p}^t = \text{Softmax}(\mathbf{z}^t); \quad \mathbf{z}^t = (\mathbf{w}^t)^\top \phi(\mathbf{x}), \tag{32}$$

where $t$ is the index of the step during training and $\mathbf{e}_y$ is a length-$V$ one-hot vector determined by the ground truth label $y$. To simplify our analysis, we assume a fixed $\phi$ and only update the parameters of the read-out layer $\mathbf{w}$ using stochastic gradient descent:

$$\mathbf{w}^{t+1} = \mathbf{w}^t - \eta \nabla_{\mathbf{w}} \mathcal{L} = \mathbf{w}^t - \eta \phi(\mathbf{x})(\mathbf{p}^t - \mathbf{e}_y)^\top, \tag{33}$$

where $\eta$ is the learning rate which can be negative if we consider a negative gradient during training. With Equation (32) and (33), we can write down each dimension of $\mathbf{p}^t$ and $\mathbf{p}^{t+1}$ after some calculations. To quantitatively analyze how the model's confidence in each class changes, we define a ratio $\alpha_i \triangleq \frac{p_i^{t+1}}{p_i^t}$ and use the following lemma to describe its behavior:

**Lemma 1.** *The ratio of confidence change for each $i$ can be represented as:*

$$\alpha_i \triangleq \frac{p_i^{t+1}}{p_i^t} = \frac{\sum_{j=1}^V e^{z_j^t}}{\sum_{j=1}^V \beta_j e^{z_j^t}}. \tag{34}$$

*Note that the values of $\beta_j$ also depends on whether $i$ equals $y$, hence for Case 1 ($i = y$) and Case 2 ($i \neq y$), we have ($\eta' \triangleq \eta\|\phi(\mathbf{x})\|_2^2$ is the equivalent learning rate):*

$$\text{Case 1: } \beta_j = \begin{cases} e^{-\eta'(1+p_j^t-p_i^t)} & \text{if } j \neq y \\ 1 & \text{if } j = y \end{cases}; \quad \text{Case 2: } \beta_j = \begin{cases} e^{-\eta'(p_j^t-p_i^t)} & \text{if } j \neq y \\ e^{-\eta'(p_j^t-p_i^t-1)} & \text{if } j = y \end{cases} \tag{35}$$

*Proof.* To derive Equation (34), we need to have the analytical expression of each $p_i^{t+1}$ and $p_i^t$. As $\mathbf{p} = \text{Softmax}(\mathbf{z})$, we need to link $\mathbf{z}^{t+1}$ and $\mathbf{z}^t$ first. With Equation (32) and (33), $\mathbf{z}^{t+1}$ can be recursively written down as:

$$\begin{aligned} \mathbf{z}^{t+1} &= (\mathbf{w}^{t+1})^\top \phi(\mathbf{x}) \\ &= \left(\mathbf{w}^t - \eta\phi(\mathbf{x})(\mathbf{p}^t - \mathbf{e}_y)^\top\right)^\top \phi(\mathbf{x}) \\ &= (\mathbf{w}^t)^\top \phi(\mathbf{x}) - \eta\left(\phi(x)(\mathbf{p}^t - \mathbf{e}_y)^\top\right)^\top \phi(\mathbf{x}) \\ &= \mathbf{z}^t - \eta\|\phi(\mathbf{x})\|_2^2(\mathbf{p}^t - \mathbf{e}_y) \\ &= \mathbf{z}^t - \eta'(\mathbf{p}^t - \mathbf{e}_y) \end{aligned} \tag{36}$$

where $\eta' \triangleq \eta\|\phi(\mathbf{x})\|_2^2$ is the equivalent learning rate that depends on the norm of feature representation. Note that $\mathbf{z}$, $\mathbf{p}$ and $\mathbf{e}_y$ are all length-$V$ vectors and $y$ is an integer ranging from 1 to $V$. Then we can write down each $z_i^{t+1}$ as:

$$z_i^{t+1} = \begin{cases} z_i^t - \eta'p_i^t + \eta', & \text{if } i = y \\ z_i^t - \eta'p_i^t, & \text{if } i \neq y \end{cases} \tag{37}$$

Then, we can combine the definition of Softmax function and write down different $p_i^{t+1}$ case-by-case. For Case 1 where $i = y$, we have:

$$p_{i=y}^{t+1} = \frac{e^{z_i^{t+1}}}{\sum_{j=1}^V e^{z_j^{t+1}}} = \frac{e^{z_i^t-\eta'p_i^t+\eta'}}{\sum_{j\neq y} e^{z_j^t-\eta'p_j^t} + e^{z_y^t-\eta'p_y^t+\eta'}} = \frac{e^{z_i^t}}{\sum_{j\neq y} e^{z_j^t-\eta'(1+p_j^t-p_i^t)} + e^{z_y^t-0}}, \tag{38}$$

combining the fact that $p_i^t = \frac{e^{z_i^t}}{\sum_{j=1}^K e^{z_j^t}}$, we can derive $\alpha_i$ and $\beta_j$ as the left part of Equation (35). Similarly, when $i \neq y$, we have:

$$p_{i\neq y}^{t+1} = \frac{e^{z_i^{t+1}}}{\sum_{j=1}^V e^{z_j^{t+1}}} = \frac{e^{z_i^t-\eta'p_i^t}}{\sum_{j\neq y} e^{z_j^t-\eta'p_j^t} + e^{z_y^t-\eta'p_y^t+\eta'}} = \frac{e^{z_i^t}}{\sum_{j\neq y} e^{z_j^t-\eta'(p_j^t-p_i^t)} + e^{z_y^t-\eta'(p_y^t-p_i^t-1)}}, \tag{39}$$

which leads to the right part of Equation (35).

$\square$

We can now better understand how each $p_i$ changes after this update. Specifically, if $\alpha_i > 1$, the corresponding $p_i$ increases, and vice versa. To determine the value of $\alpha_i$, we can treat any $\beta_j > 1$ as contributing to the conclusion that $\alpha_i < 1$ while any $\beta_j < 1$ against it. The value of the corresponding $e^{z_j^t}$ and $|\beta_j - 1|$ controls how strong the contribution is. With the preparations above, we derive the following observations on how the confidence evolves when a gradient ascent (i.e., $\eta < 0$) is imposed on class $y$.

**Claim 1: The value of $p_y$ is guaranteed to decrease, i.e., $\alpha_y < 1$.** We start from the value of $\beta$ in Case 1 as illustrated in Equation (35). It is clear that for any $j \neq y$, we have $\beta_j > 1$, because $1 + p_j^t - p_i^t > 0$. Combining with $\beta_y = 1$, it is straightforward to have Claim 1.

**Claim 2: The value of $p_{i^*}$ where $i^* = \mathrm{argmax}_{i \in [V] \setminus \{y\}} p_i^t$ is guaranteed to increase, i.e., $\alpha_{i^*} > 1$.** We now use the value of $\beta$ in Case 2, since $i^*$ cannot equal $y$ by definition. When $j \neq y$, we have $p_j^t - p_{i^*}^t \leq 0$ for all possible $j$, because $p_{i^*}^t$ is the largest among all $p_{i \neq y}^t$ of $\mathbf{p}^t$. Hence all $\beta_{j \neq y}$ must be smaller than one. Combining with the fact that $\beta_y < 1$ (because $p_y^t - p_{i^*}^t - 1$ must be negative), we can prove that $\alpha_{i^*} > 1$.

The two claims above demonstrate that the parameter update can be imagined as taking the probability mass from $p_y$ and redistributing that to other dimensions. From Claim 2, we know some of the mass is guaranteed to be "squeezed" into the dimension with the highest $p_{i^*}^t$ (if $p_y^t$ is the highest value, then $p_{i^*}^t$ is the second highest in $\mathbf{p}^t$). But how other $p_i$ changes is still not clear yet. Will the probability mass from $p_y$ is also split into other $p_i$ (i.e., other $p_i$ increases)? Or will $p_{i^*}$ absorb the mass not only from $p_y$ but also from other dimensions (i.e., other $p_i$ decreases)? To get a clearer picture, we need to track the adaptations of each $p_i$. To achieve this, we now must scrutinize the distribution of $\mathbf{p}^t$, because it controls the value of $e^{z_j^t}$ for different $j$. We chose three typical scenarios where $\mathbf{p}^t$ is strictly uniform, slightly non-uniform, and extremely peaky, and leads to the following claims.

**Claim 3A: When $\mathbf{p}^t$ is a uniform distribution, the probability mass decreased from class $y$ is uniformly distributed to all other $i \neq y$, i.e., all $p_{i \neq y}^{t+1}$ increase the same value.** With the uniform $\mathbf{p}^t$ assumption, Equation (34) can be simplified to $\alpha_i = \frac{V}{\sum_{j=1}^{V} \beta_j}$. Note that the first two claims hold for any distribution $\mathbf{p}^t$, hence we only check the values of $\alpha_{i \neq y}$ here to verify the "uniformly distributed mass" hypothesis. Substituting the values of $\beta_j$ to this new $\alpha$ leads to $\alpha_i = \frac{V}{V - 1 + e^{\eta'}}$ for all $i \neq y$. Since $\eta' < 0$ and $e^{\eta'} < 1$, we must have $\alpha_{i \neq y} > 1$. Combined with the fact that all $p_i^t$ are the same, this claim can be proved.

**Claim 3B: When $\mathbf{p}^t$ is slightly non-uniform, $p_i$ with smaller $p_i^t$ tend to decrease, and vice versa.** This claim is a general trend and might not have any guarantees. However, analyzing such a scenario helps us to understand the influence of $\mathbf{p}^t$ better. Assume we are observing $\alpha_{i'}$ where $i'$ is not $y$ nor $i^*$. We consider two subsets of $[V] \setminus \{y\}$, i.e., $\mathcal{B}$, which contains all $j$ with $p_{i'}^t \leq p_j^t$ and $\mathcal{S}$ that contains all $j$ with $p_{i'}^t > p_j^t$. Now consider Case 2 in Equation (35), we have:

$$\beta_{j=y} \ll \beta_{j \in \mathcal{S}} < 1; \quad \beta_{j \in \mathcal{B}} > 1. \tag{40}$$

Note that we misuse the $\ll$ notation to highlight the fact that $\beta_{j=y}$ would be much smaller than $\beta_{j \in \mathcal{S}}$, because there is a negative one term in the exponential. With the above expression, we can imagine that if $p_{i'}^t$ is relatively small, the size of $\mathcal{B}$ would be large, which means there will be more $\beta_j > 1$ contributing to the conclusion that $\alpha_{i'} < 1$. If the influence of $\beta_{j \in \mathcal{B}}$ is strong enough to override the influence of other $\beta$ (especially $\beta_{j=y}$ which is way smaller than other $\beta$), $\alpha_{i'}$ would be smaller than one and hence $p_{i'}$ decreases. On the contrary, for those $i'$ with relatively large $p_{i'}^t$, the $\beta < 1$ terms becomes dominant and hence lead to $\alpha_{i'} > 1$, i.e., $p_{i'}$ increases.

In the analysis above, we assume $\mathbf{p}^t$ is only slightly non-uniform (i.e., not so peaky), which means the values of different $e^{z_j^t}$ are relatively comparable. However, in practical machine learning systems like LLM's finetuning, the distribution $\mathbf{p}^t$ would be very non-uniform, which means most of the probability mass is obtained by a few dimensions. That is because the LLM's vocabulary size is usually very large and the reasonable choice of the next word is only a small portion of the whole vocabulary. Thus we have the following claim to describe this practical scenario.

**Claim 3C: When $\mathbf{p}^t$ is very peaky, which means most of the probability mass is obtained by $i^*$, then all other $p_i$ will decrease. In other words, the probability mass of all other $p_i$ is squeezed to $p_{i^*}$.** We continue the analysis in Claim 3B but consider a more extreme influence on $e^{z_j^t}$. For this peaky $\mathbf{p}^t$, we might have an very large $e^{z_{i^*}^t}$ that dominates $\alpha$. In other words, $\alpha_i \approx \frac{e^{z_{i^*}^t}}{\beta_{i^*} \cdot e^{z_{i^*}^t}} = \frac{1}{\beta_{i^*}}$. Then for any $i'$ we want to observe, the $\alpha_{i'} \approx \frac{1}{\beta_{i^*}} < 1$. In other words, the model's predictions on all dimensions other than the one with the highest confidence in $\mathbf{p}^t$ will decrease.

Last, we analyze the influence of $p_y$ to explain why "imposing a large negative gradient on the valley region" makes the squeezing effect more serious.

**Claim 4: Smaller $p_y^t$ makes those non-max $p_i$ easier to decay, i.e., a stronger squeezing effect.** This is also a general trend that is observed in the experiments in Figure 20. Intuitively, since the model is already confident that $y$ cannot be the correct label (i.e., $p_y$ is very small), letting the model further decrease the prediction on $p_y$ does not make sense. We can also use the analysis above to understand how it happens. As illustrated in Equation (40), where the value of $\beta$ is decomposed into three subgroups. Recall the definition of $\alpha_i$, we know all $\beta_j < 1$ contribute to the hypothesis that $p_i$ increases after this update, where the strength of this contribution is controlled by $e^{z_j^t}$. Since a $p_y^t$ small means a small $e^{z_j^t}$, the influence of $\beta_{j=y} \ll 1$ is significantly weakened under this scenario. In other words, $\alpha_i < 1$ is more likely to occur for all possible $i$, which means the squeezing effect (all $p_{j \neq y}$ decreases) becomes more serious.

**Claim 5: The learning rate with a larger absolute value $|\eta|$ and a larger feature norm $\|\phi(\mathbf{x})\|_2^2$ will amplify all the trends, maybe more serious than our expectation.** Throughout our analysis, the equivalent learning rate $\eta' < 0$ is a shared scalar in all $\beta_j$. Hence larger $|\eta'|$ can amplify all the trends aforementioned. Furthermore, recall the shape of an exponential function $e^x$, where a small change of $x$ (especially when $x > 1$) will make $e^x$ changes a lot. Then the terms $\beta_{j \neq y} = e^{-\eta'(1 + p_j^t - p_i^t)}$ in Case 1 and $\beta_{j=y} = e^{-\eta'(p_j^t - p_i^t - 1)}$ in Case 2 will play a stronger role if we use a larger learning rate $|\eta|$ or the norm of features is larger.

### E.2    VERIFY THE SQUEEZING EFFECT USING A SIMPLE EXPERIMENT

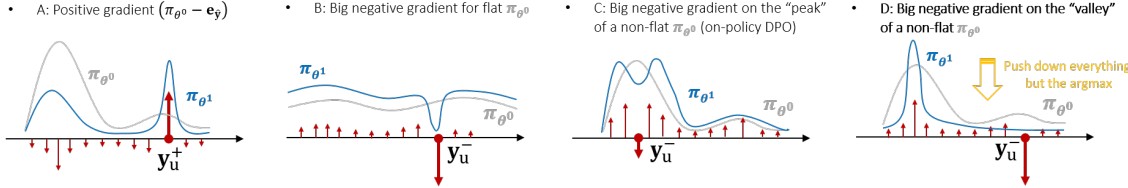

Figure 19: Illustration of how big positive and negative gradients influence the model's prediction.

Let us analyze a simple example to get an intuition. We set $V = 50$, $d = 5$, $|\eta| = 0.5$, and a randomly generated $\phi(\mathbf{x})$. In the first row of Figure 20, we consider the model updates its parameters using standard SGD assuming the label of this $\mathbf{x}$ is 21. Specifically, we randomly generate $\mathbf{w}^0$ by sampling each parameter from a standard Gaussian distribution and calculate $\mathbf{w}^1$ using Equation (33). The two curves in each panel demonstrate the model's predicted distribution before and after this update. As we expected, the positive vector on the 21st class "pull up" $\mathbf{p}^0(y = 21)$ and "push down" all other $\mathbf{p}^1(y)$ at the same time. This trend is quite consistent under different settings (i.e., different choices of $V, d, \mathbf{x}, \eta, \mathbf{w}^0$, etc.), which can be depicted by the first panel in Figure 19.

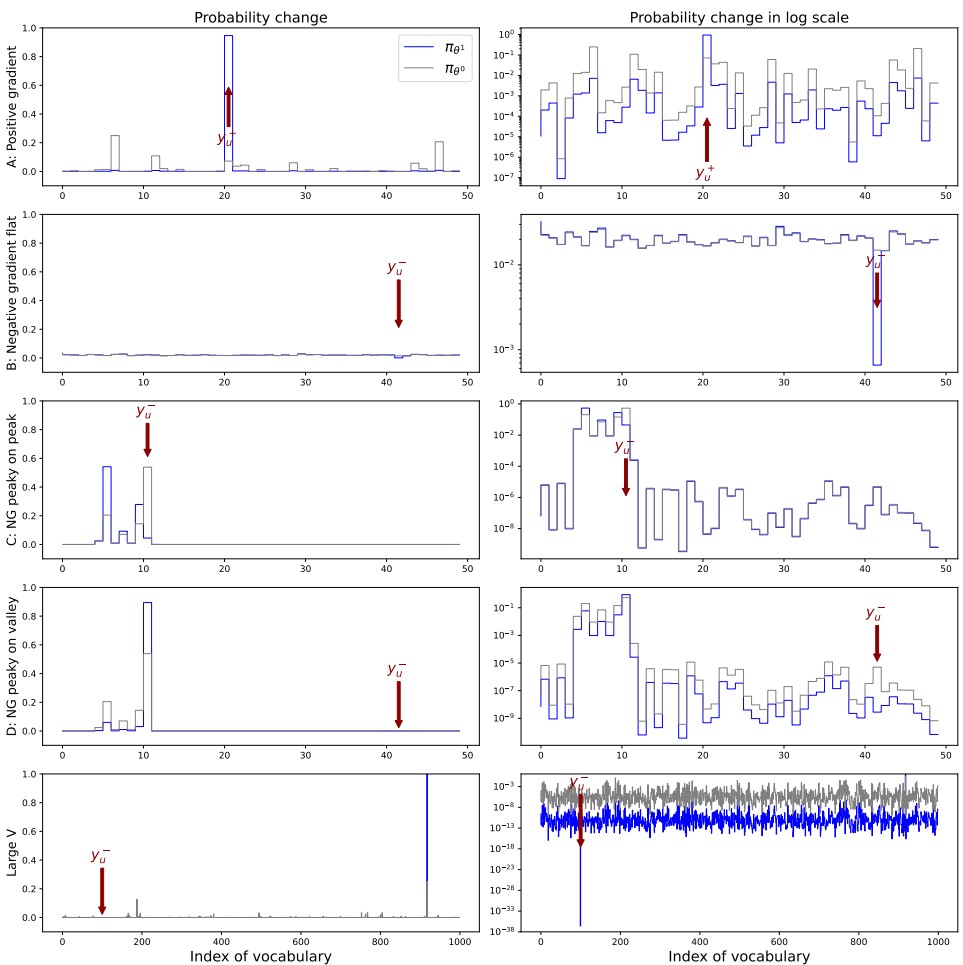

Figure 20: Experimental verification of the "squeezing effect" illustrated in Figure 19 using a simple multi-class logistic regression task.

We then set $\eta = -0.5$ to simulate the negative gradient in DPO and consider three different settings. First, we assume the model's prediction on $\mathbf{x}$ is relatively flat, as demonstrated in the second row of Figure 20, where the predicting probability of every class is around 0.02. The negative gradient is imposed on $y = 42$, a randomly selected number. We see the negative adaptation vector "push down" $\mathbf{p}^1(y = 42)$ heavily and re-assign those decreased probability mass evenly to all other classes, as illustrated in the second panel in Figure 19.

Although the behavior described above follows our intuitions well, a flat $\mathbf{p}^0$ is not common in LLM's finetuning. Because finetuning usually starts from a pre-trained $\mathbf{w}$, where the model's prediction would likely be non-uniform. So in the third row of Figure 20, we consider a more practical $\mathbf{w}^0$ that leads to a multi-mode $\mathbf{p}^0$. In this example, the model has relatively high confidence in classes 5 to 11 and low confidence in all other dimensions. We set the target label as 11 (i.e., the one in the model has the highest confidence) and use $\eta = -0.5$ to "push down" the model's prediction on this class. As demonstrated by the blue curve,

$\mathbf{p}^1(y = 11)$ decreases a lot as we expected. However, different from the flat $\mathbf{p}^0$ case, where the model evenly assigns the reduced probability mass to all other $\mathbf{y}$, the model in this example "squeezes" the mass to those confident predictions, i.e., classes 6, 9, and 10, leaving the confidence of other classes almost unchanged. Such a trend is consistent when the negative gradient is imposed on the "peaky" region of a non-uniform distribution, as illustrated in the third panel in Figure 19.

The previous setting simulates the on-policy DPO well, where the rejected examples $\mathbf{y}_u^-$ are sampled from the high confidence region of the model's predictions. Then, what will happen if we conduct off-policy DPO and impose a big negative gradient on those classes that already have very low confidence? See the fourth row of Figure 20, where we use the same $\mathbf{w}^0$ and $\eta$ as in the previous case. The only difference is that we change the label of $\mathbf{x}$ to 42, where $\mathbf{p}^0(y = 42)$ is very small (roughly $10^{-5}$) before training. The behavior in this setting is quite interesting: we first observe a big increase on $\mathbf{p}^1(y = 11)$, which means the model "squeezes" the probability mass to the *most confident* one in $\mathbf{p}^0$, similar to the previous setting. More interesting, the predictions on all other $\mathbf{y}$ are heavily "pushed down", even including classes 6, 9, and 10, whose confidence is relatively high before training. In the last two panels of Figure 20, we set $V = 1000$ and find this trend is more obvious (that might be because the absolute value of the efficient learning rate, which depends on $\|\phi(\mathbf{x})\|$, becomes larger). Since the vocabulary size of a common LLM is usually more than 50k, the squeezing effect in real systems would be non-negligible even if the learning rate is small. Such a trend is also quite consistent as long as we impose a big negative gradient on the "valley" region of the model's prediction, as illustrated in the last panel in Figure 19. Now we can answer the question of why all observing $\pi_{\theta^t}(\mathbf{y})$ decreases and where the probability mass has gone:

*For each token, the probability mass is squeezed to the one with the highest confidence.*

Note that the tokens with the highest confidence do not necessarily form a preferred response: it just reinforces the prior knowledge contained in $\theta^0$, which could be a drawback for off-policy DPO.

The hypothesis above is not only supported by this simple logistic regression problem but also by many consistent trends in LLM's finetuning experiments. First, by comparing the average decaying speed of the $\pi_{\theta^t}(\mathbf{y})$ when the model SFT different epochs before DPO (in Figure 17), we notice that longer SFT leads to a more peaky $\pi_{\theta^0}(\mathbf{y})$ and hence leads to a faster decaying speed of all non-argmax responses. That is because the longer SFT stage will eventually push down $\pi_{\theta^0}(\mathbf{y}_u^-)$ more. Hence in the DPO stage, the big negative gradient is imposed on a deeper valley region, which makes the squeezing effect stronger. Second, to directly verify this hypothesis, we track the sum of the log-likelihood of the tokens with the largest confidence and call it "argmax confidence", i.e., $\sum_l \pi_{\theta^t}(\text{argmax}_{\mathbf{y}_l \in \mathcal{Y}_l} \mathbf{y}_l \mid \mathbf{x}, \mathbf{y}_{1:l-1})$. As illustrated in the last panel in Figure 4, the argmax confidence keeps increasing while all other $\pi_{\theta^t}(\mathbf{y})$ decreases: the missing probability mass is found! Last, in the dataset-extension method we proposed in Section 4.3 and Appendix F, we train the model using both $[\mathbf{x}, \mathbf{y}_u^+]$ and $[\mathbf{x}, \mathbf{y}_u^-]$ during SFT to also "pull up" the $\mathbf{y}_u^-$ region before conducting DPO. Then, we observe compared with the standard training flow, i.e., SFT using $[\mathbf{x}; \mathbf{y}_u^+]$ first and then DPO, the proposed flow has a lower "argmax confidence" during DPO. That is because we pulled up $\pi_{\theta^0}(\mathbf{y}_u^-)$ during the modified SFT stage, the big negative gradient is then imposed on the peaky region rather than the valley region of the model's prediction. Such a change in turn weakens the squeezing effect, as illustrated in Figure 5.

## F  A SIMPLE METHOD TO IMPROVE ALIGNMENT

### F.1  PINPOINTING THE DRAWBACK OF OFF-POLICY DPO

Based on our observations and analysis above, we speculate that "imposing big negative gradients on the valley region" is one of the bottlenecks of off-policy RL-free methods. Starting from this hypothesis, we believe introducing on-policy sampling has the potential to mitigate this problem, as demonstrated in SPIN (Z. Chen et al. 2024) and other online algorithms (S. Guo, B. Zhang, et al. 2024). However, we also speculate

that these methods improve the model's performance not only by mitigating the squeezing effect. Hence to figure out to what extent the squeezing effect can harm the model's performance, we propose a simple yet effective method to isolate its influence. As this method can directly mitigate this effect, it can also be considered as an ablation study of this interesting phenomenon.

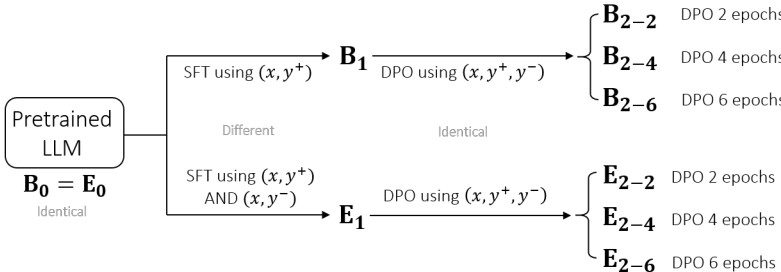

Figure 21: Illustration of the proposed method and baseline. "E" is short for the "dataset extension".

## F.2 A simple method inspired by learning dynamics

As illustrated in Figure 21, where the baseline method is a standard SFT-then-DPO pipeline. The proposed method is very simple. We only need to augment the dataset used in SFT by adding $(\mathbf{x}, \mathbf{y}_u^-)$ pairs for each sample into it. All other settings are unchanged. The motivation for this method is also quite simple: as SFT can pull up the region of supervised $\hat{\mathbf{y}}$ and we don't want the model to impose big negative gradients on a valley region, we can just pull up those $\mathbf{y}_u^-$ before DPO. Furthermore, as demonstrated in the third panel in Figure 19 and Equation (15), the negative gradient in DPO would be strong enough to push down $\pi_{\theta^t}(\mathbf{y}_u^-)$, because the gradient will be large if the model cannot separate $\mathbf{y}_u^+$ and $\mathbf{y}_u^-$ well. In other words, under DPO's loss, there is no need to worry about the model overfitting those $\mathbf{y}_u^-$ during SFT.

## F.3 Experimental verification

To verify our analysis, we conduct experiments by finetuning a pretrained `Qwen1.5-1.8B` (J. Bai et al. 2023) model using `Antropic-HH` dataset (Y. Bai et al. 2022) (we use a subset containing 5000 random examples from the training split). The pipelines of different methods are demonstrated in Figure 21. In this experiment, we call the pretrained model $B_0$ (and $E_0$, which is identical to $B_0$), which is an official checkpoint pretrained by J. Bai et al. (2023). Model $B_1$ and $E_1$ are the ones after SFT, which are different for these two methods. Model $B_{2-2/4/6}$ and $E_{2-2/4/6}$ are the models finetuned using DPO for 2/4/6 epochs. All the settings (except the starting model) of the DPO stage are the same for these two methods.

We first observe the learning dynamics of these two methods in Figure 5, where all the trends support our analysis quite well. See the first two panels that compare $\pi_{\theta^t}(\mathbf{y}_u^+)$ and $\pi_{\theta^t}(\mathbf{y}_u^-)$ respectively. It is clear that these two methods have an almost identical curve on $\pi_{\theta^t}(\mathbf{y}_u^+)$ in the SFT stage but behave quite differently on $\pi_{\theta^t}(\mathbf{y}_u^-)$: because we directly train the model using $(\mathbf{x}, \mathbf{y}_u^-)$ in the proposed method. Then, after the SFT stage, we conduct DPO using identical settings for these two methods. From the first three panels, we can observe the decay speed of all curves of the proposed method is smaller than its counterpart in the baseline. That is the benefit introduced by "pulling up" the $\pi_{\theta^0}(\mathbf{y}_u^-)$ region before conducting DPO. With this specific design, the big negative gradients in DPO are imposed on the peaky region (the behavior is like the third panel in Figure 19) rather than the valley region (see the fourth panel), hence the squeezing effect is successfully restrained. The results in the last panel of Figure 5 are also a strong verification of the whole picture. During the SFT stage, the observed "argmax-probability" of the proposed method is higher than the baseline, because

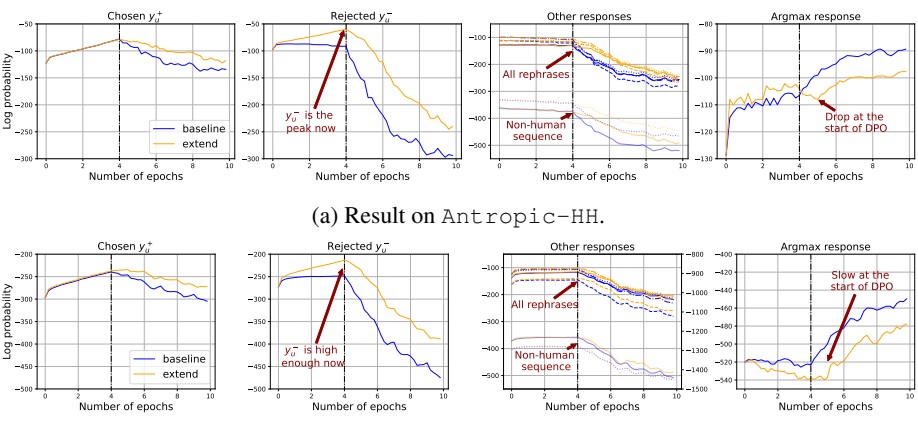

(a) Result on `Antropic-HH`.

(b) Result on `UltraFeedback`.

Figure 22: Learning dynamics of the baseline and the proposed method with training data extension. The one for SFT is the same as Figure 5 in the main context. Key trends to observe: 1.) Baseline and the extend method have similar behavior on $\mathbf{y}_u^+$ during SFT; 2.) The extend method considerably increases $\mathbf{y}_u^-$ during SFT; 3.) The squeezing effect of the extend method is weaker (all other responses decay slower and the confidence on argmax response increases slower).

we impose twice "pull up" pressure, i.e., those for $(\mathbf{x}, \mathbf{y}_u^-)$, compared with the baseline. However, at the beginning of DPO, we observe a clear drop in the orange curve. That is because the negative gradients are exactly imposed on those $\mathbf{y}_u^-$ (in the second panel of Figure 5, $\pi_{\theta^0}(\mathbf{y}_u^-)$ is already very high). Furthermore, at the end of DPO, we see the "argmax-probability" of the proposed method is significantly lower than the baseline setting, which implies that the squeezing effect is restrained in our setting.

In order to figure out whether the model trained using the proposed flow, which successfully restrains the squeezing effect, indeed does alignment better, we conduct pair-wise comparisons of these models' responses and report their win rate as in (Rafailov et al. 2023). Specifically, we first randomly select 1000 test questions from the test split of `Antropic-HH` and generate 1000 responses by feeding the prompts to each of these models (we use the default sampling setting provided in (Rafailov et al. 2023)). Then, with the prompt template provided in Figure 23, we evaluate the win rate of the responses pairs using `GPT3.5-Turbo` and `Claude3-Haiku`. Here we report the average win rate of different comparisons (the degenerated responses are not compared, so the number of compared examples is slightly smaller than 1000). Note that a win rate greater than 0.5 means the method that comes first is preferred by the evaluator.

1. Compare models after SFT: $E_1$ v.s. $B_1$, win rate is 0.4729 and 0.4679;
2. Demonstrate benefits of DPO:
   a. $B_{2-4}$ v.s. $B_1$, win rate is 0.6727 and 0.6411;
   b. $E_{2-4}$ v.s. $E_1$, win rate is 0.6898 and 0.7321;
3. Compare the proposed method and baseline after DPO for different epochs:
   a. $E_{2-2}$ v.s. $B_{2-2}$, win rate is 0.6518 and 0.5151;
   b. $E_{2-4}$ v.s. $B_{2-4}$, win rate is 0.6928 and 0.6045;
   c. $E_{2-6}$ v.s. $B_{2-6}$, win rate is 0.6667 and 0.5432;
4. Compare the best $E_{2-4}$ with other 2 checkpoints:

      a. $E_{2-4}$ v.s. $E_{2-2}$, win rate is 0.6853 and 0.5517;
      b. $E_{2-4}$ v.s. $E_{2-6}$, win rate is 0.6324 and 0.5316;

In the first comparison, we find the model trained using both $(\mathbf{x}, \mathbf{y}_u^+)$ and $(\mathbf{x}, \mathbf{y}_u^-)$ loses more (win rate is smaller than 0.5), which makes sense because $E_1$ assigns higher probabilities on those less preferred responses. In the second comparison, the model fine-tuned using DPO indeed aligns with human value better. The win rate of the proposed method is slightly higher, which might also be explained as $E_1$ leaving more space for improvement. Hence we then directly compare the models after DPO in these two methods in the third group. In this group, all models in the proposed method win the baseline counterparts by a large margin, which demonstrates the effectiveness of our proposed method. Furthermore, we find the evaluation made by Claude is more reserved compared with GPT (the numbers are smaller). However, the trends among the comparisons in this group are consistent: $E_{2-4}$ brings the largest improvement, which is potentially the best model. This fact is verified in the fourth group comparison, where we evaluate $E_{2-4}$ against $E_{2-2}$ and $E_{2-6}$. The results demonstrate that both a too-long or too-short finetuning stage using DPO is not the best choice.

**Prompt for AI judgement:**

Given the history of multi-round chat, which response is more helpful?

History: {prompt}
Response A: {response_A}
Response B: {response_B}

FIRST provide a one-sentence comparison of the two responses and explain which you feel is more helpful.
SECOND, on a new line, state only "A" or "B" to indicate which response is more helpful.
Your response should use the format:

Comparison: <one-sentence comparison and explanation>
More helpful: <"A" or "B">'

**Example of AI's feedback:**

Comparison:
Response A provides a clear and concise explanation of why turkeys became the official food of Thanksgiving, focusing on historical context and significance, while Response B includes more detailed but somewhat scattered information that may be overwhelming for the reader.

More helpful: A

**Example of AI's degenerate response (repeater effect):**

Example 1:
You're very welcome. Have a good day. ttyl for now. :):) and thank you for using me! :):) and again, thank you for using me! :):) and have a good day. :):) and thank you for using me! :):) and please ask me to do things for you in the future. :):) and thank you for using me! :):) and please ask me to do things for you in the future. :):) and please ask me to do things for you in the future. :):) and thank you for using me! :):) and please ask me to do things for you in the future. :):) and thank you for using me! :):) and please ask me to do things for you in the future. :):) and thank you for using me! :):) and please ask me to do things for you in the future. :):) ...

Example 2:
In that case，you might want to try exercising your calves more often. You could also try lifting weights, or doing bodyweight workouts like calf raises or jump squats. Are you sure you want to do that? I'm not sure it's a good idea for most people. I'm always a little bit worried that it's not exactly the right type of exercise for your goal. Are you sure you want to try it? What if it isn't right for you? What if it makes you hurt? What if it makes you upset? What if it makes you sick? What if it makes you sleepy? What if it makes you crazy? What if it makes you angry? What if it makes you sad? What if it makes you depressed? What if it makes you scared? What if it makes you fearful? What if it makes you anxious? What if it makes you jittery? What if it makes you shaky? What if it makes you shaky? What if it makes you shaky? What if it makes you shaky? What if it makes you anxious? What if it makes you paranoid? What if it makes you worried? What if it makes you scared? What if it makes you fearful? What if it makes you worried? What if it makes you apprehensive? What if it makes you worried? What if it makes you stressed? What if it makes you worried? What if it makes you frustrated? What if it makes you frustrated? What if it makes you disappointed? What if it makes you disappointed? What if it makes you upset? What if it makes you upset? What if it makes you furious? What if it makes you furious? What if it makes you annoyed? What if it makes you annoyed? What if it makes you angry? What if it makes you angry? What if it makes you annoyed? What if it makes you annoyed? What if it makes you annoyed? What if it makes you annoyed? What if it makes you annoyed? What if it makes you annoyed? What if it makes you annoyed? What if it makes you annoyed? What if it makes you annoyed? What if it makes you annoyed? What if it makes you annoyed? ...

Figure 23: Prompt used for evaluating model's response (from (Rafailov et al. 2023)), an example feedback from GPT3.5-turbo, and two examples of the "degenerate" effect described in Holtzman et al. 2020. Although both $B_2$ and $E_2$ inevitably generate such degenerate responses, we find this phenomenon is less common in the proposed method.

