# OpenReview forum: "Learning Dynamics of LLM Finetuning"
_ICLR.cc/2025/Conference — ICLR 2025 Oral_

### Official Review · Reviewer_W188 · 2024-10-28

**Soundness:** 3
**Presentation:** 3
**Contribution:** 2
**Rating:** 6
**Confidence:** 3

**Summary:**

This paper analyzed the learning dynamics of LLM fine-tuning, in particular, the change of the loss function for other examples delta f(x_o) during training time parameter updates (from gradient descent with respect to input examples x_i), for both SFT and DPO. For SFT, the research used MNIST classification as an illustrative example to indicate that if x_o is similar to x_u but classified differently, the predicted probability of the x_u's class for x_o also got pulled up. For DPO, the research found that log likelihoods of both the positive examples y_u^+ and negative examples y_u^- all decreased, and the only example with log likelihood increase is the greedy decoding result y_u^*. This phenomenon could potentially explain the increased repetitiveness of DPO generated results towards the end of fine-tuning, and the increase of halluncinations during SFT. The researcher proposed a method to mitigate this effect by first SFT-ing on both positive examples and negative examples, and then perform DPO.

**Strengths:**

1. Analyzed the theoretical process behind DPO gradient updates. Explained deeply how the effect of gradient updates with respect to the training examples affect the model's performance on unrelated examples. Provided a theoretical framework for (potentially) understanding the origin of hallucinations.
2. Explained a common problem in DPO that was not illustrated well in prior work. For example, Rafael et.al. 2024 only vaguely claimed that "DPO decreases the likelihood of all data in favor of extrapolated responses", but this paper clearly indicated that the decreased likelihood all (or mostly) added to the greedy decoding sample (y^*).
3. The paper is well written and easy to follow.

**Weaknesses:**

Disclaimer: Only checked the main text and the Appendix A. Did not check Appendices B,C,D.

1. Need a "related work" section. The paper in its current form does not explain the relations between "learning dynamics" with related literature, for example, preventing negative transfers during transfer learning, neural tangent kernels, or loss landscape that enabled adversarial attacks. Offering a "related work" section whereas reducing the theoretical focus a little bit would help ICLR readers better position your work.

2. Need to indicate assumptions in your prop 1 and section 3.2 clearer. As far as I could see, there are at least 2 assumptions:
- Higher order terms has little to no impact to learning dynamics
- K changes slowly during the initial training stages and thus has a limited impact to training dynamics

Please list out these assumptions (potentially what I did not listed out) so that users will follow and understand the limitations of your analysis.

3. Should design more experiments that

- (a) During off-policy DPO, show that no other sentence's log probability significantly increased other than y^*. Although your theoretical analysis seems to have proved this, we still need to show this point in experiments due to your analysis assumptions. The sampling should be much more extensive to show the non-existence of such examples (and statistically significant)
- (b) For SFT, quantitatively show the causal effect of increased hallucinations due to the "pull up" effect of similar but negative examples (scaling up more than MNIST)
- (c) For both experiments, show that neuron tangent kernels (your K-term) did not change much, and thus the learning dynamics can be fully attributed to your G-term
- (d) Qualitatively demonstrate the repetitiveness of off-policy DPO towards the end of fine-tuning
- (e) Quantitatively measures how your proposed approach reduced hallucinations / answer repetitiveness via more rigorous benchmarks, other than just a measurement of the win rate

**Questions:**

1. In your page 5, "make any no assumptions" -> "make no assumptions"
2. In your appendix A, add more derivations to show that -(e_y_u^+ / pi)^T [A(x_u)]_l = pi(y_l | x_u) - e_y_u^+. All parts in Appendix A until this point is easy to follow.
3. In your page 16, lines 712-713, are there some nablas that are left behind?

---

> ### Author Response · Authors · 2024-11-21
> **Pointwise resposne (1/2)**
>
> Thank you for your great comments and efforts in reviewing our paper. Please also refer to the overall response, in addition to the response below.
>
> > 1. Need a "related work" section. …
>
> Thanks very much for pointing this out. Please also refer to the third section in the overall response. Due to the page limits, we added only the related work section in Appendix E (the last appendix in the original version). We agree with the reviewer that offering a more thorough related work section would help ICLR readers better. The new related work section will not only cover learning dynamics and LLM finetuning but also mention that the “peakiness effect” was previously observed (see the third point of the response to the reviewer 4pPy), where we find their experiments match our analysis well.
>
> > 2. Need to indicate assumptions in your Prop 1 and Section 3.2 clearly …
>
> As the assumption we made on the eNTK term is mentioned by all the reviewers, we explain this in the first section of the overall response, and will make this prominent in the revised paper.
>
> The statement of Proposition 1, however, is fully accurate; the lower-order terms are unambiguously $\\mathcal O(\\eta^2)$ as in the statement. Since the learning rate is usually quite small during finetuning, over a short time period the first-order effects should dominate; over a longer time period, other factors can grow important. This is in fact the same concern as in the previous section; if we really followed only the first-order terms, the eNTK would remain constant. In adding our discussion of the previous point, we will make sure to emphasize this.
>
> > 3.a Should design more experiments … During off-policy DPO, show that no other sentence's log probability significantly increased …
>
> Unfortunately, the Y space is far too large for us to evaluate the probability of all other responses as we did for the MNIST experiment – which is probably why it was not previously noticed that it was the greedy decoding that took most of the increase during DPO. Towards verifying this, however, we will add the following discussions and experiments in the next section (details in Figure F.4 in the rebuttal appendix).
>
> For the greedy $y^*$, we also track the “except-argmax” probability, i.e., besides tracking  $\\pi(y^*\|x\_u)=\\sum_l p(y^*\_l|y\_{<l}^+)$, we also track $\pi(y^*|x\_u) ^C\triangleq\sum\_l p(y\neq y\_l^*|y\_{<l}^+)$ and compare them in the same figure. It is clear that $\\pi(y^*|x\_u) ^C$ decreases as $\\pi(y^*|x\_u)$ increases. The important thing is that for DPO, we find $\\pi(y^*|x\_u) ^C$ ends up much smaller than in the SFT case. Note that this term is calculated by summarizing all other $V-1$ dimensions. In other words, during the late stage of DPO, for each token’s prediction, the model’s confidence in each token is greater than 0.5, which makes it unlikely for the existence of other high-likelihood responses.
>
> Furthermore, we can also infer the non-existence of higher-confidence sequences by thinking one step further from our original results. Recall we mentioned that in the DPO case, the log probability of $y^*$ is approaching -63 in Line 406. Since the average response length of $y_u^+$ is 261 in our probing dataset, the model’s average confidence on each token is roughly $e^{-63/261}\\approx 0.79$, much highher than 0.5. Hence we believe it is unlikely to have to have other sequences with significantly higher confidence than $y^*$.
>
> > 3.b For SFT, quantitatively show the causal effect of increased hallucinations due to …
>
> Thanks very much for this suggestion. We first apologize for the inadvertant over-claim in our original version that the “pull-up” effect explains why hallucination occurs. What we want to claim is a hypothetical explanation of a specific type of hallucination, i.e., using the answer or phrases in question B to answer question A, since the confidence on $y\_{j\neq i}$ keeps increasing during the SFT stage. We absolutely do not claim that this is representative of all or even most types of model hallucinations. We looked at related work on quantitatively analyzing the hallucination effect, e.g., [1], but those we read rely more on evaluating whether the answer provided by the model comes from the data or its internal information. The hallucination type we mentioned is more about whether the answer provided by the model wrongly mixes the facts provided by the dataset, which is a bit harder to directly observe. We think learning dynamics and our particular results might provide a useful jumping-off point for further exploration of this important effect, but it certainly does not trivially solve the problem.
>
> (**...To be continued**)

---

> > ### Author Response · Authors · 2024-11-21
> > **Pointwise resposne (2/2)**
> >
> > > 3.c For both experiments, show that neuron tangent kernels (your K-term) did not change much …
> >
> > We totally agree that directly observing the norm of the K term could make our analysis more solid. As we mentioned in the first section of the overall response, however, calculating the eNTK term for LLM is too computationally complex; roughly speaking, it requires computing and multiplying vectors of length several billion for each kernel entry.
> >
> > However, since our true assumption is only the “relative stability” of this term, it is possible for us to indirectly measure the relationship of $\\|K^t\_{uo}\\|\_F$ across different $o$ by tracking a lower bound. Specifically, we derived two metrics estimating the lower bound of the K term, and tracked their stability instead. In calculating them, we directly track $\\|A\_o^t\\|_F$, $\\|G^t\_u\\|\_F$, $\\|\Delta\_o^t\\|\_F$, and find that they all match our analysis well. Although we only plan to add experiments on these two metrics in the next version due to the page limits, we can provide the details of other intermediate terms if it would be helpful.
> >
> > > 3.d and e Qualitatively demonstrate the repetitiveness of off-policy DPO towards the end of fine-tuning … Quantitatively measures how your proposed approach reduced hallucinations …
> >
> > Thanks very much for the suggestion. We have added more experiments discussing repetitiveness in the second section of the overall response. However, since evaluating the proposed method is not the main focus of this paper (the hyperparameters for the proposed methods are also not carefully tuned), and the proposed method plays a more important role in verifying the squeezing effect (Figure 5 of the original version), we only provide some early-stage results to show its effectiveness in the rebuttal appendix. A more refined version of the proposed method (e.g., selecting the SFT data samples more carefully) is a good area for future work.
> >
> > > 4. In your Appendix A, add more derivations to show Eq.11. …
> >
> > Thanks very much for pointing out this, please refer to Appendix F.3 for more details.
> >
> > > 5. In your page 5, "make any no assumptions” … In your page 16, lines 712-713 …
> >
> > Thanks very much for pointing out these typos. We will fix them in the next version. The nablas should be in front of the L_sft term in lines 712-713 as well.
> >
> > [1] Orgad, Hadas, et al. "LLMs Know More Than They Show: On the Intrinsic Representation of LLM Hallucinations." arXiv 2024.

---

> > > ### Author Response · Authors · 2024-11-29
> > >
> > > Hi,
> > >
> > > Since there's not long left in the discussion period, we wanted to see if you have any thoughts on our responses (the two comments above as well as the general response). We think we've mostly addressed your concerns, including running several new experiments, although a few of your suggestions seemed difficult to fully implement without major amounts of work. Do you have any remaining concerns or things we should address to improve the paper?
> > >
> > > Thanks again for your feedback!

---

### Official Review · Reviewer_4pPy · 2024-10-31

**Soundness:** 4
**Presentation:** 4
**Contribution:** 3
**Rating:** 8
**Confidence:** 3

**Summary:**

The learning dynamics of a linear-softmax classifier on top of fixed features is analyzed. The method is applied to the settings of LLM finetuning, by considering particular loss/objective functions specific to this settings and analyzing their effect on learning dynamics. The results helps elucidating previously observed phenomenon in LLM finetuning, and constructing new methods for overcoming potential "bottlenecks" in the process.

**Strengths:**

The paper is overall well written, clearly explains the motivation and the approach. The use of simple "pedagogical" examples also helps in communicating the main message of the paper.

The main contribution seems to be in the application of the theory to the different loss functions used in LLM finetuning, making explicit use of the "decomposition" of the effective learning dynamics into 3 different terms, and identify which one directly depends on the loss function being used. This is demonstrated to be productive as it can provide (at least a preliminary) explanation for different observed behaviors.

Overall, this is an interesting paper and the results are likely to be of interest to the community.

**Weaknesses:**

There are two major limitations here that, while are (somewhat) acknowledged by the authors, can benefit from a more careful discussion and/or analysis.

1. The first is the obvious limitation that the entire analysis is being done under the assumption that the "feature map" is held fixed, and only the classification layer is changing during learning, but the extent to which this assumption holds in practice is not being evaluated at all. Without such an evaluation (for example, either by quantifying/tracking the change in $\mathcal{K}^t$, or by repeating some experiments when the weights are *actually* frozen) it is hard to say whether the theoretical explanation about the squeezing effect is in fact an important part of the observed phenomena (or whether it is largely / to some extent driven by more complicated dynamics that involves changes in the feature map as well). I think this should, at the very least, be acknowledged more explicitly in the text.

2. The second limitation is more nuanced, and has to do with the relying of some intuitive understanding of "similarity" (between prompts/examples), that may or may not be valid. Again no evaluation is offered -- despite the fact that the authors explicitly specify how this similarity should be understood (i.e., it is measured by the eNTK, $\mathcal{K}$ ). It is far from being clear a-priori that the learned features agree with some intuitive judgement of similarity. Moreover, what "dissimilar" means, in the extremely vast space of strings, is also non-trivial. For example (and as pointed out by the authors), it might be that all "well-formed"/grammatical sentences are somewhat similar, and the model is able to push probability mass to entirely non-grammatical/nonsensical strings. This leads to some arguments being made only on the basis of some intuition. For example, "the model’s confidence on y+u keeps increasing and the update energy... gradually decreases"  (Line 350) -- this seems to be inconsistent with the fact that we see no slow-down of the updates for $\mathbf{y}^+_{u}$ itself.

Finally I would also like to point out that peakiness (/"squeezing") effect in text generating models followed by RL tuning is a phenomena that has, in fact, been observed and to some reason explained before, including the observation that the on/off policy choice might play an important role in dealing with the "exposure bias". The paper will benefit from referring to some of these previous works, for example:
- Caccia et al., Language GANs Falling Short, (arxiv 2018 / ICLR 2020)
- Choshen et al., On the Weaknesses of Reinforcement Learning for Neural Machine Translation (arxiv 2019 / ICLR 2020)
- Kiegeland and Kreutzer, Revisiting the Weaknesses of Reinforcement Learning for Neural Machine Translation (naacl 2021)

**Questions:**

Line 31: "Many interesting phenomena" -- such as? (not saying there aren't, but this is an odd statement to have without providing at least some refs)

Line 61: I feel that this is conflating different things, i.e some notion of "interpretability" (some "compact" understanding of how model predictions/behavior are determined by the parameters), and notions of learning dynamics (how model parameters evolve during training). One can in principle discuss the one without the other, and it would be better to spell out more clearly / explicitly the current motivation.

Figure 1: The red arrows on panel a, top do not actually represent $\Delta\theta$ but rather $\Delta \pi$. It's also not clear why a continuous curve is plotted, given the fact that $\pi$ is discrete -- why not show all 10 digits as bars (as is later done in panels (c) and (d))? Perhaps most importantly, it's not clear whether or not this figure should be understood only qualitatively, or is this the result of an actual computation.
The same comments apply for all panels in Fig.1 (b). If this is an actual result then many details are missing, both in presentation (bars; scale of the y-axis) and perhaps more crucially, at what stage of learning this is done. The entire argument relies on some implicit assumption that "4" images are similar to "9" images, but this crucially depends on the current learned representation of the model (since this is a "feature-learning" rather than "lazy" regime). If this is just an illustration, then it is highly misleading presenting it along with panels (c) and (d) which seems to be actual numeric results.
I'm also unable to understand how to interpret panel (c). The caption says "accumulated change" but the the charts seems to indicate absolute probabilities. It's also unclear what is being tracked, over how many epochs, and what are the examples being presented. The main text (lines 134--138) is unhelpful and reads as another explanation of what's actually illustrated in panels (a)-(b) (i.e., the single-step influence)
Overall, the entire discussion of the motivating example in Section 2 is hard to follow. I think the overall intention of illustrating the idea in a simple example is a good one, but the example itself should be made much more clear.


Line 233: "Any no assumptions" -> "no assumptions"


Line 253: Calling this "unexpected behavior" might be too strong, see previous literature
Line 314: "Fine grind" -> "Fine grained"

---

> ### Author Response · Authors · 2024-11-21
> **Pointwise resposne (1/2)**
>
> Thank you for your great comments and efforts in reviewing our paper. Please also refer to the overall response, in addition to the response below.
>
> > 1. The first is the obvious limitation that the entire analysis is being done under the assumption that the “feature map” is held fixed …
>
> Thanks very much for this question, which helped us realize what the assumption we actually need is. Please see the first point in the overall response, which thoroughly discusses the core assumption we need (which is far weaker than that the feature map is fixed). Furthermore, in the rebuttal appendix (Appendix F), we derived two metrics estimating the lower bound of the K term, and tracked their stability instead. In calculating them, we directly track $\\|A\_o^t\\|\_F$, $\\|G^t\_u\\|\_F$, $\\|\Delta\_o^t\\|\_F$, and find that they all match our analysis well. Although we only plan to add experiments on these two metrics in the next version due to the page limits, we can provide the details of other intermediate terms if it would be helpful.
>
> > 2.1 The second limitation is more nuanced, and has to do with the relying of some intuitive understanding of "similarity" …
>
> This is a great point that we had to think seriously about. The relative strength of the eNTK is an objective quantity which can (at least conceptually) be computed, and it is indeed (up to a first-order approximation) exactly what “similarity” means to the current model's gradient updates. How thoroughly this aligns with our human intuition about “similarity”, however, is much less clear, and some of our discussions certainly relied on intuitive notions of similarity rather than measuring the eNTK. This measurement is easier for the MNIST example, as the CNN extracts semantic information fairly similar to our intuition, and we find that “4” and “9” are similar in both intuition and experiments. For language data, on the other hand, there are many aspects of similarity that can easily be task-relevant. For instance, two sentences can be similar in their meaning or in their form. This is why we created two rephrases by ChatGPT to try to find some hints. The original Figures 3 and 4 verify our intuition that natural language is dissimilar to random sequences, the chosen response and its rephrases are more similar than the rejected ones, etc. The fact that the LLM relies more on semantic similarity is also mentioned in related work.
>
> In our new experiments in Appendix F.2 (inspired in large part by this question), though, we discovered a new similarity in “where the sequence is generated”. In Figure F.3, it is clear that learning examples generated from ChatGPT influence more on those rephrases than its original response. A fuller intuitive understanding of similarity as defined by the eNTK remains an open problem; we plan to explore this further in the future.
>
> > 2.2 For example, "the model’s confidence … (Line 350)
>
> This is another very good point. As demonstrated by Figure F.5-(b) in Appendix F, we see that the increase of $\\pi(y\_u^+)$ does indeed slow down a bit after the 2nd epoch; this is difficult to see by eye, but holding a piece of paper up to the endpoints makes it clear there is some curve to it. Furthermore, it is true that from the curve of  $\\log\\pi^t$ (as in the figures in the original version), we cannot see a clear drop in the updating energy. That is perhaps because the term $A\_o^t$, whose Frobenius norm is $\\|A\_o^t\\|\_F = V \\|\pi^t\\|\_2^2 + V - 2$, will also gradually increase as $\\pi^t$ becomes peakier (see the derivation in Appendix F.2). Note that the 2-norm of a one-hot distribution is one, while the 2-norm of a uniform distribution is $\\frac{1}{\\sqrt{V}}$. So, even though the upwards pressure indeed decreases, it is partially mitigated by the increase in $\\|A\_o^t\\|\_F$.
>
> (**...To be continued**)

---

> > ### Author Response · Authors · 2024-11-21
> > **Pointwise resposne (2/2)**
> >
> > > 3. Finally I would also like to point out that peakiness (/"squeezing") effect …
> >
> > These papers are extremely interesting; thank you for pointing to them, and we will absolutely discuss them in the revised version. One nice thing is that they in fact provide significant support for our analysis, from a different setting. For example, Equation 1 of [1] shows that we are minimizing a negative loss in a standard GAN, which with our arguments might help explain the peakiness. Furthermore, in Table 1 and Table 2 of [2], we see the peakiness (measured by $\\Delta p\_\\text{top10}, \Delta p\_\\text{mode}$) of the “PG-average” method is stronger than the standard PG method. Note that the “PG-average” method will map a reward ranging from 0 to 1 to a centered one ranging from -0.5 to 0.5; since a negative reward can introduce a negative gradient, the peakiness increases.
> >
> > However, we would also emphasize that the “harmful squeezing effect” discussed in this paper has another important factor: that it is applied to an already-unlikely output. As demonstrated by our Figure 16, imposing a negative gradient on the likely output only introduces a mild adaptation, while doing it on an unlikely output causes drastic squeezing. Since the training examples in both on-policy DPO and the scenarios of [1,2,3] are on-policy and hence reasonably likely under the current model, the peakiness effect mentioned there is not that harmful (or even beneficial). The method we proposed in section 4.3 is exactly trying to convert a harmful peakiness effect to a benign or even beneficial one, by “pulling up” the unlikely outputs before DPO.
> >
> > > 4. Line 31: "Many interesting phenomena” …
> >
> > Thanks for this question. We will mention some interesting phenomena like zigzag [4], auto-noise fixing [5], grokking [6], and many related works discussed in the workshop [7] from the most recent ICML. We hope this list can be further extended.
> >
> > > 5. Line 61: I feel that this is conflating different things …
> >
> > True; we will change the discussion and narrow down the scope to “how prediction changes given an update”. We do not particularly focus on the values of the parameters, but we instead use the parameters as a bridge to connect learning and the changes in function space.
> >
> > > 6. Figure 1: The red arrows on panel a, top do not actually represent …
> >
> > This is a good point. In the next version, we will change the figure to Figure F.6. There, plots in (b) are the real categorical distributions that come from one update (with a very large learning rate for visibility). (c) now compares the initial distribution and the final distribution, to show the accumulated influence. Note that the distribution is averaged on all images for their classes. (d) shows the final distribution.
> >
> > For concerns about lazy vs feature learning regimes, please refer to the first section in the overall response. The experiments in Appendix F verify the fact that the model’s similarity aligns with our intuition to some extent. These similarities often align well with human intuition, e.g., a cat is more similar to a dog than to a truck.
> >
> > > 7. Line 233:  … Line 253 … Line 314 …
> >
> > Thanks for pointing them out. We will fix those in the next version.
> >
> > [1] Caccia et al., Language GANs Falling Short, ICLR 2020
> >
> > [2] Choshen et al., On the Weaknesses of Reinforcement Learning for Neural Machine Translation, ICLR 2020
> >
> > [3] Kiegeland and Kreutzer, Revisiting the Weaknesses of Reinforcement Learning for Neural Machine Translation, NAACL 2021
> >
> > [4] Ren et. al Better Supervisory Signals by Observing Learning Paths, ICLR 2022
> >
> > [5] Liu et. al Early-learning regularization prevents memorization of noisy labels, NeurIPS 2020
> >
> > [6] Liu et. al Omnigrok: Grokking Beyond Algorithmic Data, ICLR 2023
> >
> > [7] Workshop on High-dimensional Learning Dynamics (HiLD) [https://sites.google.com/view/hidimlearning/home](https://sites.google.com/view/hidimlearning/home)

---

> ### Comment · Reviewer_4pPy · 2024-11-25
>
> I thank the authors for their detailed response.
> I think the revisions proposed in the new rebuttal appendix contribute to a better understanding of the paper (and Fig 1 is definitely improved). The issue of what "similarity" is persists, but as the authors (rightly) mention -- this is, in effect, a larger open question for the field. And methods of the form proposed in this paper might eventually contribute to a better understanding of that question.
> Following the authors response I have raised the "soundness" score to 4, and overall I stand by my evaluation that this is a good paper and should be accepted.

---

> > ### Author Response · Authors · 2024-11-27
> > **Thanks for your reply.**
> >
> > Thanks very much for your reply. We will further improve the paper in the next version based on your suggestions.

---

### Official Review · Reviewer_yrnd · 2024-11-02

**Soundness:** 4
**Presentation:** 4
**Contribution:** 4
**Rating:** 8
**Confidence:** 3

**Summary:**

The paper extends the existing learning dynamics for deep learning systems to LLMs and  argues that doing so is a non-trivial problem due to high dimensionality and sequential nature of both inputs and outputs. The paper talks about a “squeezing effect” where running DPO for too many epochs causes outputs to deviate from ideal behavior.

**Strengths:**

1) The paper has strong theoretical and experimental backing for their analysis.
2) First paper to propose a framework extending learning dynamics to LLMs.

**Weaknesses:**

1) Did not find any specific weaknesses.

**Questions:**

How does the squeezing effect differ from overfitting with DPO?

---

> ### Author Response · Authors · 2024-11-21
> **Pointwise resposne**
>
> Thank you for your great comments and efforts in reviewing our paper. Please also refer to the overall response, in addition to the response below.
>
> > How does the squeezing effect differ from overfitting with DPO?
>
> Thanks very much for this insightful question. We think in both the squeezing effect and overfitting, the model becomes more confident on some predictions, i.e., the predicting probability becomes peakier. However, the main difference between them is **which dimension they are converging to**. From the first panel in Figure 2, overfitting will make the model remember all the details (even noise) in the dataset, and hence peak at the dimension of the supervisory signals (usually, the ground-truth labels). The squeezing effect, on the other hand, makes the model **amplify its own current beliefs**, whatever they may be, and hence peak at the dimension with the highest output prior to the update step (this is guaranteed to happen). Furthermore, as analyzed in Appendix C, the squeezing effect will decrease the probability mass on all the dimensions other than the most confident one, which means the catastrophic forgetting in the squeezing effect might be more serious than in overfitting. We believe this is also an interesting direction to explore in the future.

---

> > ### Comment · Reviewer_yrnd · 2024-11-22
> > **Response**
> >
> > My queries have been answered and I stand by my score. Thank you for your response.

---

> > > ### Author Response · Authors · 2024-11-27
> > > **Thanks for the reply.**
> > >
> > > Thanks very much for your reply. We will further improve the paper in the next version based on your suggestions.

---

### Official Review · Reviewer_5a38 · 2024-11-03

**Soundness:** 4
**Presentation:** 4
**Contribution:** 4
**Rating:** 10
**Confidence:** 4

**Summary:**

This work studies the learning dynamics of LLM finetuning by analyzing the way in which likelihoods assigned by the model to different completions of training prompts evolve throughout the finetuning process. By explicitly writing out the single step update rules in terms of the change in model parameters and change in the functional output of the model, and then leveraging the empirical NTK for the LM head/logit layer of the model, they show that one can analytically describe the sample wise influence dynamics of both the SFT and DPO objectives for LLMs. Corroborating prior observations of model distributions degenerating during (off-policy) DPO, their analysis describes a "squeezing effect" due to the performing of gradient _ascent_ during standard preference tuning, and they validate this and other qualitative predictions via the finetuning of 1B-3B parameter LLMs on carefully augmented preference tuning data.

**Strengths:**

1. They motivate and ground the study of training dynamics for LLMs well and highlight key difficulties in applying standard influence function style analyses to LLM finetuning and preference optimization, a highlight of the preliminary material.
2. The formalism and analysis in Section 3 is generally clear and well written. Figures 2 and the bullet point enumeration in 3.3 are concise presentations of the core claims of the analysis.
3. The experimental design of the empirical verification section is well done and cleanly visualized, particularly the use of completions $y^{+}$ gpts, gptf, test, hum.
4. Proposal for a mitigation to the "squeezing effect" (while not the main contribution of the paper) is well motivated by the analysis and simple to implement.

**Weaknesses:**

1. The clarity of Sec 3.1 when discussing the causal masking and teacher-forced production of the full next-token logits set could be improved, though this is admittedly tricky.  It is possible that for some readers, a diagram of the matrix structure here could be helpful since most papers don't tackle the more complex formulation of the influence problem and so readers may not be clear on it. (The reviewer imagines a teacher-forced causally masked model forward on input and label sequences X,Y as a forward on an augmented view of X,Y according to causal mask M, such that if the length of X+Y=N, the input has X' now has dim NxN (lower triangular consisting of all prefixes of the sequence X+Y) and we consider the output to be labels Y_i for each row of X' corresponding to the next token following the tokens in row X'_i.)

2. Additionally, while the analysis relies on the kernel values between the feature maps of $\mathcal{X_0}$ and $\mathcal{\tilde{X_u}}$ to implement the pressure transfer between various samples during learning, these intermediate values are not actually experimentally estimated. To concretely tie the analysis to the empirical observations, it would be useful to compute examples of the decomposition values for the terms in Eq 5 and Eq 7. If this intermediary relationship can be shown as clearly as the loss trends described in Figures 3 and 4, a stronger assertion of causality between the theoretical model described by Eq 5 and Eq 7 and the empirical learning dynamics can be made, strengthening the impact of the work.

**Questions:**

1. Can you discuss the analytic or empirical argument for "Peakier πθt suffer a more serious squeezing effect." in a bit more detail? I see arguments titles Claim 3A and 3B on pg 27, but they seem to be more intuitive that formal. It could be useful to separate the claims in the Sec 3.3 enumeration into those with direct proof/analytic evidence and those that are more informal or empirical -- this would not weaken the arguments, but rather making these subtle distinctions clear would increase the soundness of the presentation.

2. The color coding of the variables in section 2 and 3 is non-standard but fine if the authors like this (and no other reviewers mind it). However, it seems inconsistent to not continue this into Sec 4 and 5? Additionally, if there is a way to continue the scheme into the figures, especially Fig 2, that would make more full use of the fact that  the x and y's are colored in the related written sections.

---

> ### Author Response · Authors · 2024-11-21
> **Pointwise resposne**
>
> Thank you for your great comments and efforts in reviewing our paper. Please also refer to the overall response, in addition to the response below.
>
> > 1. The clarity of Sec 3.1 when discussing the causal masking …
>
> Thanks for pointing this out. We have added a figure demonstrating the causal masking and a short paragraph to explain this point more clearly; it is currently in Figure F.5a, and will be merged into the main paper in the next version.
>
> > 2. Additionally, while the analysis relies on the kernel values between the feature maps …
>
> Thanks for this great question. Since this concern also aligns with other reviewers, we discuss this point in the overall response. Specifically for your concern, in Appendix F.2, we derived two metrics estimating the lower bound of the K-term and tracked their stability instead. In calculating them, we directly track $\\|A\_o^t\\|\_F$, $\\|G^t\_u\\|\_F$, $\\|\Delta\_o^t\\|\_F$, and find that they all match our analysis well (Figure F.3 and F.4). Although we only plan to add experiments on these two metrics in the next version due to the page limits, we can provide the details of other intermediate terms if it would be helpful.
>
> > 3. Can you discuss the analytic or empirical argument for "Peakier …
>
> Thanks very much for pointing this out. It is true that the claims about the relationship between “how peaky $\pi$ is” and “how serious the squeezing effect is” (Claims 3A and 3B) are based on intuition and numerical simulations on equation (27)-(32). 3A is more formal because we can substitute the definition of uniform distribution for these equations, and find deterministic relationships. We will make the distinction between fully-analytical statements and those stemming from empirical results clearer in revision.
>
> > 4. The color coding of the variables in section 2 and 3 …
>
> Thanks for pointing this out. The coloring system indeed becomes more chaotic after section 3.1, as more notations (especially about DPO) are introduced. Our motivation for using these colors is to remind the readers that the orange thing is the observation input and the blue thing is the updating input. However, this color encoding contradicts those used in Fig.2 and other experimental results (e.g., chosen and rejected responses). In the next version, we plan to only use these two colors in section 2 (and the appendix), and change the others back to black.

---

> ### Comment · Reviewer_5a38 · 2024-11-26
> **Response to rebuttal**
>
> (apologies for the delay in responding)
>
> I appreciate the authors' efforts to add the additional diagram into a final draft, and the analyses for intermediate values in the influence metrics. I think that the F.3 panel is quite interesting and so maybe a concise summary chart with a few curves to be highlighted could be worked into the main body (or at least noted in poster/talks etc).
>
> Assuming inclusion of the other minor adjustments in terms of precise organization of claims and notational polishing in the main body discussed in this specific rebuttal, and in the general discussion with all reviewers, I think that this work is of high overall quality and potential impact, and is appropriate for the ICLR venue. It should be considered for highlighting at the conference.

---

> > ### Author Response · Authors · 2024-11-27
> > **Thanks for your reply.**
> >
> > Thanks very much for your reply. We will further improve the paper in the next version based on your suggestions.

---

### Author Response · Authors · 2024-11-21
**Overall Response (1/2)**

We appreciate all the reviewers' insightful feedback, which has helped us improve the paper further. Since there were some overlapping concerns across multiple reviewers, we’ll address those here. Please also refer to the individual responses. To ease the process of locating the modifications, we added a section called “Appendix Rebuttal” at the end of the new pdf; we will incorporate this into the rest of the paper for the final revision.

### 1. The “relatively stable eNTK” assumption

We first apologize for not clearly enough articulating the core assumption we made through our theoretical analysis. This gave the readers different understandings: some believe we assume the eNTK changes slowly, while others think we assume a constant eNTK. In fact, the assumption we really use on $K^t(x\_u,x\_o)$ (which we write $K^t\_{uo}$ for brevity) is:

**During training, the *relative* influence from $x_u$ to each other $x_o$ *is relatively stable.***

In other words, if we consider $K^t_{uo}$ as a similarity measurement from the model’s perspective, then in a reasonably short period, e.g., several epochs or updates during training, the relative similarity between $x_u$ and $x_o$ is almost unchanged. Figures F.2 and F.3 in the rebuttal Appendix are good examples of understanding this concept of “relatively stable”; the eNTK does change noticeably, but the relative influence of points changes much more slowly.

The “lazy-NTK” assumption is then a **sufficient but not necessary condition** for our analysis. In fact, in the MNIST example, where the model is trained from scratch rather than finetuned on a well-trained checkpoint, the model indeed does conduct “feature learning” where the eNTK changes significantly, as demonstrated in the last row of Figure F.2. However, as shown in other panels, as the relative similarity among examples during training (after 30 epochs) is stable, the “upwards” pressure is still consistent enough to form the “pairing effect” between “4” and “9”.

Since this is the core assumption of our paper, we will add more discussion of these points in the next version. The main points of the discussions are summarized as the following:

1. We will give a direct verification of the “relatively stable” assumption in the MNIST example. In Appendix F.1, we extend the MNIST experiment by directly tracking $\\|K^t\_{uo}\\|\_F$ of different sample pairs. Specifically, for each $x_u$, we calculate its eNTK term with all other $x_o$. By comparing the ranking relationships of these norms throughout different stages of the training, we see the relative similarity of different classes is almost unchanged, although the eNTK indeed changed a lot at some stages. Remember that our analysis only needs such a relative similarity to be stable, i.e., $\\|K^t\_{4,9}\\|\_F>\\|K^t\_{4,0}\\|\_F$ most of the time.

2. We will verify the “relatively stable” assumption in real LLM cases by tracking more metrics related to the eNTK term. Ideally, we would directly observe the stability of rankings or the norms of these eNTK terms in a real LLM. However, the huge number of parameters and depth of the model make the eNTK, or even approximations to it, prohibitively expensive to compute. Thus we instead track two quantities related to the eNTK term: a lower bound on its norm, and the sign of the influence it imposed on the model. Please refer to Appendix F.2 for more details about how these two metrics are derived. As demonstrated in Figures F.3 and F.4, we not only see the relative ranking of the influence is relatively stable in both SFT and DPO, but also find a “pairing effect” among different responses.

3. Last, we will show some evidence that supports the “lazy-NTK” condition (even though we do not require it), from results on small-scale models to related works on larger-scale systems. Specifically, by directly observing $\\|K^t\_{uo}\\|\_F$ and $\\|K^t\_{uo}-K^{t-1}\_{uo}\\|\_F$ during training in the MNIST experiment (i.e., the last row of Figure F.2), we find that compared with the early stage training phase, the converging stage (which is similar to the behavior of finetuning) indeed has a lazier eNTK. Extending these findings to LLM is a bit hard for us since pretraining an LLM is very time-consuming. Fortunately, the discussions in sections 5 and 6 of [1] support this claim well. The authors empirically showed that finetuning a RoBERTa-level model (~100M params) on real NLP datasets can be approximated by the lazy kernel regime. Furthermore, we find this claim (that late training stage or finetuning has lazy-NTK property) is also supported by a recent study about pretraining GPT-level LLM from scratch [2]. (**...To be continued**)

---

> ### Author Response · Authors · 2024-11-21
> **Overall response (2/2)**
>
> Specifically, in almost all the training curves demonstrated in [2], we observe a slow-change norm and relatively stable learning dynamics in the latter phase, which also aligns with Figure 6 in [3], which measures the eNTK change in an image classification problem. In summary, although it is hard to directly calculate the huge NTK matrix for the GPT-based model (>1.0B params), likely, the relaxed version of the relative-lazy-NTK assumption (i.e., the one we used) holds for different settings. We will also indirectly verify this using experiments later.
>
> ### 2. More about the evaluation of the repeating effect
>
> In the next version, we also add more results showing the relationship between repeatness, DPO, and the proposed method. Specifically, we use ChatGPT and Claude API as a judgment to count how many “non-human-like-repeated ” responses exist in the model’s response. Although we only show the example of one run in one setting, some vague trends still exist here. In the following table, we see the number of non-human-like-repeated indeed increased as the DPO continues (after SFT → DPO 2 epochs → DPO 6 epochs). Comparing the proposed method and baseline, the "repeater effect" is indeed slightly reduced. However, due to time constraints, we haven’t tracked whether false positive and false negative judgments exist (we believe there might be some). We would leave the more detailed experiments in our future work.
>
> |         |          | SFT | DPO2 | DPO6 |
> |:-------:|:--------:|:---:|:----:|:----:|
> |  Claude | baseline | 108 |  115 |  118 |
> |         | proposed | 115 |  111 |  111 |
> | ChatGPT | baseline | 114 |  121 |  118 |
> |         | proposed | 110 |  111 |  110 |
>
> ### 3. Limitations and related work discussion
>
> In the next version, we will put some experimental results in the appendix and add a more thorough related work section (a version is now in Appendix E, but we will indeed add more discussion), as well as further discussing the limitations of our theory.
>
> One important update is about the “benign peakiness” effect mentioned in [5-7], which strongly supports our discussions on the “squeezing effect”. The discussions in the previous work confirm our claim that the negative gradient is one important cause of the squeezing effect. For example, Equation 1 in [5] shows that we minimize a negative thing in a standard GAN loss, which might explain why peakiness occurs. Furthermore, in Table 1 and Table 2 of [6], we see the peakiness (measured by $\Delta p_{top10}, \Delta p_{mode}$) of the “PG-average” method is stronger than the standard PG method. Note that the “PG-average” method will map a reward ranging from 0 to 1 to a centered one ranging from -0.5 to 0.5. Since the negative reward can introduce a negative gradient, the peakiness increases.
>
> However, we would also emphasize that the “harmful squeezing effect” discussed in this paper has another important factor. As demonstrated by our Figure 16: imposing a negative gradient on a highly-likely point only introduces a mild adaptation, while one on a very-unlikely output leads to drastic squeezing. In on-policy DPO and the scenarios of [5-7], the negative losses are imposed on a point generated by the current model, hence one which is reasonably likely; thus the peakiness effect in those settings is not very harmful, or even beneficial. The method we proposed in section 4.3 exactly tries to convert a harmful peakiness effect to a benign or beneficial one by “pulling up” the valley before DPO.
>
> [1] Malladi, Sadhika, et al. "A kernel-based view of language model fine-tuning." ICML 2023
>
> [2] Li, Ming, et.al  "What Happened in LLMs Layers when Trained for Fast vs. Slow Thinking: A Gradient Perspective." arXiv 2024
>
> [3] Ren, Yi, et al. "How to prepare your task head for finetuning." ICLR 2023
>
> [4] Fort, Stanislav, et al. "Deep learning versus kernel learning: an empirical study of loss landscape geometry and the time evolution of the neural tangent kernel." NeurIPS 2020
>
> [5] Caccia et al., Language GANs Falling Short, ICLR 2020
>
> [6] Choshen et al., On the Weaknesses of Reinforcement Learning for Neural Machine Translation, ICLR 2020
>
> [7] Kiegeland and Kreutzer, Revisiting the Weaknesses of Reinforcement Learning for Neural Machine Translation, NAACL 2021

---

### Meta-Review · Area_Chair_akU7 · 2024-12-18

**Metareview:**

### Summary of the paper
The paper analyzes the learning dynamics of LLM finetuning by examining how model-assigned likelihoods to different prompt responses evolve during training. Key findings and contributions include:
- Development of an analytical framework to describe sample-wise influence dynamics for both SFT and DPO objectives using empirical NTK
- Identification of a "squeezing effect" during DPO training where gradient ascent causes model distributions to degenerate
- Studying cause of hallucination by demonstrating that when training on one response, similar responses get "pulled" toward similar prompts, even if incorrectly
- Finding out where the probability mass has gone in DPO: finding that during DPO, log likelihoods decrease for both positive and negative examples, with only the greedy decoding result showing increased likelihood

### Strengths
- Strong theoretical foundation with clear formalism and analysis
- Well-designed experimental verification
- Provides both theoretical analysis and experimental verifications for several unsolved but popular phenomenon in LLM finetuning, like "where has the probability mass gone in DPO", what is the cause of repeated patterns after finetuning, and the cause of hallucination (might have been better understood in previous works compared to the other two, but conclusions are consistent)
- Practical contribution through proposed mitigation strategy for the "squeezing effect"

### Weaknesses
- The theoretical analysis assumes fixed features with only classification layer changes. This assumption may not hold well in practice. However, the authors added the “relatively stable eNTK” assumption together with new experiments in the rebuttal to show the eNTK does change noticeably.
- Lack of quantitative demonstration of hallucination increases in SFT. But this has been shown in previous works.

**Additional Comments On Reviewer Discussion:**

The discussions were mainly around empirical verifications of the assumptions and further discussions on related works. The authors have addressed all these concerns.

---

### Decision · Program_Chairs · 2025-01-22

Accept (Oral)